# Evaporation, infiltration and storage of soil water in different vegetation zones in the Qilian Mountains: A stable isotope perspective

**Guofeng Zhu [1,2], Leilei Yong [1,2], Xi Zhao [1,2], Yuwei Liu [1,2], Zhuanxia Zhang [1,2], Yuanxiao Xu [1,2], Zhigang Sun [1,2], Liyuan Sang [1,2], Lei Wang [1,2]**

[1] College of Geography and Environment Science, Northwest Normal University, Lanzhou 730070, China

[2] Shiyang River Ecological Environment Observation Station, Northwest Normal University, Lanzhou 730070, Gansu, China

**Correspondence to:** Guofeng Zhu (zhugf@nwnu.edu.cn)

**Abstract:** The processes of water storage have not been fully understood in different vegetation zones in mountainous areas, which is the main obstacle to further understanding hydrological processes and improving water resource assessments. To further understand the process of soil water movement in different vegetation zones (alpine meadow, coniferous forest, mountain grassland, and deciduous forest) in mountainous areas, this study monitored the temporal and spatial dynamics of hydrogen and oxygen stable isotopes in the precipitation and soil water of the Xiying River Basin. The results show that the order of soil water evaporation intensities in the four vegetation zones was mountain grassland ($SWL_{slop}$: 3.4) > deciduous forest ($SWL_{slop}$: 4.1) > coniferous forest ($SWL_{slop}$: 4.7) > alpine meadow ($SWL_{slop}$: 6.4). The soil water in the alpine meadow and coniferous forest evaporated from only the topsoil, and the rainfall input was fully mixed with each layer of soil. The evaporation signals of the mountain grassland and deciduous forest could penetrate deep into the middle, and lower layers of the soil as precipitation quickly flowed into the deep soil through the soil matrix. Each vegetation zone's water storage capacity of the 0-40 cm soil layer followed the order of alpine meadow (46.9 mm) > deciduous forest (33.0 mm) > coniferous forest (32.1 mm) > mountain grassland (20.3 mm). In addition, the 0-10cm soil layer has the smallest soil water storage capacity (alpine meadow:43.0 mm; coniferous forest: 28.0 mm; mountain grassland: 17.5 mm; deciduous forest:

29.1 mm). This work will provide a new reference for understanding soil hydrology in arid headwater areas.

**Key words:** Xiying River; Stable isotope; Drought, Soil water storage

## 1. Introduction

In arid inland river basins, climate and vegetation changes will affect the hydrological cycle (Sharma et al., 2021; Tetzlaff et al., 2013). As an essential part of the water cycle, soil water in the unsaturated zone can be converted from precipitation into the stream or groundwater recharge. Determining soil water's evaporation, infiltration, and storage properties are critical to understand the regional hydrological cycle and water balance under climate and vegetation changes (Brooks et al., 2010; Dubbert and Werner, 2019; Grant and Dietrich, 2017).

As "fingerprints" of water, isotopes have been used to track ecohydrological characteristics, such as evaporation (Barnes and Allison, 1988; Zhu et al., 2021b), groundwater recharge (Koeniger et al., 2016), infiltration paths (Duvert et al., 2016; Tang and Feng, 2001; Zhu et al., 2021a), evapotranspiration distribution (Gibson et al., 2021; Xiao et al., 2018), and the water absorption by plants (Rothfuss and Javaux, 2017).

Water seepage in the unsaturated soil zone and water evaporation at the air–soil interface are the primary forms of soil water transport. The dynamic water process reflected by the displacement of the isotope signal on the soil profile is called the "memory effect". Understanding the "memory effect" will help us to trace the dynamic changes in climate and soil hydrology (Kleine et al., 2020). The change of stable isotopes in near-surface soil water may reflect the precipitation variation, but these variations decrease with depth unless there is preferential flow (Peralta-Tapia et al., 2015; Sprenger et al., 2016; Sprenger et al., 2017). Evaporation mainly occurred in the near-surface part of the soils (0-10 cm), and the light isotope molecules ($^1$H and $^{16}$O) evaporated preferentially, resulting in the enrichment of heavy isotopes ($^2$H and $^{18}$O) on the soil surface (Ferretti et al., 2003). Dansgaard (1964) proposed the concept of d-excess (d-excess=$\delta^2$H-8$\delta^{18}$O) to illustrate the intensity of evaporation

fractionation. Assuming that evaporation occurs in the atmosphere with a humidity of
75%, it shows that the d-excess value of atmospheric moisture accounts for the
d-excess value of 10‰ in the atmospheric moisture, which conforms to the worldwide
average isotopic labelling of meteoric waters. Landwehr and Coplen (2006) defined
line conditioned excess as the difference between the $\delta^2H$ value of the water sample
and the $\delta^{18}O$ linear transform value of the same sample, where the linear
transformation reflects the relevant referenced meteoric water relationship. Compared
with d-excess, lc-excess can explain the evaporative fractionation process better. The
main reason is that lc-excess of precipitation and soil water changes smoothly and has
relatively small seasonal changes (Landwehr et al., 2014). The dynamic changes of
isotopes record the signal of soil water evaporation. This enrichment of this dynamic
fractionation exists in soil water isotopes in different climatic regions. Compared with
temperate regions, the evaporation signals in arid and Mediterranean environments
penetrate deeper into the soil (Sprenger et al., 2016). After evaporation and seepage,
some water is stored in the soil . The water storage capacity in humid areas is higher
than that in arid areas, that in forest is higher than that in grassland, and that in surface
soil layer is lower than that in deep soil layers with high clay content (Kleine et al.,
2020; Milly, 1994; Snelgrove et al., 2021; Sprenger et al., 2019).
In alpine mountains, climate warming has accelerated the melting of glaciers and
frozen soil, and the dynamic interaction between water bodies stored in different
media has become the main influencing factor of the water cycle (Penna et al., 2018).
Interactions between precipitation and the soil-plant-atmosphere system determine the
distribution of water in various storage reservoirs and the subsequent release of water
vapor to the atmosphere. These interactions include mainly interception, throughfall,
canopy drip, snow accumulation and ablation, infiltration, surface and subsurface
runoff, soil moisture, and the partitioning of evapotranspiration between canopy
evaporation, transpiration, and soil evaporation. As the main links of the hydrological
cycle, these processes have a profound impact on regional water balance and flux
distribution.

In the past, studies on the evaporation, infiltration and storage of soil water mostly focused on vegetation types in the same climatic region or different climatic regions. Understanding the climatic and hydrological conditions of different vertical vegetation zones and clarifying the regulating role of vegetation in the water cycle can help better adapt to climate change's influences on the hydrological cycle in source areas. In this study, we monitored the stable isotope composition of precipitation and soil water and the spatio-temporal dynamics of soil water storage in four vegetation zones (alpine meadow, coniferous forest, mountain grassland, and deciduous forest) at different temperatures and humidity in the Xiying River basin. To explore the differences in soil water evaporation, infiltration, and storage processes in these four different climates, vegetation types, and terrain types, the following research objectives were proposed: (1) to explore the evolution of isotope evaporation signals and the "memory effects" of precipitation input, mixing and rewetting; (2) to understand the soil water storage capacity and influencing factors of four vegetation areas in mountainous areas. It is hoped that this study can further improve the understanding of the water cycle process and provide a scientific theoretical reference for water resource utilization and ecological restoration in fragile environments. More importantly, it can provide paradigms for research at different spatial scales (latitude zone, longitude zone, watershed, etc.) based on the knowledge of soil moisture evaporation, infiltration, and water storage in typical vertical vegetation zones.

## 2. Study area

The Xiying River originates from Lenglongling and Kawazhang in the eastern Qilian Mountains ($101°40'47''\sim102°23'5''$E, $37°28'22''\sim38°1'42''$N) (Fig. 1). As the largest tributary of the Shiyang River, it is formed by the Shuiguan River, Ningchang River, Xiangshui River, and Tatu River converging from southwest to northeast and ultimately flowing into the Xiying Reservoir. The average annual runoff of the Xiying River is 388 million $m^3$, which is mainly replenished by mountain precipitation and melting water of ice and snow. The runoff is mainly concentrated in summer. The basin elevation is between 2000 m and 5000 m, corresponding to a

temperate semiarid climate with strong solar radiation, a long sunshine time, and a large temperature difference between day and night. The average annual temperature of the basin is 6°C, the annual average evaporation is 1133 mm, the annual average precipitation is 400 mm, and the precipitation from June to September accounts for 69% of the annual precipitation. Precipitation increases with elevation, while temperature decreases with elevation in this area (Table 1) (Ma et al., 2018). The zonal differentiation of vegetation in the basin is dominated by deciduous forest, mountain grassland, cold temperate coniferous forest, and alpine meadow. The soils mainly include lime, chestnut, alpine shrub meadow, and desert soil (Fig. 1).

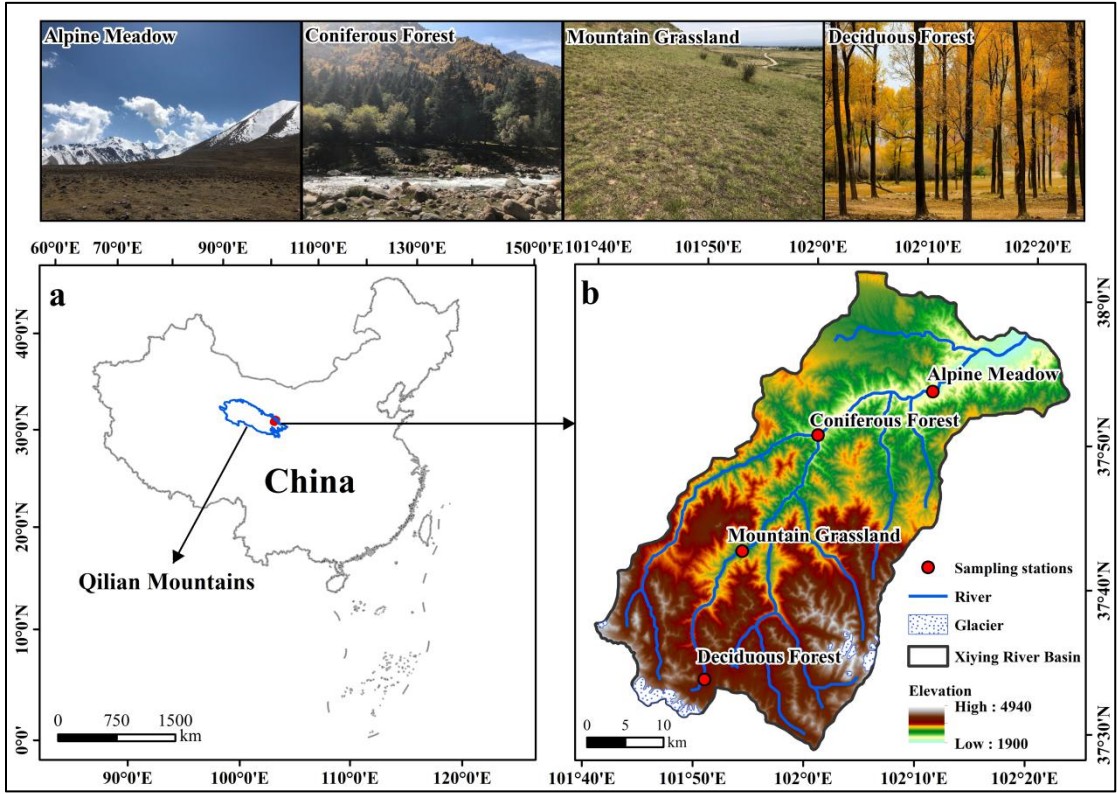

**Fig. 1** Study area and location of sampling points (a. The location of the Xiying River Basin in China; b. The terrain and sampling points of the Xiying River Basin)

## 3. Data and methods

### 3.1 Sample collection

In this study, soil water and precipitation samples were collected from four vegetation zones in the Xiying River basin from April to October in 2017 (plant growing season). In 2017, the precipitation in the alpine meadow, coniferous forest, mountain grassland and deciduous forest were 595.1 mm, 431.9 mm, 363.5 mm and

262.5 mm, respectively. The average daily temperatures in the alpine meadow,
coniferous forest, mountain grassland and deciduous forest were -0.19℃, 3.34℃,
6.6℃ and 7.9℃, respectively (Table 1).

Collection of soil samples: Soil samples were collected once a month at depths

of 0-10, 10-20, 20-30, 30-40, 40-50, 50-60, 60-70, 70-80, 80-90, and 90-100 cm from
the soil layers in the four vegetation zones. Three duplicate samples were collected for
each soil layer. We placed the collected soil sample into a 50 mL glass bottle, sealed
the bottle mouth with Parafilm and marked the sampling date. We froze the sample for
storage until experimental analysis. Each sample was collected separately in an
aluminum box.

Collection of precipitation samples: The precipitation samples were collected by

a plastic funnel bottle device. After each precipitation event, the collected
precipitation samples were immediately transferred to an 80 mL high-density
polyethylene bottle, and the bottle mouth of the samples was sealed with Parafilm;
these samples were also frozen and stored until experimental analysis.

Meteorological data: During the sampling period, the local meteorological data

were obtained and recorded by automatic weather stations (Watchdog 2000 series
weather stations) set up near the sample plot.

**Table 1** Basic data of each Vegetation zone from April to October 2017 (*Long*-Longitude,

*Lat*-Latitude, *Alt*-Altitude, *T*-Air Temperature (daily mean temperature), *P*-Precipitation (total
precipitation during the observation period), *h*-Relative Humidity (daily mean relative humidity))

| Vegetation zone | Geographical parameters | | | Meteorological parameters | | | Number of samples | |
|---|---|---|---|---|---|---|---|---|
| | *Long*(°E) | *Lat*(°N) | *Alt*(m) | *T*(℃) | *P*(mm) | *h*(%) | Precipitation | Soil |
| Alpine Meadow | 101°51'16" | 37°33'28" | 3637 | -0.19 | 595.1 | 69.2 | 72 | 47 |
| Coniferous Forest | 101°53'23" | 37°41'50" | 2721 | 3.34 | 431.9 | 66.6 | 42 | 41 |
| Mountain Grassland | 102°00'25" | 37°50'23" | 2390 | 6.6 | 363.5 | 60.4 | 37 | 54 |
| Deciduous Forest | 102°10'56" | 37°53'27" | 2097 | 7.9 | 262.5 | 59.8 | 40 | 53 |

**3.2 Sample determination**
The analysis of $\delta^2$H and $\delta^{18}$O values of all the above water samples was
completed using a liquid water isotope analyzer (DLT-100, Los Gatos Research, USA)
in the stable isotope laboratory of Northwest Normal University. Before analyzing the
isotope values of soil water, the soil water was extracted from the collected soil
samples by a low-temperature vacuum condensation system (LI-2100, LICA United
Technology Limited, China). Both the water and isotope standard samples were
injected 6 times during the analysis. To avoid the "memory effect" of isotope analysis,
we discarded the first two injection values and used the average value of the last four
injections as the final result (Penna et al., 2012; Qu et al., 2020). The analysis results
were relative to VSMOW (Vienna Standard Mean Ocean Water):

$$\delta = \left( \frac{R_{\text{sample}}}{R_{\text{standard}}} - 1 \right) \times 1000‰ \tag{1}$$

where $R_{sample}$ is the ratio of $^{18}$O/$^{16}$O or $^2$H/$^1$H in the sample and $R_{standard}$ is the ratio of
$^{18}$O/$^{16}$O or $^2$H/$^1$H in the VSMOW. The test error of the $\delta^2$H value does not exceed
±0.6‰, and the test error of the $\delta^{18}$O value does not exceed ±0.2‰.
**3.3 Analysis method**
**3.3.1 Lc-excess**
The linear relationship between $\delta^2$H and $\delta^{18}$O in precipitation and soil water is
defined as the LMWL (local meteoric water line) and SWL (soil waterline),
respectively, which are of great significance for studying the evaporative fractionation
of stable isotopes during the water cycle. We further calculated the line-conditioned
excess for each soil water and precipitation sample. The lc-excess in different water
bodies can characterize the evaporation index of different water bodies relative to the
local precipitation (Landwehr and Coplen, 2006).

$$\text{lc} - \text{excess} = \delta^2\text{H} - a \times \delta^2\text{H} - b \tag{2}$$

where $a$ and $b$ are the slope and intercept of the LMWL, respectively, and $\delta^2$H and
$\delta^{18}$O are the isotopic values of hydrogen and oxygen in the sample. The physical
meaning of lc-excess is expressed as the degree of deviation of the isotope value in
the sample from the LMWL, indicating the nonequilibrium dynamic fractionation
process caused by evaporation. Generally, the change in lc-excess in local
precipitation is mainly affected by different water vapor sources, and the annual
average is 0. Since the stable isotopes in soil water are enriched by evaporation, the
average lc-excess is usually negative (Landwehr et al., 2014; Sprenger et al., 2017).
**3.3.2 Potential evapotranspiration**

The potential evapotranspiration was calculated based on the Penman-Monteath

equation (Allen, 1998):

$$\text{PET} = \frac{0.408\Delta(R_n - G) + \gamma \dfrac{900}{T+273} u^2 (e_s - e_a)}{\Delta + \gamma(1 + 0.34u^2)} \tag{3}$$

where PET is the daily potential evapotranspiration (mm day$^{-1}$), $R_n$ is the net radiation
(MJ m$^2$ day$^{-1}$), $G$ is the soil heat flux density (MJ m$^2$ day$^{-1}$), $\gamma$ is the psychrometric
constant (kPa°C$^{-1}$), $u_2$ is the wind speed at 2 m height (m s$^{-1}$), $T$ is the mean daily air
temperature at 2 m height (°C), $\Delta$ is the slope of the vapor pressure curve (kPa°C$^{-1}$), $e_a$
is the actual vapor pressure (kPa) and $e_s$ is the saturated vapor pressure (kPa). These
data come from nearby weather stations.
**3.3.3 Soil water storage**

Soil water storage is the thickness of the water layer formed by all the water in a

certain soil layer (Milly, 1994) and is expressed by the following formula:

$$S = R \times W \times H \times 10 \tag{4}$$

where $S$ is the soil water storage in a certain thickness layer (mm), $R$ is the soil bulk
density (g cm$^{-3}$), and $H$ is the soil thickness (cm). $W$ is the gravimetric water content,
which is expressed by the following formula:

$$W = \frac{M_1 - M_2}{M_2} \times 100\% \tag{5}$$

in the formula, $M_1$ is the gravimetric value of wet soil (g), and $M_2$ is the gravimetric
value of dry soil (g).
**4. Results and analysis**
**4.1 Hydrological climate**

PET and runoff are important indicators that reflect the dry-wet conditions of river basins. During the study period (April-October 2017), in the Xiying River Basin, the potential evapotranspiration was 872.8 mm, the daily evapotranspiration ranged from 7.5 mm (July 14) to 0.9 mm (October 9), showing a fluctuating trend around July, and the PET value in April-July was higher than that in August-October. The input of summer precipitation and ice/snow meltwater increased runoff, resulting in a trend similar to PET. During the observation period, the total runoff was $3.1 \times 10^9$ m, accounting for 89% of the annual runoff. The variation range of the daily runoff was 286848 m³ (April 17) to 6125760 m³ (July 13). The basin before July was drier than that after July (Fig. 2).

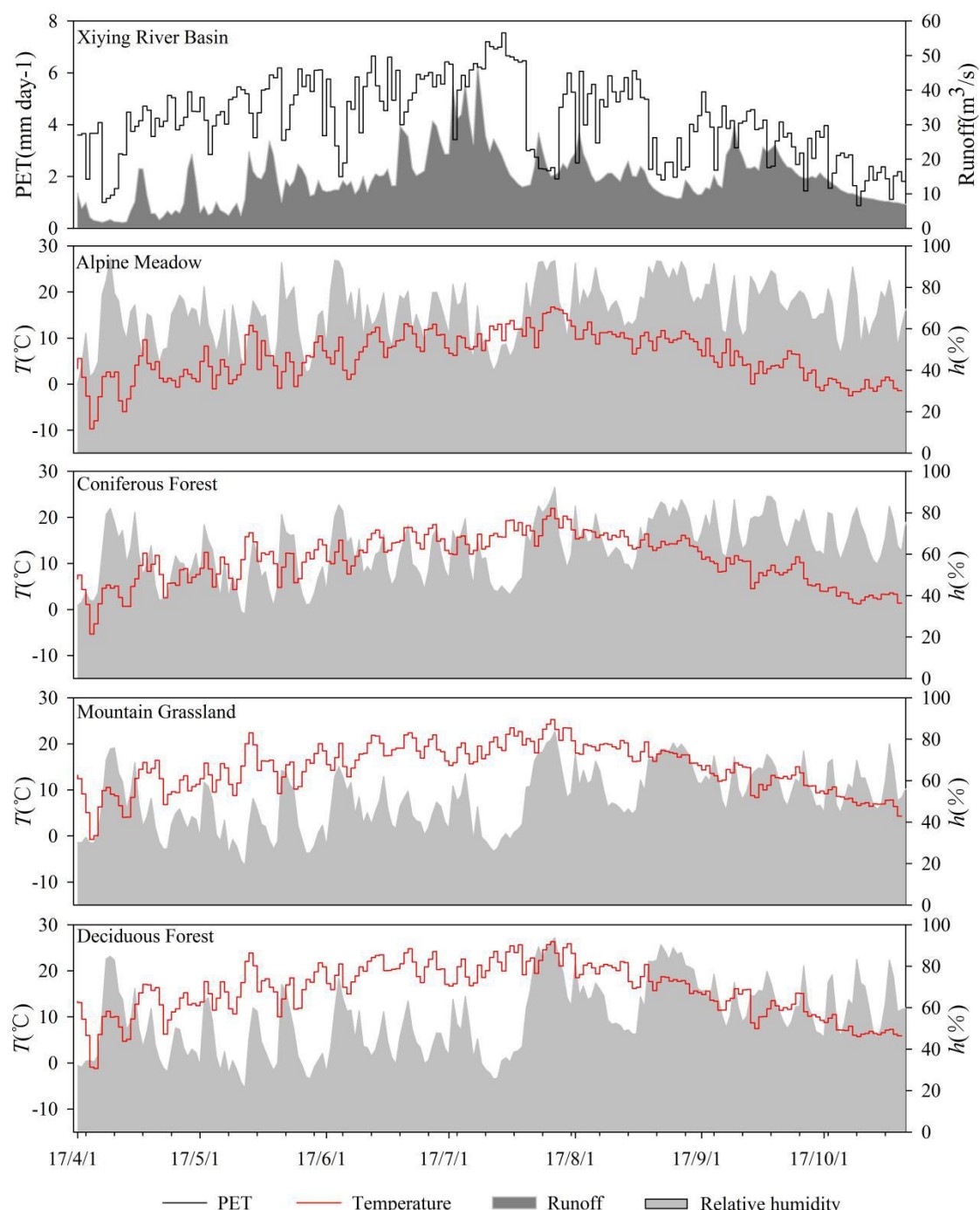

**Fig. 2** Climatic and hydrological conditions of Xiying River basin and its vegetation zones

To explore the differences in the natural environment in different vegetation zones, air temperature, atmospheric humidity, and precipitation were used to indicate each research site's temperature and moisture conditions. The hilltop is a typical alpine meadow zone, with a daily average temperature of 6.1°C, ranging from -9.7°C

(April 5) to 16.8°C (July 27). The daily average humidity was 68.2%, with little
difference in different periods. During the observation period, there were 72
precipitation events in the alpine meadow zone, and the total precipitation was 534.3
mm, which was relatively evenly distributed each month. In the coniferous forest zone,
the daily average temperature during the study period was 10.9°C, ranging from
-5.4°C (April 5) to 22.0°C (July 27). The daily average humidity was 62.5%, and the
precipitation was 400.6 mm, mainly concentrated from early August to late September.
Close to the foothills is the mountain grassland zone, with a daily average temperature
of 14.9°C, ranging from -0.7°C (April 5) to 25.3°C (July 27). The average daily
humidity was 51.1%, and the precipitation of the vegetation zone during the
observation period was 327.2 mm, mainly from late July to mid-August. During the
observation period, the daily average temperature in the deciduous forest zone was
15.8°C, ranging from -1.2°C (April 6) to 26.3°C (July 27). The daily average
humidity was 54.7%, and the total precipitation was 250.6 mm, which was
concentrated in the month from late July to late August. The temperatures of the
studied regions were ordered as follows: AM (alpine meadow) < CF (coniferous
forest) < MG (mountain grassland) < DF (deciduous forest). The humidities of the
studied regions were ordered as follows: AM > CF > MG > DF (Fig. 2).
**4.2 Temporal variation in water stable isotopes in different vegetation zones**
Influenced by different water sources and complex weather conditions in the
precipitation process, the isotopic compositions of precipitation in the four vegetation
zones were different during the study period. The mean values of $\delta^2H$ and $\delta^{18}O$ in the
alpine meadow zone (number of samples: 72) were -73.1‰±36.3‰ (-163.9~13.7‰)
and -10.0‰±4.3‰ (-23.1~-1.3‰), respectively. The average $\delta^2H$ and $\delta^{18}O$ values of
the coniferous forest zone (number of samples: 42) were -42.0‰±37.2‰
(-117.8~13.0‰) and -7.1‰±4.7‰ (-17.4~-0.1‰), respectively. The average $\delta^2H$ and
$\delta^{18}O$ values of the mountain grassland zone (number of samples: 37) were
-37.4‰±30.5‰ (-103.1~4.2‰) and -5.9‰±3.9‰ (-15.1~-0.9‰), respectively. The
average $\delta^2H$ and $\delta^{18}O$ values of the deciduous forest zone (number of samples: 40)
were -31.8‰±42.8‰ (-110.2~23.2‰) and -5.8‰±5.5‰ (-15.2~3.2‰), respectively
(Table 2). The maximum isotopic values of the four vegetation zones appeared on
August 4 (AM: 13.7‰, $\delta^2$H; -1.3‰, $\delta^{18}$O), August 10 (CF: 13.0‰, $\delta^2$H; -0.1‰,
$\delta^{18}$O), August 7 (MG: 4.2‰, $\delta^2$H; -0.9‰, $\delta^{18}$O) and August 13 (DF: 23.2‰, $\delta^2$H;
3.2‰, $\delta^{18}$O). The highest temperature in each vegetation zone appeared on July 27.
The high temperature caused the precipitation to undergo strong below-cloud
evaporation during the fall, leading to the enrichment of isotopes. In addition, the
atmospheric precipitation isotopes of the four vegetation zones had similar temporal
variations: from April to August, the fluctuations in $\delta^2$H and $\delta^{18}$O increased, reached
the maximum in mid-August, and then gradually decreased (Fig. 3).
**Table 2** General characteristics of precipitation $\delta^2$H and $\delta^{18}$O in different vegetation areas
from April to October 2017

| Vegetation zone | $\delta^2$H/‰ | | | | $\delta^{18}$O/‰ | | | |
|---|---|---|---|---|---|---|---|---|
| | Max | Min | mean | SD | Max | Min | mean | SD |
| AM | 13.7 | -163.9 | -73.1 | 36.3 | -1.3 | -23.1 | -10.0 | 4.3 |
| CF | 13.0 | -117.8 | -42.0 | 37.2 | -0.1 | -17.4 | -7.1 | 4.7 |
| MG | 4.2 | -103.1 | -37.4 | 30.5 | -0.9 | -15.1 | -5.9 | 3.9 |
| DF | 23.2 | -110.2 | -31.8 | 42.8 | 3.2 | -15.2 | -5.8 | 5.5 |

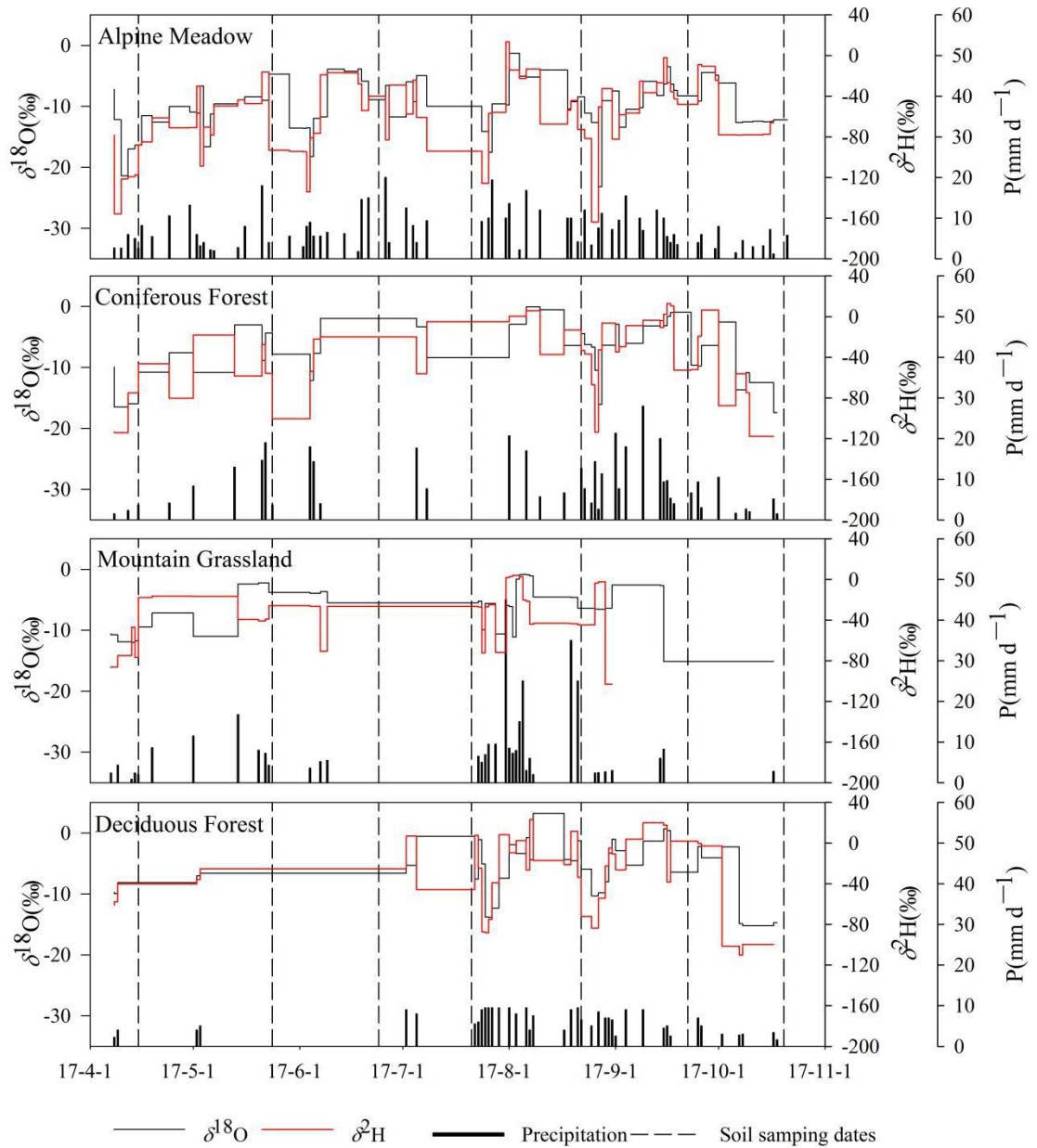


**Fig. 3** Time series of rainfall and isotope characteristics in different vegetation
zones in Xiying River Basin, with dotted lines indicating the date of soil water
sampling
The monthly variation in soil water isotopes records the signal of precipitation
input and evaporation. The low-temperature environment and abundant precipitation
events in the alpine meadow make the monthly average $\delta^2H$ and $\delta^{18}O$ of soil water
more depleted than other vegetation zones (-69.4~-51.6‰, $\delta^2H$; -7.5~-10.3‰, $\delta^2H$).
Despite this, the SWlc-excess of most samples at this station was still negative, and
there were different degrees of evaporation in the process of precipitation penetrating

the soil and mixing with original pore water, among which evaporation fractionation was stronger in July (-11.9‰ lc-excess) and October (-14.5‰ lc-excess). The soil water isotopes of the coniferous forest gradually changed seasonally. From April to July, precipitation was scarce, the temperature rose, and the isotopes of soil water were gradually enriched on the surface (-52.7~-29.5‰, $\delta^2$H; -7.0~-2.1‰, $\delta^2$H), reaching the peak value of the observation period in July (-29.5‰, $\delta^2$H; -2.1‰, $\delta^{18}$O), and continuous rainfall input from late July to mid-August resulted in soil water isotope depletion (-57.0‰, $\delta^2$H; -8.1‰, $\delta^{18}$O). SWlc-excess was an obvious fractionation signal opposite to the trend of isotope change, reaching the lowest value (-26.3‰) in the sampling period in July, and the change in air temperature and precipitation controlled the evaporation intensity. From April to July, the isotopic value of surface soil water in the mountain grassland was higher ($\delta^{18}$O was greater than zero), and SWlc-excess was lower than -30‰. During this period, the evaporation and fractionation of shallow soil water were intense. Similar to in the coniferous forest, in the mountain grassland, the input of heavy precipitation from late July to mid-August led to the depletion of soil water isotopes. There was only sporadic rainfall in the deciduous forest from April to July, and the soil water isotopes were gradually enriched on the surface (-46.1~-18.2‰, $\delta^2$H; -4.7~0.2‰, $\delta^2$H), reached a peak in June when there was no rainfall event (-18.2‰, $\delta^2$H; 0.2‰, $\delta^{18}$O), and then became depleted (-53.2‰, $\delta^2$H; -5.2‰, $\delta^{18}$O). In addition, due to the influence of the Xiying Reservoir and vegetation coverage, the isotopic enrichment degree of soil water in this vegetation zone was lower than that in the mountain grassland. As the most intuitive form of water change, the gravimetric water content was always at a low value in July (AM: 21.0%; CF: 14.8%; MG: 11.9%; DF: 14.9%), when the evaporation was the strongest, and it was most obvious in shallow soil (Table 3) (Fig. 4).

**Table 3** General characteristics of soil water δ²H, δ¹⁸O, lc-excess and GWC in different

vegetation areas from April to October 2017

| Month | Vegetation zone | $\delta^2H$/‰ | | | $\delta^{18}O$/‰ | | | lc-excess/‰ | | | GWC/% | | |
|---|---|---|---|---|---|---|---|---|---|---|---|---|---|
| | | Max | Min | Mean | Max | Min | Mean | Max | Min | Mean | Max | Min | Mean |
| 4 | AM | -55.2 | -70.7 | -65.6 | -8.5 | -10.8 | -10.1 | -2.7 | -7.1 | -4.7 | 25.9 | 23.0.0. | 24.7 |
| | CF | -52.7 | -72.2 | -63.9 | -7.0 | -9.9 | -8.9 | -4.0 | -12.0 | -8.4 | 27.6 | 14.9 | 20.0 |
| | MG | -7.32 | -50.6 | -41.0 | 2.8 | -5.8 | -3.9 | -8.8 | -36.8 | -19.4 | 21.7 | 6.5 | 14.7 |
| | DF | -46.1 | -69.4 | -62.1 | -4.7 | -9.9 | -8.5 | -2.5 | -23.2 | -9.7 | 27.7 | 19.4 | 21.8 |
| 5 | AM | -46.1 | -76.5 | -66.4 | -7.4 | -12.2 | -10.1 | -2.6 | -7.7 | -4.9 | 32.6 | 23.2 | 28.9 |
| | CF | -45.8 | -61.9 | -53.5 | -5.3 | -8.4 | -7.0 | -9.3 | -17.7 | -13.0 | 22.6 | 9.0 | 16.1 |
| | MG | -6.7 | -47.3 | -39.2 | 2.9 | -6.5 | -4.3 | -4.5 | -36.2 | -14.4 | 15.7 | 7.6 | 11.2 |
| | DF | -30.8 | -63.5 | -53.8 | -1.9 | -9.4 | -6.9 | -3.2 | -30.1 | -13.6 | 26.0 | 11.7 | 17.7 |
| 6 | AM | -62.5 | -83.9 | -69.4 | -8.9 | -12.6 | -10.3 | -1.5 | -8.4 | -5.8 | 33.3 | 21.9 | 26.0 |
| | CF | -45.8 | -78.4 | -58.7 | -5.1 | -12.0 | -7.8 | 5.5 | -26.6 | -8.5 | 32.1 | 10.0 | 21.3 |
| | MG | -19.7 | -74.9 | -46.9 | 0.8 | -11.8 | -5.8 | 13.0 | -33.7 | -11.0 | 19.3 | 7.5 | 14.2 |
| | DF | -18.2 | -64.9 | -51.7 | 0.2 | -9.0 | -5.9 | -4.6 | -38.2 | -19.4 | 13.5 | 8.4 | 11.1 |
| 7 | AM | -47.3 | -60.1 | -51.6 | -6.9 | -8.4 | -7.5 | -8.8 | -14.8 | -11.9 | 25.4 | 19.0 | 21.0 |
| | CF | -29.5 | -51.4 | -41.6 | -2.1 | -7.9 | -5.6 | -2.6 | -26.3 | -11.2 | 24.3 | 7.2 | 14.8 |
| | MG | -10.6 | -48.4 | -39.2 | 2.3 | -6.4 | -4.1 | -5.8 | -35.8 | -16.1 | 18.7 | 6.3 | 11.9 |
| | DF | -35.1 | -69.0 | -54.1 | -1.7 | -8.7 | -5.5 | -14.8 | -35.3 | -24.5 | 18.2 | 11.8 | 14.4 |
| 8 | AM | -58.5 | -80.3 | -66.6 | -8.4 | -11.6 | -9.6 | -6.1 | -15.4 | -9.7 | 28.1 | 19.5 | 25.1 |
| | CF | -57.0 | -75.5 | -66.4 | -8.1 | -9.8 | -9.2 | -2.5 | -13.1 | -8.3 | 21.4 | 8.7 | 16.3 |
| | MG | -34.2 | -53.8 | -44.0 | -3.2 | -5.5 | -4.4 | -14.7 | -22.6 | -18.7 | 11.3 | 9.5 | 10.4 |
| | DF | -53.2 | -84.3 | -67.6 | -5.2 | -13.5 | -9.2 | 6.8 | -26.1 | -9.6 | 23.6 | 14.7 | 20.6 |
| 9 | AM | -48.0 | -79.2 | -61.0 | -7.8 | -11.1 | -9.2 | -4.3 | -10.4 | -7.2 | 29.9 | 20.3 | 25.3 |
| | CF | -52.5 | -67.7 | -60.7 | -7.8 | -10.1 | -8.8 | -0.1 | -11.3 | -6.0 | 31.3 | 9.3 | 20.5 |
| | MG | -32.3 | -45.3 | -38.8 | -3.5 | -4.4 | -4.0 | -9.1 | -23.8 | -16.5 | 15.3 | 9.1 | 13.0 |
| | DF | -30.5 | -77.0 | -59.8 | -3.1 | -11.4 | -8.2 | -1.8 | -19.3 | -9.3 | 25.8 | 14.7 | 19.1 |
| 10 | AM | -42.4 | -73.5 | -58.9 | -6.1 | -9.8 | -7.9 | -12.2 | -18.2 | -14.5 | 36.2 | 25.4 | 29.5 |
| | CF | -59.1 | -66.3 | -61.7 | -8.8 | -10.5 | -9.5 | 5.1 | -5.3 | -1.5 | 30.0 | 16.8 | 23.1 |
| | MG | -50.3 | -66.7 | -58.3 | -5.6 | -8.3 | -7.1 | -5.5 | -18.4 | -11.9 | 18.3 | 11.4 | 15.8 |
| | DF | -38.0 | -61.8 | -48.3 | -2.7 | -8.2 | -4.9 | -11.9 | -34.8 | -23.9 | 25.5 | 8.9 | 17.2 |

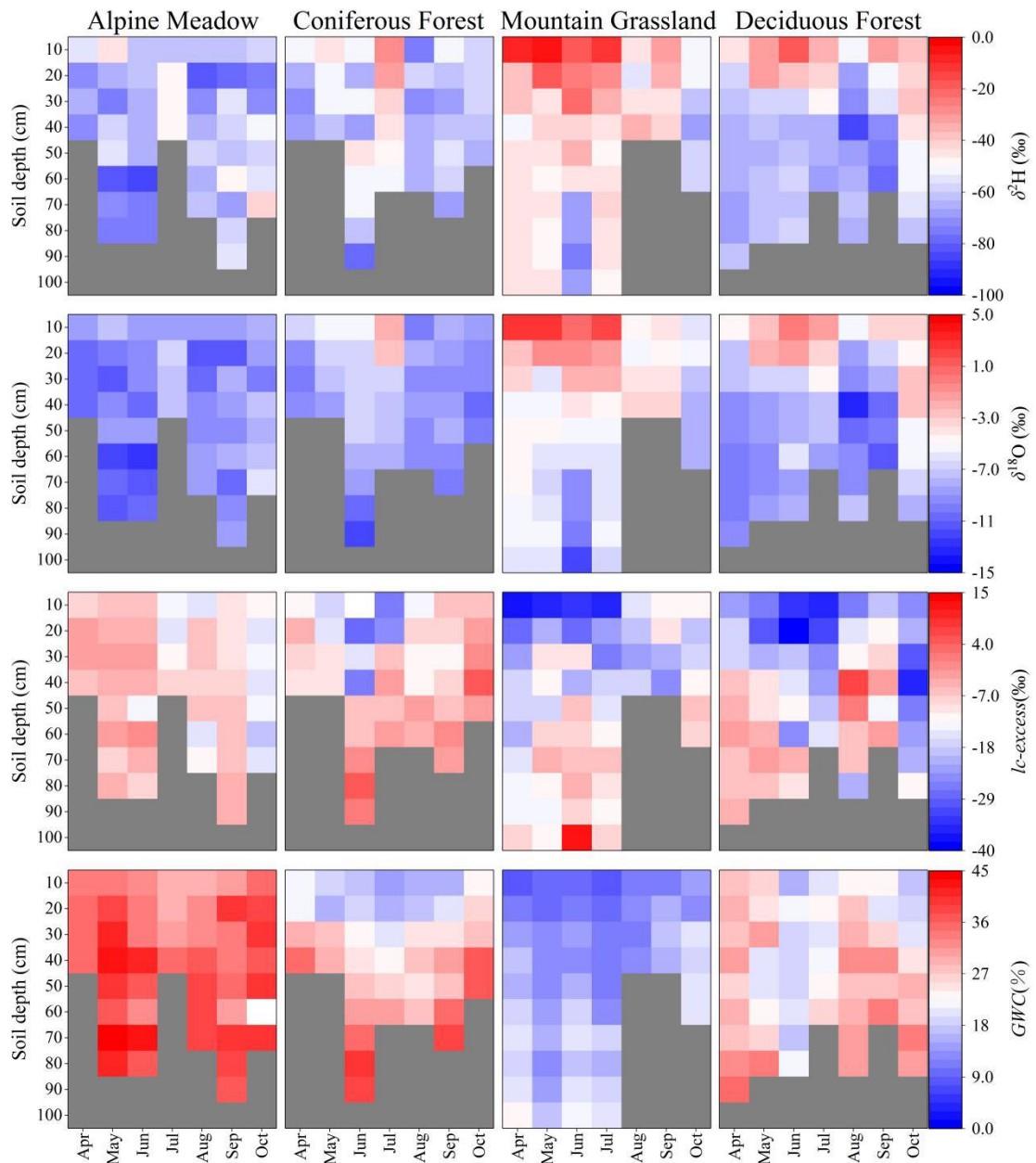

**Fig. 4** Heat map of the soil depth profile of δ²H, δ¹⁸O, lc-excess and GWC in different vegetation zones, and the layer lacking measurement is indicated by the grey color

## 4.3 Spatial variation in water stable isotopes in different vegetation zones

Isotope data of precipitation and soil water obtained from different vegetation zones are shown in dual-isotope space in Fig. 5. At the alpine meadow observation station, the slope (8.4) and intercept (23) of the LMWL were higher than those of the GMWL. The slope of the LMWL in the other three vegetation zones was lower than that of the GMWL and gradually decreased with decreasing altitude. With the

decrease in altitude, the slope of the SWL in all vegetation zones except for the deciduous forest SWL decreased (AM: 6.4; CF: 4.7; MG: 3.4; DF: 4.1), indicating that the evaporation of soil moisture increased. On the one hand, the vegetation coverage of the deciduous forest site was higher. On the other hand, the Xiying Reservoir enhanced the regional air humidity and decreased the local water vapor circulation driving force.

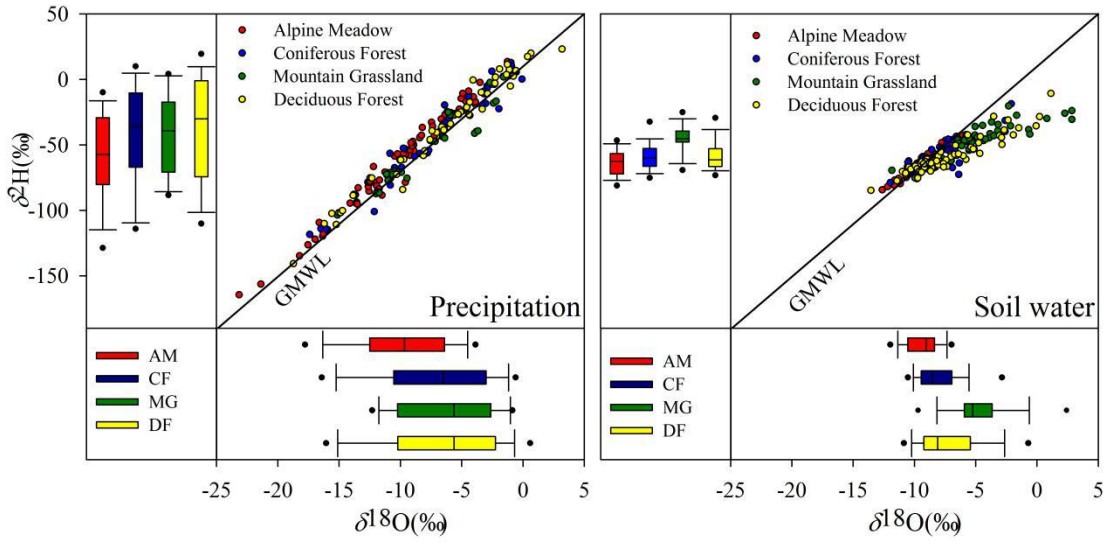

**Fig. 5** Dual-isotope space of precipitation (left) and soil water (right) isotope data of
four vegetation zones. In the box plots, the box represents the 25%-75% percentile,
the line in the box represents the median (50th percentile), the required line
indicates 90th and 10th percentile, and the point indicates the 95th and 5th
percentile.

During the study period, compared with that in other vegetation belts, the surface isotopic value of the soil water in the mountain grassland was relatively enriched (-24.3‰, $\delta^2H$; -0.8‰, $\delta^{18}O$), the lc-excess was smaller and deeper into the middle and lower soil layers (-25.8‰), and the gravimetric water content was relatively low (8.4%). Due to the difference in vegetation types and the influence of reservoirs, this change did not have an obvious elevation effect. Although the elevation was low, the soil water of the deciduous forest had more depleted isotopic characteristics and higher soil moisture than those of the mountain grassland in most samples. Soil profiles obtained from different vegetation zones can reflect the evaporation signals of

water. The low-temperature natural environment made alpine meadow soil less affected by evaporation (lc-excess > -20‰), and the gravimetric water content was high (gravimetric water content > 20%) during the whole study period. The surface soil water of the coniferous forest was easily affected by climate and had a higher isotopic composition (-29.5‰, $\delta^2H$; -2.1‰, $\delta^{18}O$) and lower lc-excess (-26.3‰). Due to evaporation, soil water isotopes in the mountain grassland and deciduous forest areas were enriched in the surface soil layer. In particular, in the mountain grassland, the average values of $\delta^2H$ and $\delta^{18}O$ in the 0-10 cm soil layer were as high as -24.4‰ and -1.2‰, respectively, and SWlc-excess was lower than -25‰, even close to -40‰ in some samples. In this case, the evaporation signals can easily penetrate the deep soil, making the gravimetric water content values at all the sampling sites lower than 20% (Fig. 4; Fig. 6).

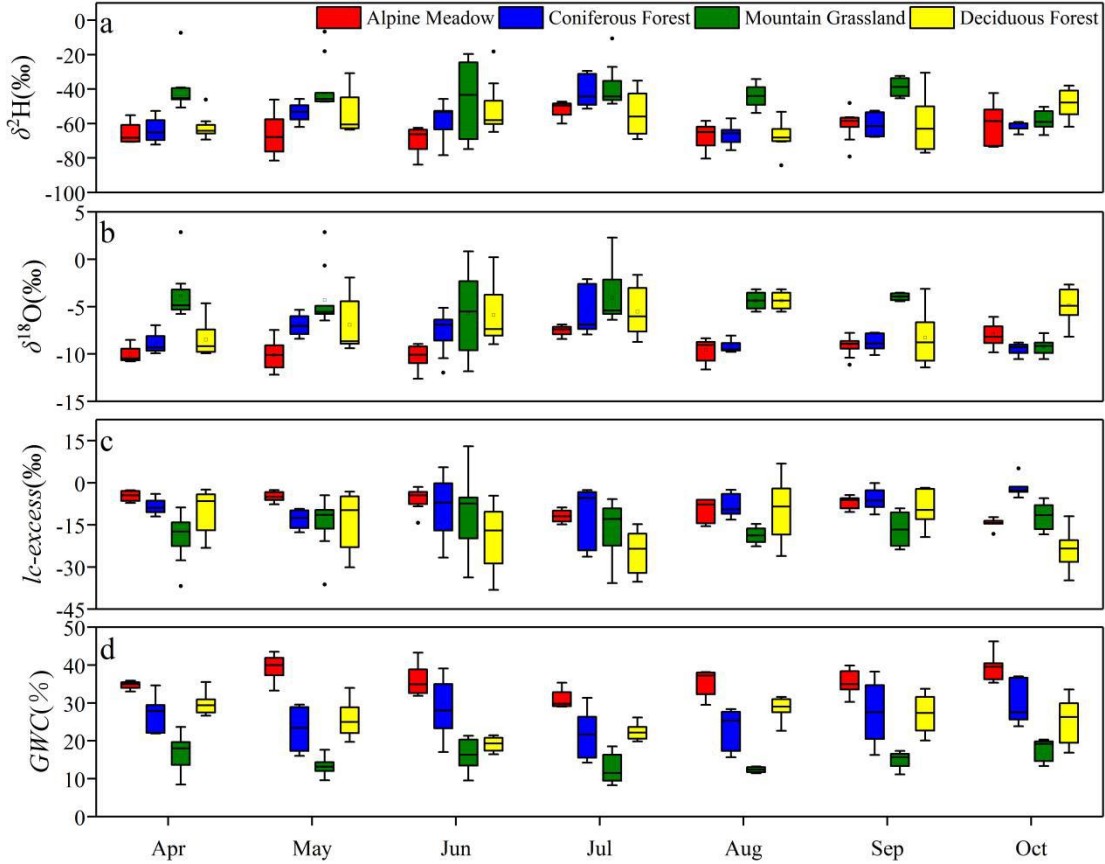

**Fig. 6** The variation of $\delta^2H$, $\delta^{18}O$, lc-excess and GWC in different vegetation zones in each sampling

## 4.4 Variations in the water storage capacity of the 0-40 cm soil layer in different vegetation areas

This study used soil water to calculate the water storage of the 0-40 cm soil layer in the four vegetation zones during the observation period (Fig 7). The water storage capacity of the alpine meadow gradually decreased from April to July (209.7~167.2 mm), and the water storage capacity increased after July (167.2~201.8 mm). The monthly average water storage capacity was the lowest at 0-10 cm (43.0 mm) and the highest at 30-40 cm (51.7 mm). The water storage capacity of the coniferous forest gradually decreased from April to July (150.1~101.2 mm), and the water storage capacity increased after July (101.2~160.0 mm). The monthly average water storage capacity was the lowest at 0-10 cm (28.0 mm) and the highest at 30-40 cm (40.0 mm). The water storage capacity of the mountain grassland gradually decreased from April to July (80.3~64.0 mm), and the water storage capacity increased after July (64.0~104.6 mm). The monthly average water storage capacity was the lowest at 0-10 cm (17.5 mm) and the highest at 20-30 cm (22.0 mm). The water storage capacity of the deciduous forest gradually decreased from April to June (159.3~104.0 mm), the water storage capacity increased from June to August (104.0~154.0 mm), and there was a decrease from August to October (154.0~111.8 mm). The monthly average water storage capacity was the lowest at 0-10 cm (29.1 mm) and the highest at 20-30 cm (35.0 mm). In general, the soil water storage capacity of the 0-10 cm soil layer was less than that of the other soil layers. The order of the water storage capacity of the 0-40 cm soil layer in the four vegetation zones is alpine meadow (46.9 mm) > deciduous forest (33.0 mm) > coniferous forest (32.1 mm) > mountain grassland (20.3 mm).

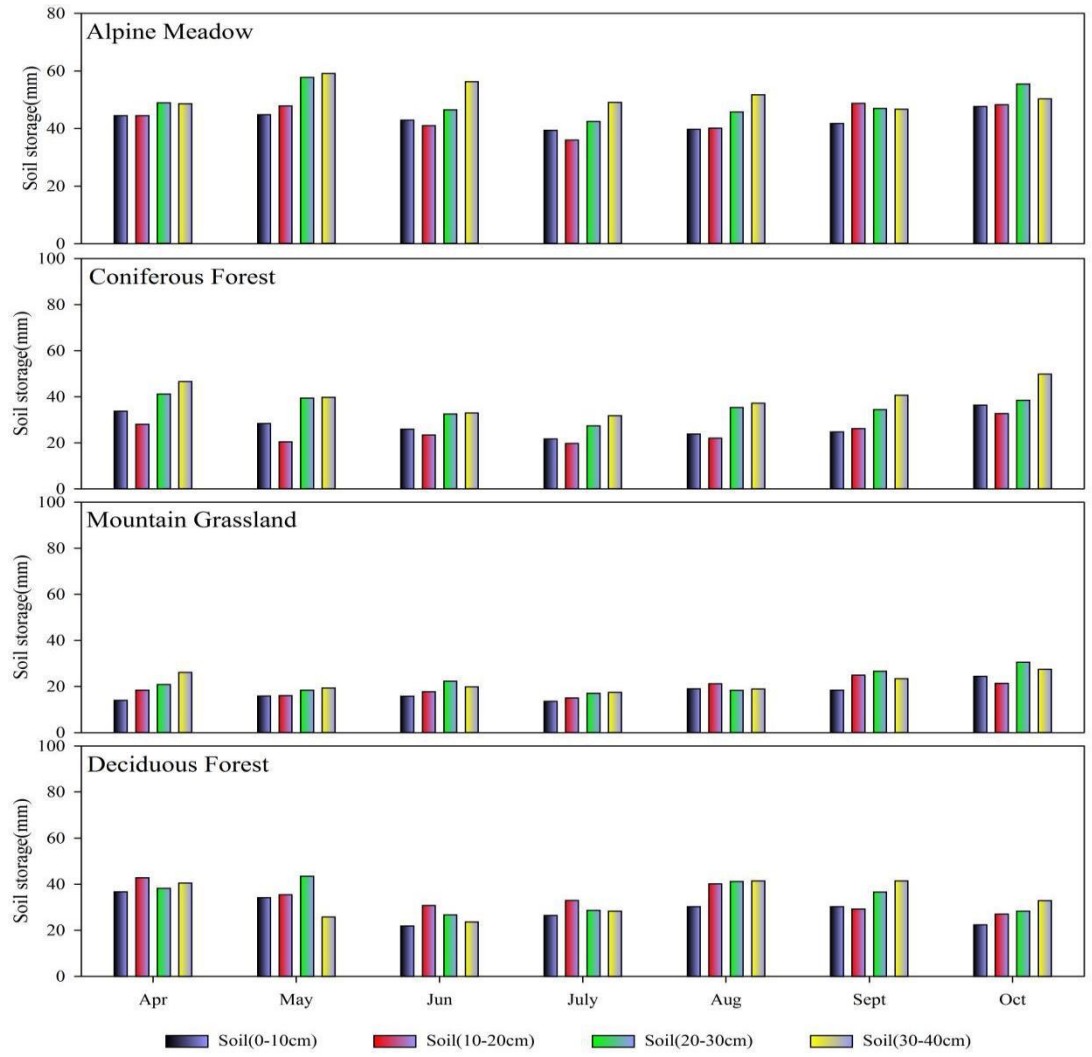

**Fig. 7** Monthly variation of soil water storage in 0-40cm soil layer of different

vegetation zones

## 5. Discussion

**5.1 Evaporation of soil moisture in different vegetation zones**

In the arid river source area, the replenishment of soil moisture mainly comes from precipitation. The slope of the regional atmospheric precipitation line can reflect the strength of local evaporation. Due to a low atmospheric temperature, low cloud base height, and low air-saturated water vapor loss, the alpine meadow zone was weakly affected by secondary evaporation during precipitation. There, the slope of the LMWL (8.4) was even higher than that of the GMWL (Hughes and Crawford, 2012). As the altitude decreased, the secondary evaporation under clouds strengthened, and

the slope of the LMWL of each vegetation zone decreased (Pang et al., 2011). The
slope of SWL can indicate the strength of soil moisture evaporation in each vegetation
zone, the evaporation intensity results of the four vegetation zones followed the order
of mountain grassland (SWL$_{slop}$: 3.4) > deciduous forest (SWL$_{slop}$: 4.1) > coniferous
forest (SWL$_{slop}$: 4.7) > alpine meadow (SWL$_{slop}$: 6.4) (Fig 5). The dynamic changes in
lc-excess of soil profiles in different vegetation areas reflect the process of soil water
evaporation caused by drought during the study period. The monthly average value of
SWlc-excess in the alpine meadow zone was less than 0, and the minimum value was
-11.9‰ (July). Although the vegetation belt was subject to different degrees of
evaporation each month, it was less affected by drought, and it was difficult for
evaporation to penetrate into the middle and lower soil layers. The SWlc-excess of the
coniferous forest belt was greater than that of the alpine meadow from April to June.
The evaporation was the strongest in July (-11.2‰ lc-excess). Similar to in the alpine
meadow, in the coniferous forest belt, evaporation mainly occurred in the topsoil. The
vegetation coverage of the mountain grassland zone was low, and the arid
environment made the isotopes of the surface soil produce strong evaporation signals
(lc-excess was close to -40‰). In most samples, the SWlc-excess of the 60-80 cm soil
layer was negative. The evaporation signal shifted to the lower layer of the soil
(Barnes and Allison, 1988; Zimmermann et al., 1966). Similar evaporation signals
have been found in the Mediterranean and arid climate regions (McCutcheon et al.,
2017; Sprenger et al., 2016). Evaporation signals exist in only the surface soil in
humid areas, and there is no difference between lc-excess and 0 in the soil layer below
20 cm (Sprenger et al., 2017). The monthly surface soil evaporation of deciduous
forest was less than that of mountain grassland from April to June, and it was greater
than that of mountain grassland after July, mainly due to the influence of the
vegetation and reservoirs. There were commonalities in the soil moisture changes in
different vegetation zones characterized by more enriched isotopes, stronger
evaporation signals, and lower moisture content in the shallow soil. With increasing
soil depth, the isotope gradually became depleted, and the evaporation signal was
gradually weakened until it disappeared. The evolution of the investigated isotopes,
lc-excess, and gravimetric water content in the unsaturated soil showed differences
among different vegetation zones. From a high altitude to a low altitude, the isotopic
value of the surface gradually increased, and the evaporation signal increased (Fig 4;
Fig 6).
**5.2 Memory effects of precipitation input, mixing and rewetting**
The changes in soil water isotopes and soil moisture can evaluate the input,
mixing, and rewetting precipitation process in different vegetation areas. The main
methods of precipitation input are plug flow and preferential flow. Plug flow is the
complete mixing of water through the soil matrix with shallow free water. Under the
action of plug flow, precipitation infiltrates along the hydraulic gradient, pushing the
original soil water downward. Preferential flow means that precipitation uses soil
macropores to quickly penetrate shallow soil to form deep leakage (Tang and Feng,
2001). After precipitation, the variability of isotope signals at a certain soil depth can
identify the seepage method of water (Peralta-Tapia et al., 2015). During the study
period, the soils of the alpine meadow and coniferous forest areas were seasonally
frozen and thawed year-round, and the difference in the soil isotope profile was small.
The soil moisture profile showed a trend of water increasing from top to bottom,
indicating the influence of the previous precipitation. The soil was humid, so the
replenishment of soil water by precipitation had the characteristics of top-down piston
replenishment. Preferential infiltration showed high variability in isotopic signals
(Brodersen et al., 2000), and the rainwater in mountain grassland and deciduous forest
flowed into the deep soil rapidly through the soil matrix via exposed soil fissures and
roots. This resulted in the sudden depletion of soil isotopes at a depth of 60-100 cm.
This may be due to the more recent depleted precipitation that quickly reached this
depth and the preferential infiltration into the soil. Water movement and mixing in the
unsaturated zone can be observed in the spatiotemporal variation in isotopes within 1
m of the soil profile, and the alpine meadow and coniferous forest zones underwent
considerable rainfall. After a short period of weak evaporation, the soil was rewetted
by the next rainfall. In the alpine meadow, the soil moisture remained above 20% each
month. The mountain grassland and deciduous forest zones had only sporadic
precipitation from mid-May to late July, and the soil moisture evaporated rapidly.
With the decrease in air temperature and the occurrence of continuous precipitation
after July, the soil was rewetted after two months of drought, and both vegetation
zones showed the replacement and mixing of soil water isotopes and precipitation.
The results showed that the soil water storage capacity of the alpine grassland was
seriously insufficient, reflecting the incomplete rewetting of the soil by precipitation
at the end of the study. In addition, low soil water storage capacity will enrich the
remaining soil water isotopes (Barnes and Allison, 1988; Zimmermann et al., 1966).
We observed the memory effect of soil rewetting caused by precipitation input and the
mixing of different vegetation areas during the entire study period. The changes in
soil moisture in each vegetation area reflect different climatic and hydrological
characteristics (Fig. 4; Fig. 6).
**5.3 Influencing factors of soil water storage capacity in arid headwater areas**

As the temperature decreased rapidly with increasing height, precipitation and

humidity increased to a certain extent, and the vegetation showed a strip-like
alternation approximately parallel to the contour line, forming zonal vegetation with
obvious differentiation (Yin et al., 2020). The dry-wet conditions of different
vegetation zones restricted the soil water storage capacity in the basin. In the process
of low-altitude vegetation zone replacement, the precipitation decreased, the
temperature rose, the groundwater level dropped, and the soil water storage capacity
was weak (Coussement et al., 2018; Kleine et al., 2020). The soil water storage
capacity of the alpine meadow zone with low-temperature and rainy weather was
higher than that of other vegetation zones (results of the 0-40 cm soil layers from
April to October: AM: 187.8 mm; CF: 128.4 mm; MG: 81.2 mm; DF: 132.1 mm).
During the study period, the soil water storage capacity (0-40 cm) exceeded 165 mm
each month. With the decrease in altitude, the monthly difference in dry-wet
conditions in each vegetation zone gradually became obvious. With the increase in
temperature in summer, the environment became dry, and the soil water storage
capacity weakened (Sprenger et al., 2017). The soil water storage capacity of the
coniferous forest zone began to decrease in April, and the water storage capacity of
the 0-40 cm layer reached the minimum value (101.2 mm) in July. The variation in
temperature and precipitation was the main reason for the monthly difference
(Dubber and Werner, 2019). Although there was a certain water storage capacity in
the coniferous forest with some transpiration loss, the soil water storage capacity in
this vegetation zone was not strong. The water storage capacity of mountain grassland
soil was lower than that of other vegetation zones. The continuous dry and warm
weather in spring and summer led to the water storage capacity of 0-40 cm soil being
lower than that of 100 mm every month. In particular, drought stress leads to
insufficient soil moisture, making it difficult to maintain plant demand, resulting in
sparse vegetation and large-scale exposed surface soil, which further accelerates
surface water loss. The continuous precipitation from the end of July prevented
further drought development, and the water input gradually restored the soil water
storage capacity (Kleine et al., 2020). The deciduous forest had hydrothermal
conditions similar to those of the mountain grassland, but the soil porosity of the
forest zone was obviously larger than that of the barren land, and its permeability was
higher than that of the barren land. Precipitation infiltrated the ground through roots
and turned into groundwater. The forest acted as a reservoir due to its strong water
storage and soil conservation capacity (Sprenger et al., 2019). The water storage
capacity of the 0-40 cm soil layer in the deciduous forest was higher than 100 mm at
each sampling time. In addition, the water content of the 0-40 cm soil layer in each
vegetation zone increased with the deepening of the soil layer, and the water storage
capacity of the surface soil was weak. The difference in soil properties also led to
more water storage in the middle and lower soil layers with higher clay contents
(Milly, 1994) (Fig. 7). Climate warming and the spatiotemporal imbalance of water
resources have disturbed the ecological-water balance of different vegetation zones in
inland river source areas (Liu et al., 2015). Plant growth mainly depends on the water
stored in shallow soil layers (Amin et al., 2020). Drought reduces soil water storage
and inhibits plant growth (Li et al., 2020). To effectively improve and manage water
resources in arid water source areas, exploring the heterogeneity of hydrological
processes among different vegetation zones is necessary. This will provide a reference
for the formulation of ecological policies.

## 6. Conclusion

This work provides further insights into the movement and mixing of soil water in
different vegetation zones in arid source regions. During the study period, the
dynamic changes in lc-excess in the soil profiles of different vegetation zones
reflected the evaporation signals caused by drought. Soil water evaporation in spring
and summer and insufficient precipitation during the drought period were the main
driving forces of isotopic enrichment in the surface soil. The soil water evaporation
intensity results of the four vegetation zones followed the order of mountain grassland
(SWL$_{slop}$: 3.4) > deciduous forest (SWL$_{slop}$: 4.1) > coniferous forest (SWL$_{slop}$: 4.7) >
alpine meadow (SWL$_{slop}$: 6.4). In the mountain grassland and deciduous forest zones,
drought caused the evaporation signal to penetrate deep into the middle and lower soil
layers. The SWlc-excess below 70 cm of the ground surface remained negative. Soil
water isotopes and gravimetric water content record the process of soil rewetting
caused by precipitation input and mixing. The alpine meadow and coniferous forest
zones were enriched in precipitation. After a short period of weak evaporation, the
soil was rewetted by the next precipitation event. There was only sporadic
precipitation in the mountain grassland and deciduous forest belt from mid-May to
late July. After July, the temperature dropped, and continuous precipitation wet the
soil again after two months of drought. The mountain grassland and deciduous forest
zones had only sporadic precipitation from mid-May to late July. With the decrease in
air temperature and continuous precipitation after July, the soil was rewetted after two
months of drought. Moisture and temperature conditions were the key factors that
restricted the soil water storage capacity in the different vegetation zones. The water
storage capacity of the 0-40 cm soil layer results followed the order of alpine meadow
(46.9 mm) > deciduous forest (33.0 mm) > coniferous forest (32.1 mm) > mountain
grassland (20.3 mm). The water storage capacity of the surface soil in each vegetation
zone was weak, and more water was stored in the middle and lower soil layers with

higher clay contents. The research results can be applied to arid and semi-arid alpine

regions and have reference significance for latitude and longitude differentiation. This

study mainly emphasized the Spatio-temporal heterogeneity of soil water evaporation,

infiltration, and water storage in different vegetation zones. These results are of great

value for understanding regional hydrological processes and ecological restoration

services in environmentally fragile areas. Furthermore, we hope this study can be used

as a basic statement because we continue to use stable water isotopes as a data source

to understand hydrological processes from the perspective of process mechanisms.

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

**Acknowledgments**

This research was financially supported by the National Natural Science Foundation of China (41661005, 41867030, 41971036). The authors much thank the colleagues in the Northwest Normal University for their help in fieldwork, laboratory analysis, data processing.

**Author Contribution statement**

Guofeng Zhu and Leilei Yong conceived the idea of the study; Yuanxiao Xu and Qiaozhuo Wan analyzed the data; Zhigang Sun and Leilei Yong were responsible for field sampling; Zhuanxia Zhang participated in the experiment; Lei Wang participated in the drawing; Leilei Yong wrote the paper; Liyuan Sang, Xi Zhao and Yuwei Liu checked and edited language. All authors discussed the results and revised the manuscript.

**Additional Information**

Competing Interests: The authors declare no competing interests.