# Peer review of "Evaporation, infiltration and storage of soil water in"

_Hydrology and Earth System Sciences, 2021_

## Author Comment (AC1)

**Revision Notes**

Dear Editor and Reviewers:

Thank you for your letter and for the reviewers' comments concerning our manuscript entitled "Evaporation, infiltration and storage of soil water in different vegetation zones in Qilian mountains: From a perspective of stable isotopes" (Manuscript Number: Hess-2021-376).

According to the reviewers' comments, we have revised our manuscript carefully. The revised portions have been marked in red in the revised version of the manuscript. The main corrections and the response to the reviewers' comments are as follows.

**Responses to the reviewer's comments:**

**Response to Reviewer #1**

**Reviewer#1: This is an interesting study conducted in a mountain region in China. The research is based on the use of stable isotope in water to understand the main mixing processes and hydrological processes. The manuscript is logically organized and clearly illustrated. However, there are serious issues that in my opinion should prevent the publication of this manuscript in the present form. First of all, the English language is very poor, and sometimes (especially in the Introduction) it's hard to follow the reading. Secondly, the are serious flaws in the Abstract and mostly in the Introduction that fails to reach the focal point of this work (see specific comments below). Third, the discussion is well complemented with literature references but is quite often vague and appears to be not well supported by the observations. No reference to figures and tables are reported in the discussion and it seems that the processes explained by the Authors are based on a previous knowledge of the area and by results taken from the literature than supported by their own results. I suggest to more strictly base the inference on hydrological processes on the observed results.**

**In the end I suggest to resubmit this manuscript after fixing all these major issues.**

Thanks for your comments.

**Specific comments**

**Comment 1:** The title does not read well. I suggest changing into "Evaporation, infiltration and soil storage in different mountain zones" or "Evaporation, infiltration and soil storage in different vegetation zones in a mountain catchment". Perhaps also "Evaporation, infiltration

and soil storage in different mountain zones: an isotope perspective". But there is no strict need to stress the isotopic perspective, in my opinion.

**Response 1:** Based on the suggestions of the two reviewers, I have revised the title of the manuscript.

L1-3: Evaporation, infiltration and storage of soil water in different vegetation zones in the Qilian mountains: a stable isotope perspective

**Comment 2:** The abstract lacks to report the main objective, or the research questions.

**Response 2:** I added the main research objectives and questions in the abstract according to your suggestion.

L13-17: In order to further understand the process of soil water movement and runoff generation in different vegetation zones (Alpine Meadow, Coniferous Forest, Mountain Grassland and Deciduous Forest) in mountain areas, this study monitored the temporal and spatial dynamics of hydrogen and oxygen stable isotopes in the precipitation and soil water of the Xiying River.

**Comment 3:** The introduction suffers from different weak points and needs a severe restructuring.

1) It is not clear what different vegetation zones are, and what role they play in water exchange in the environment, and why they are important in this research.

2) The text focuses too much on the variability of the isotopic composition in vegetated environments without going deeper into the main physical processes that still need to be understood. Isotopes are just a tool, and the goal here is to understand hydrological processes with the help of isotopes, not which are the factors affecting isotopic variability.

3) Very importantly, no research gaps is put forward. We understand that studying and understanding hydrological processes in different vegetated areas is important but what is the real problem here? As a result, the specific objectives are disconnected from the rest of the Introduction and fluctuate in their own space. Moreover, what is the memory effect? Why is it important? What is not known about it? How does it fit the general story behind this paper?

**Response 3:** I rewrote the introduction based on your suggestions and other reviewers' comments. The memory effect refers to the indicative profile produced by the stable isotope of soil water in response to the event after precipitation or drought. We highlighted its importance in the manuscript. The memory effect refers to the indicative profile produced by

the response of stable isotope of soil water to precipitation or drought. We highlighted its importance in the manuscript.

L31-98: In arid inland river basins, changes in climate and vegetation will affect the hydrological cycle process, which is essential for assessing regional water balance and future changes in water resources (Wang et al., 2012; Tetzlaff et al., 2013; Ning et al., 2020; Sharma et al., 2021). To cope with the changing natural environment, managers have formulated a series of scientific ecological management policies based on species selection (Wookey et al., 2010), crop rotation (Zhu et al., 2019), and ecological water conveyance (Zhang et al., 2019), which has been improving their adaptability to the changing natural environment.

Soil water in the unsaturated zone can be converted from precipitation to steam or groundwater recharge. Evaporation, infiltration and water storage are critical for understanding the regional hydrological process and water balance under the background of climate and vegetation changes (Brooks et al., 2010; Grant and Dietrich, 2017). Isotopes, as "fingerprints" of water, have been used to track eco-hydrological processes, such as evaporation (Barnes and Allison, 1988; Zhu et al., 2021b), groundwater recharge (Koeniger et al., 2016), infiltration path (Tang and Feng, 2004; Duvert et al., 2016; Zhu et al., 2021a), evapotranspiration distribution (Xiao et al., 2018; Gibson et al., 2021) and water absorption by plants (Rothfuss and Javaux, 2017).

The water seepage in the unsaturated soil zone and the evaporation of water at the air-soil interface are the main forms of soil water transport. Seasonal variations of precipitation isotopes are often used to track the process of soil water leakage (Stumpp et al., 2012). During the piston infiltration process, the precipitation with different $\delta^2H$ and $\delta^{18}O$ peaks retained in the soil profile and gradually disappears as the infiltration depth increases (Sprenger et al., 2016a), while the preferential flow will keep these peaks until the deep soil layer (Peralta-Tapia et al., 2015). During a precipitation event, the response of water isotope in surface soil to precipitation is more obvious, showing a changing trend similar to that of stable isotope of precipitation. With the deepening of the soil layer, the seasonal variation of precipitation isotope signals is rapidly attenuated, and the influence of precipitation on soil water gradually weakens (Sprenger et al., 2017). Evapotranspiration is the main form of soil water dissipation. Because the mass of hydrogen and oxygen atoms that make up water molecules are related to their thermodynamic properties, isotope fractionation of water will occur in the process of the water cycle. The evaporation of liquid water produces water vapor enriched in $^1H$ and $^{16}O$, while the remaining water is enriched in $^2H$ and $^{18}O$ (Ferretti et al.,

2003). Dansgaard (1964) proposed the concept of d-excess (d-excess=$\delta^2$H-8$\delta^{18}$O) to illustrate the intensity of evaporative fractionation. In the state of isotopic equilibrium, the value of the d-excess is 10. Compared with d-excess, lc-excess can explain the evaporative fractionation process better. The main reason is that lc-excess in precipitation and soil water changes smoothly and has relatively small seasonal changes (Landwehr et al., 2014). The dynamic changes of isotopes record the signal of soil water evaporation. This enrichment from dynamic fractionation exists in soil water isotopes in different climatic regions. Compared with temperate regions, the signals of evaporation in arid and Mediterranean environments penetrate deeper into the soil ( Sprenger et al., 2016b). Some water will be stored in the soil after evaporation and seepage processes. The water storage capacity of humid areas is higher than that of arid areas, the water storage capacity of forests is higher than that of grassland, and the water storage capacity of middle and lower soil layers with higher clay content is higher than that of surface soil layer ( Heinrich et al., 2019; Sprenger et al., 2019; Kleine et al., 2020; Snelgrove et al., 2021).

In alpine mountains, climate warming will accelerate the melting of glaciers and frozen soil, and the dynamic interaction between water bodies stored in different media will become the main focus of the water cycle (Penna et al., 2018). Previous studies on evaporation, infiltration and storage of soil water mostly focused on different climatic regions or vegetation types under the same climatic region. Understanding the climatic and hydrological conditions of different vertical vegetation zones and clarifying the regulating role of vegetation in the water cycle process can help to better adapt to the influence of climate change on the hydrological process in the source area. In this study, the stable isotope composition of precipitation and soil water, and soil water storage's spatiotemporal dynamics were monitored in four vegetation zones (Alpine Meadow, Coniferous Forest, Mountain Grassland and Deciduous Forest) with different hydrothermal conditions in the Xiying River Basin. In order to explore the differences in soil water evaporation, infiltration and storage processes in these four different climates, vegetation and terrain regions, the following research objectives are proposed: (1) Exploring the evolution of isotope evaporation signals and the "memory effects" of precipitation input, mixing and rewetting; (2) Understand the soil water storage capacity and influencing factors of four vegetation areas in the mountain areas.

**Comment 4:** L349-354. This paragraph and the related Fig. 7 fit much better in the Results than in the discussion. I suggest restructuring this part.

**Response 4:** According to your suggestion, I have adjusted this part of the content to the Results.

L343-365: **4.4 Variation of water storage capacity of 0-40cm soil layer in different vegetation areas**

This study used soil water to calculate the water storage of the 0-40cm soil layer in the four vegetation zones during the observation period (Fig 7). The water storage capacity of the Alpine Meadow gradually decreases from April to July (209.7 mm~167.2 mm), and the water storage capacity increases after July (167.2 mm~201.8 mm). The monthly average water storage capacity is the least for 0-10 cm (43.0 mm) and the most for 30-40 cm (51.7 mm). The water storage capacity of the Coniferous Forest gradually decreases from April to July (150.1 mm~101.2 mm), and the water storage capacity increases after July (101.2 mm~160.0 mm). The monthly average water storage capacity is the least for 0-10 cm (28.0 mm) and the most for 30-40 cm (40.0 mm). The water storage capacity of the Mountain Grassland gradually decreases from April to July (80.3 mm~64.0 mm), and the water storage capacity increases after July (64.0 mm~104.6 mm). The monthly average water storage capacity is the least for 0-10 cm (17.5 mm) and the most for 20-30 cm (22.0 mm). The water storage capacity of the Deciduous Forest gradually decreases from April to June (159.3 mm~104.0 mm), the water storage capacity increases from June to August (104.0 mm~154.0 mm), and there is a decrease from August to October (154.0 mm~111.8 mm).The monthly average water storage capacity is the least for 0-10 cm (29.1 mm) and the most for 20-30 cm (35.0 mm). In general, the soil water storage capacity of the 0-10 cm soil layer is less than that of other soil layers. The order of the water storage capacity of the 0-40 cm soil layer in the four vegetation zones is AM > DF > CF > MG.

**Comment 5:** L365-366. Which are the results that lead the Authors to believe this? Please, explain.

**Response 5:** I reinterpreted this point of view.

L417-421: The soil moisture profile showed a trend of water increase from top to bottom, indicating that this period was caused by the influence of the previous precipitation. The soil is humid, so the replenishment of soil water by precipitation has the characteristics of top-down piston replenishment.

**Comment 6:** L422-424. Again, here we need some experimental evidence about this process. Section 5.3. All this is interesting but it sounds a bit general and vague, these statements look not so related to the results, there are no references to the figures, and the reader has the impression that the Authors present their own preliminary idea that does not reflect the data. I'm happy to be mistaken here but we need to have evidence of all the described processes. Moreover, the title does not reflect the content of the section.

**Response 6:** We logically sorted out the full text based on the reviewers' comments. The discussion on runoff generation in the watershed does not match the theme of this manuscript (Evaporation, infiltration and storage of soil water in different vegetation zones in the Qilian mountains: a stable isotope perspective). Therefore, the manuscript focused on soil moisture's evaporation, infiltration, and storage mechanism in the study area. Based on this, we reorganized this part of the content.

L367-492: **5. Discussion**

**5.1 Evaporation of soil moisture in different vegetation zones**

In the arid river source area, the replenishment of soil moisture mainly comes from precipitation. The slope of the regional atmospheric precipitation line can reflect the strength of local evaporation. In Alpine Meadow, due to low atmospheric temperature, low cloud base height, and low air-saturated water vapor loss, it is weakly affected by secondary evaporation during precipitation. The slope of LMWL (8.4) is even higher than that of GMWL (Zhang et al., 2012). As the altitude decreases, the secondary evaporation under the cloud strengthens, and the slope of the LMWL of each vegetation zone decreases (Pang et al., 2011) (Fig 5). The dynamic changes of lc-excess of soil profiles in different vegetation areas reflect the process of soil water evaporation caused by drought during the study period. The monthly average value of SWlc-excess in Alpine Meadow was less than 0, and the minimum value was -11.9‰ (July). Although the vegetation belt is subject to different degrees of evaporation each month, it is less affected by drought and it is difficult for evaporation to penetrate into the middle and lower soils. The SWlc-excess of the Coniferous Forest belt is greater than that of the alpine meadow from April to June. The evaporation is the strongest in July (-11.2‰, lc-excess). Similar to alpine meadows, evaporation mainly occurs in the top soil. The vegetation coverage of Mountain Grassland is low, and the arid environment makes the isotopes of the surface soil produce strong evaporation signals (lc-excess is close to -40‰). In most samples, the SWlc-excess of the 60-80 cm soil layer is negative. The evaporation signal moves to the lower layer of the soil. (Zimmermann et al., 1966; Barnes and Allison, 1988). Similar

evaporation signals have been found in the Mediterranean and arid climate regions (Sprenger et al., 2016b; McCutcheon et al., 2017). Evaporation signal only exists in the surface soil in humid areas, and there is no difference between lc-excess and 0 in the soil layer below 20cm (Sprenger et al., 2017). The monthly surface soil evaporation of Deciduous Forest is less than that of Mountain Grassland from April to June, and it is greater than mountain grassland after July, which is mainly due to the influence of vegetation and reservoirs. 
[revised manuscript text omitted]

**Minor comments and technical corrections**

**Comment 7:** L58-59. D-excess is introduced here without any explanation on its formulation and physical meaning.

**Response 7:** "d-excess" has been introduced in detail.

L65-67: Dansgaard (1964) proposed the concept of d-excess (d-excess=$\delta^2$H-8$\delta^{18}$O) to illustrate the intensity of evaporative fractionation. In the state of isotopic equilibrium, the value of the d-excess is 10.

**Comment 8:** L96. Is this the long-term average runoff? Please, specify.

**Response 8:** This is the average runoff over the years, which has been clarified.

L104-106: The average annual runoff of the Xiying River is 388 million cubic meters, which is mainly replenished by mountain precipitation and melting water of ice and snow.

**Comment 9:** Table 1. Are the meteorological parameters averages? Over which period of time? Please, specify.

**Response 9:** Based on your suggestion, I have changed and explained them here.

L143-145: **Table 1** Basic data of each Vegetation zone from April to October 2017 (*Long*-Longitude, *Lat*-Latitude, *Alt*-Altitude, *T*-Air Temperature (daily mean temperature), *P*-Precipitation (total precipitation during the observation period), *h*-Relative Humidity (daily mean relative humidity))

**Comment 10:** Section 3 (Methods). No explanations about the determination of the gravimetric water content are given. Please fix this issue.

**Response 10:** Based on your suggestion, I have added the determination of the gravimetric water content.

L191-194: *W* is gravimetric water content, which is expressed by formula as follows:

$$W = \frac{M_1 - M_2}{M_2} \times 100\% \tag{5}$$

In the formula: $M_1$ is the gravimetric of the wet soil (g), and $M_2$ is gravimetric of the dry soil (g).

**Comment 11:** L140-142. I suggest including a reference to the correction of the memory effect (e.g., Penna et al., 2012 and/or Qu et al., 2019).

Penna, D., Stenni, B., Šanda, M., Wrede, S., Bogaard, T. A., Michelini, M., Fischer, B. M. C., Gobbi, A., Mantese, N., Zuecco, G., Borga, M., Bonazza, M., Sobotková, M., Ä©ejková, B., and Wassenaar, L. I.: Technical Note: Evaluation of between-sample memory effects in the analysis of $\delta^2$H and $\delta^{18}$O of water samples measured by laser spectroscopes, Hydrol. Earth Syst. Sci., 16, 3925–3933, https://doi.org/10.5194/hess-16-3925-2012, 2012.

Qu, D, Tian, L, Zhao, H, Yao, P, Xu, B, Cui, J. Demonstration of a memory calibration method in water isotope measurement by laser spectroscopy. Rapid Commun Mass Spectrom. 2020; 34:e8689. https://doi.org/10.1002/rcm.8689

**Response 11:** I have added the necessary references here.

L153-155: In order to avoid the memory effect of isotope analysis, we discarded the first two injection values, used the average value of the last four times as the final value (Penna et al., 2012; Qu et al., 2020).

**Comment 12:** L169. Please explain the role of 10 in the equation.

**Response 12:** Here, we calculate the water storage capacity of the 10cm soil layer, so we multiply it by 10.

**Comment 13:** L176. Add "2017".

**Response 13:** This problem has been corrected.

L198-201: During the study period (April-October 2017), the potential evapotranspiration was 872.8 mm, and the daily evapotranspiration ranged from 7.5 mm (July 14) to 0.9 mm (October 9) in Xiying River Basin, showing a fluctuating trend around July, and the PET value in April-July was higher than that in August-October.

**Comment 14:** L194: Precipitation events?

**Response 14:** This problem has been corrected.

L215-217: During the observation period, there were 72 precipitation events in the Alpine Meadow zone, the total precipitation was 534.3 mm, which was relatively evenly distributed in each month.

**Comment 15:** L213-220. This part can be reported in a Table or in a boxplot.

**Response 15:** Based on your suggestion, we have drawn a table.

L254-256: **Table 2** General characteristics of precipitation $\delta^2H$ and $\delta^{18}O$ in different vegetation areas from April to October 2017

| Vegetation zone | $\delta^2H$/‰ | | | | $\delta^{18}O$/‰ | | | |
|---|---|---|---|---|---|---|---|---|
| | Max | Min | mean | SD | Max | Min | mean | SD |
| AM | 13.7 | -163.9 | -73.1 | 36.3 | -1.3 | -23.1 | -10.0 | 4.3 |
| CF | 13.0 | -117.8 | -42.0 | 37.2 | -0.1 | -17.4 | -7.1 | 4.7 |
| MG | 4.2 | -103.1 | -37.4 | 30.5 | -0.9 | -15.1 | -5.9 | 3.9 |
| DF | 23.2 | -110.2 | -31.8 | 42.8 | 3.2 | -15.2 | -5.8 | 5.5 |

**Comment 16:** Fig. 3. Use different colours to distinguish between the different variables. Particularly, I suggest using different closures for the two isotopes and then keep them in all the other graphs.

**Response 16:** According to your suggestion, I modified Figure 3.

L257-260:

[Figure]

**Fig. 3** Time series of rainfall and isotope characteristics in different vegetation zones in Xiying River Basin, with dotted lines indicating the date of soil water sampling

**Comment 17:** L263? Does "deep" mean "grey"? In that case use the correct term.

**Response 17:** This problem has been corrected.

L300-302:

**Fig. 4** Heat map of the soil depth profile of $\delta^2H$, $\delta^{18}O$, lc-excess and GWC in different vegetation zones, and the layer lacking measurement is indicated by grey color

**Comment 18:** L266. The diagram is normally called "dual isotope space".

**Response 18:** This problem has been corrected.

L304-306: Isotope data of precipitation and soil water obtained from different vegetation zones were shown in the dual-isotope space (Fig. 5).

**Comment 19:** L272-274. This part can/could be moved to the discussion.

**Response 19:** We have moved this part to the discussion according to your suggestion.

L373-378: In Alpine Meadow, due to low atmospheric temperature, low cloud base height, and low air-saturated water vapor loss, it is weakly affected by secondary evaporation during precipitation. The slope of LMWL (8.4) is even higher than that of GMWL (Zhang et al., 2012). As the altitude decreases, the secondary evaporation under the cloud strengthens, and the slope of the LMWL of each vegetation zone decreases (Pang et al., 2011) (Fig 5).

**Comment 20:** L311. As far as I understand the plot does not show the "differences" but the raw values. Please, correct.

**Response 20:** This problem has been corrected.

L341-342:

**Fig. 6** the variation of $\delta^2H$, $\delta^{18}O$, lc-excess and GWC in different vegetation zones in each sampling

**Comment 21:** L385-386. Add a reference to a figure to corroborate the statement.

**Response 21:** This problem has been corrected.

**Comment 22:** L409. I think the correct citation is Amin et al., 2020.

**Response 22:** This problem has been corrected.

**Comment 23:** L463. Typo.

**Response 23:** This problem has been corrected.

---

## Author Comment (AC2)

**Revision Notes**

Dear Editor and Reviewers:

Thank you for your letter and for the reviewers' comments concerning our manuscript entitled "Evaporation, infiltration and storage of soil water in different vegetation zones in Qilian mountains: From a perspective of stable isotopes" (Manuscript Number: Hess-2021-376).

According to the reviewers' comments, we have revised our manuscript carefully. The revised portions have been marked in red in the revised version of the manuscript. The main corrections and the response to the reviewers' comments are as follows.

**Responses to the reviewer's comments:**

**Response to Reviewer #2**

**Reviewer#2: This a potentially interesting paper, but one that needs major attention before it is suitable for publication. The paper is poorly written in places. While I have sympathy with authors having to write in a second language, which is something that I cannot do, some sections of the paper are very difficult to follow. More importantly, the sections of the paper are not well linked. It is not clear from the Introduction how the paper addresses the important issues. The same can be said about the Discussion where it is not clear how the data in this paper inform the issues being discussed. For example, runoff generation is mentioned in the introduction and appears in the general conclusions, but there is no discussion as to how the data in the paper help us understand it (there are several similar examples as well). The sections describing the data tend to be very generalised and the data description needs to be more informative. Moreover, the data need to be presented in the paper or as a supplement.**

**Overall, the paper needs to be rewritten so that the data are discussed in a more rigorous manner that help understand the aims. I am not convinced that it actually addresses important issues or that the aims of understanding the memory effect or runoff generation are advanced by this study.**

Thanks for your comments.

**Specific comments**

**Title**

**Comment 1:** Having a title that is grammatically incorrect is not a good way to promote your research. Something like: "Evaporation, infiltration and storage of soil water in different vegetation zones in the Qilian mountains: a stable isotope perspective" would be better.

**Response:** Based on the suggestions of the two reviewers, I have revised the title of the manuscript.

L1-3: Evaporation, infiltration and storage of soil water in different vegetation zones in the Qilian mountains: a stable isotope perspective

**Abstract**

**The abstract needs improvement. Abstracts are important as they are what convince the readers to look at the rest of the paper. They should convey not only what has been studied and why, but should also contain enough details so that the main conclusions are evident. This abstract needs improving, specifically:**

**Comment 1:** Be specific: "different water bodies in different vegetation zones" does not convey what you have done.

**Response:** I have explained this point in detail.

L13-17: In order to further understand the process of soil water movement and runoff generation in different vegetation zones (Alpine Meadow, Coniferous Forest, Mountain Grassland and Deciduous Forest) in mountain areas, this study monitored the temporal and spatial dynamics of hydrogen and oxygen stable isotopes in the precipitation and soil water of the Xiying River.

**Comment 2:** Avoid qualitative terms such as "weak"

**Response:** The article carried out a more quantitative expression, such as "The evaporation intensity of the four vegetation zones was: Mountain Grassland > Deciduous Forest > Coniferous Forest > Alpine Meadow.".

**Comment 3:** Some of the sentences are unclear. I am not sure what "The water storage capacity of surface soil was weak in vegetation zones" really means as surely all the catchment is vegetated?)

**Response:** In our results, the soil water storage capacity of 0-10 cm is less than that of other soil layers, and we have clarified the soil depth here. Here, we want to show the water storage of different soil layers. In our sampling site, the soil is covered by dominant species.

L23-27: The soil water storage capacity order in each vegetation zone was: Alpine Meadow > Deciduous Forest > Coniferous Forest > Mountain Grassland. In addition, the water storage capacity of 0-10 cm soil was weak, and the water storage capacity of 10-40 cm was strong.

**Comment 4:** There are several grammatical and spelling errors (Nvertheless) that detract from the work

**Response:** We carefully checked and revised the grammar and spelling of the manuscript.

**Introduction**

**The introduction is also not very clear. Some of this reflects the writing style and occasional poor grammar. As well, there needs to be a much clearer explanation of the background. The explanations are vague and would not convey much meaning to anyone not working in the field. There needs to be clearer explanations and more precise terminology.**

**Comment 1:** L31-33. Not very clear what you mean here.

**Response:** For readers to better understand, I re-narrate this sentence.

L31-34: In arid inland river basins, changes in climate and vegetation will affect the hydrological cycle process, which is essential for assessing regional water balance and future changes in water resources (Wang et al., 2012; Tetzlaff et al., 2013; Ning et al., 2020; Sharma et al., 2021).

**Comment 2:** L48. "Storage" is not a transport mechanism.

**Response:**  I agree with your comment, this problem has been corrected.

L49-50: The water seepage in the unsaturated soil zone and the evaporation of water at the air-soil interface are the main forms of soil water transport.

**Comment 3:** L50. Do you mean on the ground surface or in the near-surface part of the soils?

**Response:** We rewrite this sentence.

L63-65: The evaporation of liquid water produces water vapor enriched in $^1$H and $^{16}$O, while the remaining water is enriched in $^2$H and $^{18}$O (Ferretti et al., 2003).

**Comment 4:** L48-75. This would not be readily understandable to many readers who had not worked with these types of data. It is too generally worded and needs details. This paragraph is important as it sets the framework for using the stable isotopes to understand processes.

(1) Define that you are discussing $^{18}O$ and $^{2}H$ data (there are lots of stable isotopes!).

(2) Terms such as "makes soil water isotopes enriched" are vague. Specifically, evaporation enriches the residual water in $^{18}O$ or $^{2}H$ (or increases the $\delta^{18}O$ and $\delta^{2}H$ values)

(3) Likewise, "soil moisture fractionation is positively correlated with evapotranspiration but negatively correlated with precipitation". Are you talking about the magnitude or sign?

(4) How significant?

(5) Define the d-excess (briefly)

(6) L63-70. Lacks detail and is unclear.

**Response:** According to the suggestions of three reviewers, this part was rewritten to solve the above problems: (1) We identified stable isotopes of hydrogen and oxygen; (2) According to your suggestion, the expression has been changed; (3) and (4) According to the reviewer's suggestion, we introduced the evaporation process more and deleted the influencing factors of evaporation; (5) We defined "d-excess"; (6) I gave a detailed description of this part to make it more expressive of the status quo of the research.

L49-98: The water seepage in the unsaturated soil zone and the evaporation of water at the air-soil interface are the main forms of soil water transport. Seasonal variations of precipitation isotopes are often used to track the process of soil water leakage (Stumpp et al., 2012). During the piston infiltration process, the precipitation with different $\delta^{2}H$ and $\delta^{18}O$ peaks retained in the soil profile and gradually disappears as the infiltration depth increases (Sprenger et al., 2016a), while the preferential flow will keep these peaks until the deep soil layer (Peralta-Tapia et al., 2015). During a precipitation event, the response of water isotope in surface soil to precipitation is more obvious, showing a changing trend similar to that of stable isotope of precipitation. With the deepening of the soil layer, the seasonal variation of precipitation isotope signals is rapidly attenuated, and the influence of precipitation on soil water gradually weakens (Sprenger et al., 2017). Evapotranspiration is the main form of soil water dissipation. Because the mass of hydrogen and oxygen atoms that make up water molecules are related to their thermodynamic properties, isotope fractionation of water will occur in the process of the water cycle. The evaporation of liquid water produces water vapor enriched in $^{1}H$ and $^{16}O$, while the remaining water is enriched in $^{2}H$ and $^{18}O$ (Ferretti et al., 2003). Dansgaard (1964) proposed the concept of d-excess (d-excess=$\delta^{2}H$-8$\delta^{18}O$) to illustrate the intensity of evaporative fractionation. In the state of isotopic equilibrium, the value of the

d-excess is 10. Compared with d-excess, lc-excess can explain the evaporative fractionation process better. The main reason is that lc-excess in precipitation and soil water changes smoothly and has relatively small seasonal changes (Landwehr et al., 2014). The dynamic changes of isotopes record the signal of soil water evaporation. This enrichment from dynamic fractionation exists in soil water isotopes in different climatic regions. Compared with temperate regions, the signals of evaporation in arid and Mediterranean environments penetrate deeper into the soil ( Sprenger et al., 2016b). Some water will be stored in the soil after evaporation and seepage processes. The water storage capacity of humid areas is higher than that of arid areas, the water storage capacity of forests is higher than that of grassland, and the water storage capacity of middle and lower soil layers with higher clay content is higher than that of surface soil layer ( Heinrich et al., 2019; Sprenger et al., 2019; Kleine et al., 2020; Snelgrove et al., 2021).

In alpine mountains, climate warming will accelerate the melting of glaciers and frozen soil, and the dynamic interaction between water bodies stored in different media will become the main focus of the water cycle (Penna et al., 2018). Previous studies on evaporation, infiltration and storage of soil water mostly focused on different climatic regions or vegetation types under the same climatic region. Understanding the climatic and hydrological conditions of different vertical vegetation zones and clarifying the regulating role of vegetation in the water cycle process can help to better adapt to the influence of climate change on the hydrological process in the source area. In this study, the stable isotope composition of precipitation and soil water, and soil water storage's spatiotemporal dynamics were monitored in four vegetation zones (Alpine Meadow, Coniferous Forest, Mountain Grassland and Deciduous Forest) with different hydrothermal conditions in the Xiying River Basin. In order to explore the differences in soil water evaporation, infiltration and storage processes in these four different climates, vegetation and terrain regions, the following research objectives are proposed: (1) Exploring the evolution of isotope evaporation signals and the "memory effects" of precipitation input, mixing and rewetting; (2) Understand the soil water storage capacity and influencing factors of four vegetation areas in the mountain areas.

**Comment 5:** L76-78. Not clear what you mean by this. Are the water resources more unstable or are they transitioning?
**Response:** I have reinterpreted this sentence for the sake of understanding.

L81-83: In alpine mountains, climate warming will accelerate the melting of glaciers and frozen soil, and the dynamic interaction between water bodies stored in different media will become the main focus of the water cycle (Penna et al., 2018).

**Comment 6:** L84. "Heat conditions" do you mean temperatures?

**Response:** We want to express the vegetation zone under different moisture and temperature conditions. Based on this, I re-narrate this sentence.

L89-93: In this study, the stable isotope composition of precipitation and soil water, and soil water storage's spatiotemporal dynamics were monitored in four vegetation zones (Alpine Meadow, Coniferous Forest, Mountain Grassland and Deciduous Forest) with different hydrothermal conditions in the Xiying River Basin.

**Comment 7:** L82-90. These are fine as general aims, but can you explain (briefly) why this is important (i.e. what are you doing that is new, what are the broader implications?). There is a disconnect here between the broad general themes in the rest of the introduction and your specific study. Also, runoff generation and the memory effect are not explicitly discussed in any depth in the paper (need to make sure that your aims are actually what you discuss).

**Response:** Previous studies on soil moisture evaporation, infiltration and storage have mostly focused on different climatic regions or vegetation types under the same climatic region, and there are few uses of isotope technology to trace the hydrological processes in the mountain vegetation vertical zone.

**Comment 8:** L88. If it is important, define the memory effect and explain why we need to understand it.

**Response:** The "memory effect" means that the temporal and spatial changes of the stable isotope profile of soil water can reflect and characterize the input, mixing, and rewetting process of precipitation. Understanding the "memory effect" helps us trace the dynamic changes of climate and soil hydrology.

**Study Area**

**This section needs referencing. Also a few more details as to the spatial variation of rainfall and temperature (I presume that the highest rainfall and lowest mean temperatures are in the mountains?)**

**Response:** I added a reference. Table 1 shows the spatial variation of precipitation and temperature.

**Table 1** Basic data of each Vegetation zone (*Long*-Longitude, *Lat*-Latitude, *Alt*-Altitude, *T*-Air Temperature, *P*-Precipitation Amount, *h*-Relative Humidity)

| Vegetation zone | Geographical parameter | | | Meteorological parametes | | | Number of samples | |
|---|---|---|---|---|---|---|---|---|
| | *Long*(° E) | *Lat*(° N) | *Alt*(m) | *T*(℃) | *P*(mm) | *h*(%) | Precipitation | Soil |
| Alpine Meadow | 101°51'16" | 37°33'28" | 3637 | -0.19 | 595.1 | 69.2 | 72 | 47 |
| Coniferous Forest | 101°53'23" | 37°41'50" | 2721 | 3.34 | 431.9 | 66.6 | 42 | 41 |
| Mountain Grassland | 102°00'25" | 37°50'23" | 2390 | 6.6 | 363.5 | 60.4 | 37 | 54 |
| Deciduous Forest | 102°10'56" | 37°53'27" | 2097 | 7.9 | 262.5 | 59.8 | 40 | 53 |

**Comment 1:** L94. What is a "first-class tributary"?

**Response:** I changed the description of this sentence.

L101-104: As the largest tributary of the Shiyang River, it is formed by Shuiguan River, Ningchang River, Xiangshui River and, Tatu River converging from southwest to northeast, and finally flowing into Xiying Reservoir.

**Comment 2:** L98-101. Probably worth reporting the Koppen Zones.

**Response:** We used the Köppen climate zone.

L106-113: The basin' elevation is between 2000 m and 5000 m, which belongs to a temperate semi-arid climate with strong solar radiation, long sunshine time, and a large temperature difference between day and night. The average annual temperature of the basin is 6℃, the annual average evaporation is 1133 mm, the annual average precipitation is 400 mm, and the precipitation from June to September accounts for 69% of the annual precipitation. The precipitation increases with the elevation, while temperature decreases with the elevation (Table 1) (Ma et al., 2018).

**Comment 3:** L103-106. Refer to Fig. 1.

**Response:** Figure 1 has been referenced here.

L113-116: The zonal differentiation of vegetation in the basin is dominated by Deciduous Forest, Mountain Grassland, Cold Temperate Coniferous Forest, and Alpine Meadows. The soils mainly include lime, chestnut, alpine shrub meadow, and desert soil (Fig.1).

**Comment 4:** Fig. 1. What is the inset on the left-hand map?

**Response:** The complete nine-dotted line is shown here.

**Data and Methods**

**The methods used here are standard and suitable for the project. As with much of the rest of the paper, there are a few details lacking and the explanations are not very clear.**

**Comment 1:** L111-113. It would be helpful here or in Section 2 to outline what 2017 was like in terms of rainfall and temperature as these vary year-by-year. In particular, distinguish between long-term averages and values in the sampling year.

**Response:** I have outlined precipitation and temperature.

L123-127: In 2017, the precipitation in Alpine Meadow, Coniferous Forest, Mountain Grassland and Deciduous Forest were 595.1mm, 431.9mm, 363.5mm and 262.5mm, respectively. The average daily values of Alpine Meadow, Coniferous Forest, Mountain Grassland and Deciduous Forest are -0.19℃, 3.34℃, 6.6℃ and 7.9℃, respectively (Table 1).

**Comment 2:** L116-122. This is rather a clunky description (not sure that you need to specify explicitly that you wrote dates on bottles). What do you mean by "four parallel samples" being also collected?

**Response:** I re-narrate the soil sampling process.

L130-134: Three duplicate samples were collected for each soil layer. The collected soil sample was placed into a 50 mL glass bottle, the bottle mouth was sealed with Parafilm marked with the sampling dates, and then frozen for storage until experimental analysis. Each sample was collected separately in an aluminium box.

**Comment 3:** L143. It's "permil" not "thousands". As written, "thousandths of the Vienna Standard Mean Ocean Water (VSMOW)" is meaningless.

**Response:** This error has been corrected.

L155-157: The analysis results were expressed in peril of the Vienna Standard Mean Ocean Water (VSMOW):

**Comment 4:** Section 3.2. The analysis is only part of the uncertainty. Did you perform multiple extractions on the same sample to test the uncertainty associated with that. This will undoubtedly be higher and needs to be considered.

**Response:** During the sampling process, we collected duplicate samples to improve the accuracy of the experiment.

**Comment 5:** Section 3.3.1 The line-conditioned excess is less used than the d-excess (but is potentially more informative). You should explain what it is (and define the term). The explanation "The physical meaning of lc-excess is expressed as the deviation degree between isotopic values in samples and LMWL, which indicates the non-equilibrium dynamic fractionation process caused by evaporation (Landwehr et al., 2014; Sprenger et al., 2017)" is not very clear.

**Response:** Based on your suggestion, we described lc-excess in detail.

L163-177: The linear relationship between $\delta^2H$ and $\delta^{18}O$ in precipitation and soil water is defined as the LMWL (local meteoric water line) and SWL (soil waterline), respectively, which is of great significance for studying the evaporative fractionation of stable isotopes during the water cycle. We further calculated the line-conditioned excess for each soil water and precipitation sample. The lc-excess in different water bodies can characterize the evaporation index of different water bodies relative to local precipitation (Landwehr and Coplen, 2006).

$$lc-excess = \delta^2H - a \times \delta^2H - b \tag{2}$$

Where *a* and *b* are the slope and intercept of LMWL, respectively, and $\delta^2H$ and $\delta^{18}O$ are the isotopic values of hydrogen and oxygen in the sample. The physical meaning of lc-excess is expressed as the degree of deviation of the isotope value in the sample from the LMWL, indicating the non-equilibrium dynamic fractionation process caused by evaporation. Generally, the change of lc-excess in local precipitation is mainly affected by different water vapor sources, and the annual average is 0. Since the stable isotopes in soil water are enriched by evaporation, the average lc-excess is usually negative (Landwehr et al., 2014; Sprenger et al., 2017).

**Comment 6:** Section 3.3.2. More details are needed as to where these data are derived from. Are they local data measured at the field site or interpolated estimates? Application of the Penman-Monteath equation is very data sensitive. What do you think the errors are here?

**Response:** These data are from nearby weather stations. According to the literature, we revised the formula.

L179-186: Calculation of potential evapotranspiration based on Penman-Monteath equation (Allen et al., 1998):

$$PET = \frac{0.408\Delta(R_n - G) + \gamma \frac{900}{T + 273} u^2 (e_s - e_a)}{\Delta + \gamma(1 + 0.34u^2)}$$
(3)

Where PET is the daily potential evapotranspiration (mm day$^{-1}$), $R_n$ is net radiation (MJ m$^2$ day$^{-1}$), $G$ is soil heat flux density (MJ m$^2$ day$^{-1}$), $\gamma$ is psychrometric constant (kPa$^{\circ}$C$^{-1}$), $u_2$ is the wind speed at 2 m height (m s$^{-1}$), $T$ is mean daily air temperature at 2 m height ($^{\circ}$C), $\Delta$ is slope vapor pressure curve (kPa$^{\circ}$C$^{-1}$), $e_a$ is actual vapor pressure (kPa) and $e_s$ is saturated vapor pressure (kPa). These data come from nearby weather stations.

**Comment 7:** Section 3.3.3. These are based on your measurements, yes? Again, do you have estimates of uncertainties. Also, some of the techniques (eg moisture content) need more detail.

**Response:** This data comes from our actual measurement, which is constant, and we added the calculation of soil moisture.

L191-194: $W$ is gravimetric water content, which is expressed by formula as follows:

$$W = \frac{M_1 - M_2}{M_2} \times 100\%$$
(5)

In the formula: $M_1$ is the gravimetric of the wet soil (g), and $M_2$ is gravimetric of the dry soil (g).

**Results**
**This section suffers from the shortcomings of the rest of the paper. The explanations are not very clear and are often overly general. Also, I cannot see where the raw data are (no Table or Appendix); presenting the actual data is required.**

**Comment 1:** L175-178. How precise are these values (i.e. is the 1dp precision warranted)? What was the rainfall during those times?

**Response:** We use FAO Penman-Monteith to calculate the potential daily evapotranspiration (possible evapotranspiration) in the study area (the software calculation results are kept to three decimal places, and we keep one decimal place in the study). This illustrates the date when the maximum and minimum values appear, and there may be no rainfall on that day.

**Comment 2:** L180-182. Not very clearly worded.

**Response:** I restated this sentence.

L201-203: The input of summer precipitation and ice-snow meltwater increases the runoff, resulting in a trend similar to PET.

**Comment 3:** L213-220. The ranges in stable isotope values are probably more useful. Suggest that you report the range and the mean (you can omit the SD as that is less useful). Also report the number of observations, so we get an idea of how much data there is. Ideally the mean should be weighted by precipitation amount (it is not clear that that is the case, but you should be clear whether it is).

**Response:** We describe the average value and variation range of the isotope, preserve the SD, and add the number of observations according to the recommendations. The average amount of isotopes is based on the weighted average of precipitation or soil water content.

L235-244: The mean values of $\delta^2H$ and $\delta^{18}O$ in Alpine Meadow (Number of samples: 72) were -73.1‰±36.3‰ (-163.9‰~13.7‰) and -10.0‰±4.3‰ (-23.1‰~-1.3‰), respectively. The average values of $\delta^2H$ and $\delta^{18}O$ of Coniferous Forest (Number of samples: 42) were -42.0‰±37.2‰ (-117.8‰~13.0‰) and -7.1‰±4.7‰ (-17.4‰~-0.1‰), respectively. The average values of $\delta^2H$ and $\delta^{18}O$ of Mountain Grassland (Number of samples: 37) were -37.4‰±30.5‰ (-103.1‰~4.2‰) and -5.9‰±3.9‰ (-15.1‰~-0.9‰), respectively. The average values of $\delta^2H$ and $\delta^{18}O$ of Deciduous Forest (Number of samples: 40) were -31.8‰±42.8‰ (-110.2‰~23.2‰) and -5.8‰±5.5‰ (-15.2‰~3.2‰), respectively (Table 2).

**Comment 4:** L220-224. Poorly worded.

**Response:** I restated this sentence.

L244-250: The maximum isotopic values of the four vegetation zones appeared on August 4 (Alpine Meadow: 13.7‰, $\delta^2H$; -1.3‰, $\delta^{18}O$), August 10 (Coniferous Forest: 13.0‰, $\delta^2H$; -0.1‰, $\delta^{18}O$), August 7 (Mountain Grassland: 4.2‰, $\delta^2H$; -0.9‰, $\delta^{18}O$) and August 13 (Deciduous Forest: 23.2‰, $\delta^2H$; 3.2‰, $\delta^{18}O$), respectively. The highest temperature in each vegetation zone appeared on July 27. The high temperature caused the precipitation to undergo strong below-cloud evaporation during the fall, leading to the enrichment of isotopes.

**Comment 5:** L224-227. This isn't really that obvious from Fig. 3. Can you report the magnitudes in the text?

**Response:** Here, we have shown the seasonal variation trend of precipitation isotope and added the maximum value of each vegetation zone and the time of its appearance in the previous analysis.

L244-250: The maximum isotopic values of the four vegetation zones appeared on August 4 (Alpine Meadow: 13.7‰, $\delta^2$H; -1.3‰, $\delta^{18}$O), August 10 (Coniferous Forest: 13.0‰, $\delta^2$H; -0.1‰, $\delta^{18}$O), August 7 (Mountain Grassland: 4.2‰, $\delta^2$H; -0.9‰, $\delta^{18}$O) and August 13 (Deciduous Forest: 23.2‰, $\delta^2$H; 3.2‰, $\delta^{18}$O), respectively. The highest temperature in each vegetation zone appeared on July 27. The high temperature caused the precipitation to undergo strong below-cloud evaporation during the fall, leading to the enrichment of isotopes.

**Comment 6:** L232-260. This section has too little detail in it to follow. You need to explain the data more specifically (avoid vague terms such as "depletion" or "enrichment" and report some values). More importantly where are the data? Fig. 4 is labelled as a "heat map" but seems to be the values (I think) and they are on Fig. 5. However, these also need to be in a Table somewhere.

**Response:** According to your suggestion, I plot the data into a table and use the data for analysis.

L262-298: The low-temperature environment and abundant precipitation events in alpine meadows make the monthly average of $\delta^2$H and $\delta^{18}$O of soil water more depleted than other vegetation zones (-69.4‰~-51.6‰, $\delta^2$H; -7.5‰~-10.3‰, $\delta^2$H). Despite this, SWlc-excess of most samples in this station was still negative, and there were different degrees of evaporation in the process of precipitation penetrating the soil and mixing with original pore water, among which evaporation fractionation was stronger in July (-11.9‰, lc-excess) and October (-14.5‰, lc-excess). Soil water isotopes of Coniferous Forest gradually changed seasonally. From April to July, precipitation was scarce, the temperature rose, and the isotopes of soil water were gradually enriched on the surface (-52.7‰~-29.5‰, $\delta^2$H; -7.0‰~-2.1‰, $\delta^2$H), reaching the peak value of the observation period in July (-29.5‰, $\delta^2$H; -2.1‰, $\delta^{18}$O), and continuous rainfall input from late July to mid-August resulted in soil water isotopes depletion (-57.0‰, $\delta^2$H; -8.1‰, $\delta^{18}$O). SWlc-excess was an obvious fractionation signal opposite to the trend of isotope change, reaching the lowest value (-26.3‰) in the sampling period in July, and the change of air temperature and precipitation controlled the evaporation intensity. From April to July, the isotopic value of surface soil water in Mountain Grassland was higher ($\delta^{18}$O was greater than zero), and SWlc-excess was lower than -30‰. During this period, evaporation and fractionation of shallow soil water

[revised manuscript text omitted]

**Comment 7:** L266-269. Speculative, can you provide a reference to show that these processes cause secondary evaporation.

**Response:** I have included references here. In addition, based on other reviewers ' suggestions, this part was moved to the discussion.

L373-376: In Alpine Meadow, due to low atmospheric temperature, low cloud base height, and low air-saturated water vapor loss, it is weakly affected by secondary evaporation during precipitation. The slope of LMWL (8.4) is even higher than that of GMWL (Zhang et al., 2012).

**Comment 8:** L270-275. A reference would also help here

**Response:** I have added references here. In addition, based on the suggestions of other reviewers, this part was moved to the discussion.

L376-378: As the altitude decreases, the secondary evaporation under the cloud strengthens, and the slope of the LMWL of each vegetation zone decreases (Pang et al., 2011) (Fig 5).

**Comment 9:** L275-276. Seems redundant as I'm not sure where else the moisture could come from.

**Response:** This part was deleted.

**Comment 10:** L288-295. Again, lacks detail. It is difficult to follow these arguments when the data is discussed in very vague terms.

**Response:** I used data to describe this part of the content.

L320-327: During the study period, compared with other vegetation belts, the surface isotopic value of soil water in Mountain Grassland was relatively enriched (-24.3‰, $\delta^2H$; -0.8‰, $\delta^{18}O$), the lc-excess was smaller and deeper into the middle and lower soil layers (-25.8‰), and the GWC was relatively low (8.4%). Because of the difference in vegetation types and the influence of reservoirs, this change did not have the elevation effect completely. Although the elevation was low, the soil water of Deciduous Forest had more depleted isotopic characteristics and higher soil moisture than Mountain Grassland in most samples.

**Comment 11:** L298. What is "dynamic fractionation"?

**Response:** Here should be "evaporation".

L328-331: The low-temperature natural environment made Alpine Meadow soil-less affected by evaporation (lc-excess > -20 ‰), and GWC was at a high value (GWC > 20%) during the whole study period.

**Comment 12:** L296-309. As with much of the rest of this section, I struggled to follow the details. The explanations are not clear, there are a fair number of general statements that lack detail, and a number of findings that are not obvious. For example, "Evaporation signal can easily penetrate deep soil, which made the GWC value of all sampling activities at this site lower than 20% (Fig.6)" which seems to be at odds with "With the increase of soil depth, the fractionation signal gradually weakened".

**Response:** I used data to describe this part of the content.

L327-339: Soil profiles obtained from different vegetation zones can reflect the evaporation signals of water. The low-temperature natural environment made Alpine Meadow soil-less affected by evaporation (lc-excess > -20 ‰), and GWC was at a high value (GWC > 20%) during the whole study period. The surface soil water of Coniferous Forest was easily affected by climate and had higher isotopic composition (-29.5‰, $\delta^2H$; -2.1‰, $\delta^{18}O$) and lower lc-excess (-26.3‰). Due to evaporation, soil water isotopes in Mountain Grassland and Deciduous Forest were enriched in the surface soil layer. Especially in the Mountain Grassland, the average values of $\delta^2H$ and $\delta^{18}O$ in 0-10cm soil layer were as high as -24.4‰ and -1.2‰, respectively, and SWlc-excess was lower than -25‰, even close to -40‰ in some samples. Evaporation signal can easily penetrate deep soil, which made the GWC value of all sampling activities at this site lower than 20% (Fig. 4) (Fig. 6).

**Discussion**

**This section has some interesting ideas in it but it is not well linked to the data in the study. You need to show how the data that you collected informs our understanding. Some of the later part of this section is written more like an introductory literature review.**

**Section 5.1**

L325-354. Some of this section describes the data (the observations on soil moisture) and that material belongs in the results.

**Response:** We reanalyzed Figure 7 in the results.

L343-365:**4.4 Variation of water storage capacity of 0-40cm soil layer in different vegetation areas**

This study used soil water to calculate the water storage of the 0-40cm soil layer in the four vegetation zones during the observation period (Fig 7). The water storage capacity of the Alpine Meadow gradually decreases from April to July (209.7 mm~167.2 mm), and the water storage capacity increases after July (167.2 mm~201.8 mm). The monthly average water storage capacity is the least for 0-10 cm (43.0 mm) and the most for 30-40 cm (51.7 mm). The water storage capacity of the Coniferous Forest gradually decreases from April to July (150.1 mm~101.2 mm), and the water storage capacity increases after July (101.2 mm~160.0 mm). The monthly average water storage capacity is the least for 0-10 cm (28.0 mm) and the most for 30-40 cm (40.0 mm). The water storage capacity of the Mountain Grassland gradually decreases from April to July (80.3 mm~64.0 mm), and the water storage capacity increases after July (64.0 mm~104.6 mm). The monthly average water storage capacity is the least for 0-10 cm (17.5 mm) and the most for 20-30 cm (22.0 mm). The water storage capacity of the Deciduous Forest gradually decreases from April to June (159.3 mm~104.0 mm), the water storage capacity increases from June to August (104.0 mm~154.0 mm), and there is a decrease from August to October (154.0 mm~111.8 mm).The monthly average water storage capacity is the least for 0-10 cm (29.1 mm) and the most for 20-30 cm (35.0 mm). In general, the soil water storage capacity of the 0-10 cm soil layer is less than that of other soil layers. The order of the water storage capacity of the 0-40 cm soil layer in the four vegetation zones is AM > DF > CF > MG.

**Section 5.2**

This section does not link well with the results. It is difficult to follow how your data help you make these conclusions. More justification and explanations are required. Moreover, there is little discussion of processes here – how does the data help understand how processes operate? You have concentrated on discussing the isotopic variability, without using it to understand what is going on.

This is the section where you should discuss aspects such as the memory effect and runoff generation, but you do not do so.

**Response:** According to your suggestion, I improved the discussion of this part.

**Section 5.3.**

This section reads more like an introduction. It is not clear how what you have done in this study relates to these broad general findings. As with the Introduction, you need to make a clearer link between your study and these general statements. This are all important issues, but there needs to be linkages.

Climate change in mentioned several times, but it is not clear how your study informs our understanding of its impacts. Those types of links need to be made clearer if they exist. Likewise, there are comments about groundwater recharge and runoff but no indication of how your results help understand those processes. Runoff generation was not actually discussed in the body of the paper (it appears in the introduction and the end of the discussion, but not in the discussion of the specific results).

Same comments apply to: subsurface runoff (presume that you mean interflow?); the management practices; human activities. These are topics that all appear in this section with no real link to the data in the rest of the paper.

There are also a number of superfluous details here. For example, why is mining waste (L426-428) relevant to this study.

**Response:** We logically sorted out the full text based on the reviewers' comments. The discussion on runoff generation in the watershed does not match the theme of this manuscript (Evaporation, infiltration and storage of soil water in different vegetation zones in the Qilian mountains: a stable isotope perspective). Therefore, the manuscript focused on soil moisture's evaporation, infiltration, and storage mechanism in the study area. Based on this,

we reorganized this part of the content. Your comments have further improved the logic of the article.

L370-494: **5.1 Evaporation of soil moisture in different vegetation zones**

In the arid river source area, the replenishment of soil moisture mainly comes from precipitation. The slope of the regional atmospheric precipitation line can reflect the strength of local evaporation. In Alpine Meadow, due to low atmospheric temperature, low cloud base height, and low air-saturated water vapor loss, it is weakly affected by secondary evaporation during precipitation. The slope of LMWL (8.4) is even higher than that of GMWL (Zhang et al., 2012). As the altitude decreases, the secondary evaporation under the cloud strengthens, and the slope of the LMWL of each vegetation zone decreases (Pang et al., 2011) (Fig 5). The dynamic changes of lc-excess of soil profiles in different vegetation areas reflect the process of soil water evaporation caused by drought during the study period. The monthly average value of SWlc-excess in Alpine Meadow was less than 0, and the minimum value was -11.9‰ (July). Although the vegetation belt is subject to different degrees of evaporation each month, it is less affected by drought and it is difficult for evaporation to penetrate into the middle and lower soils. The SWlc-excess of the Coniferous Forest belt is greater than that of the alpine meadow from April to June. The evaporation is the strongest in July (-11.2‰, lc-excess). Similar to alpine meadows, evaporation mainly occurs in the top soil. The vegetation coverage of Mountain Grassland is low, and the arid environment makes the isotopes of the surface soil produce strong evaporation signals (lc-excess is close to -40‰). In most samples, the SWlc-excess of the 60-80 cm soil layer is negative. The evaporation signal moves to the lower layer of the soil. (Zimmermann et al., 1966; Barnes and Allison, 1988). Similar evaporation signals have been found in the Mediterranean and arid climate regions (Sprenger et al., 2016b; McCutcheon et al., 2017). Evaporation signal only exists in the surface soil in humid areas, and there is no difference between lc-excess and 0 in the soil layer below 20cm (Sprenger et al., 2017). The monthly surface soil evaporation of Deciduous Forest is less than that of Mountain Grassland from April to June, and it is greater than mountain grassland after July, which is mainly due to the influence of vegetation and reservoirs. 
[revised manuscript text omitted]

**Conclusions**

As with the discussion, the links to the study are not well made. In some ways this material is less general than some of the latter parts of the Discussion (Section 5.3) and it would be worth reordering so that you have the more general ideas at the end.

**Response:** Based on the opinions of the three reviewers, we re-summarized the conclusions of the manuscript.

L496-522: This work provides further insights into the movement and mixing of soil water in different vegetation zones in arid source regions. During the study period, the dynamic changes of lc-excess in soil profiles of different vegetation zones reflected the evaporation signals caused by drought. Soil water evaporation in spring and summer and insufficient precipitation during the drought period were the main driving forces leading to isotopic enrichment in surface soil. The results show that: The evaporation intensity of the four vegetation zones was: Mountain Grassland > Deciduous Forest > Coniferous Forest > Alpine Meadow. In the Mountain Grassland and Deciduous Forest zone, drought caused the evaporation signal to penetrate deep into the middle and lower soils. The SWlc-excess below 70 cm of the ground surface was still negative. Soil water isotopes and GWC record the process of soil rewetting caused by precipitation input and mixing. Alpine Meadow and

Coniferous Forest zones were enriched in precipitation. After a short period of weak evaporation, the soil will be rewetted by the next precipitation. There was only sporadic precipitation in Mountainous Grassland and Deciduous Forest belt from mid-May to late July. After July, the temperature dropped and continuous precipitation made the soil wet again after two months of drought. The Mountain Grassland and Deciduous Forest zone had only sporadic precipitation from mid-May to late July. With the decrease of air temperature and continuous precipitation after July, the soil was re-wetted after two months of drought. Moisture and temperature conditions were the key factors that restrict the soil water storage capacity in different vegetation zones. Each vegetation area's soil water storage capacity is: Alpine Meadow > Deciduous Forest > Coniferous Forest > Mountainous Grassland. The water storage capacity of the surface soil in each vegetation zone was weak, and more water was stored in the middle and lower soil with higher clay content. This research is helpful to understand the hydrological process in different vegetation areas and to provide theoretical support for the realization of regional ecological hydrological balance.

---

## Author Comment (AC3)

**Revision Notes**

Dear Editor and Reviewers:

Thank you for your letter and for the reviewers' comments concerning our manuscript entitled "Evaporation, infiltration and storage of soil water in different vegetation zones in Qilian mountains: From a perspective of stable isotopes" (Manuscript Number: Hess-2021-376).

According to the reviewers' comments, we have revised our manuscript carefully. The revised portions have been marked in red in the revised version of manuscript. The main corrections and the response to the reviewers' comments are as follows.

**Responses to the reviewer's comments:**

**Response to Reviewer #3**

**Reviewer#3: This paper presents an interesting hydrological and runoff study from the Qilian region where water and soil water samples were obtained across different climatic, topographical and vegetative conditions in order to understand the infiltration, evaporation and storage processes. The paper is well structured, but major issues need fixing as also suggested by the other referees. Overall, the English language needs to be proofread and words such as "obvious" should be avoided. The Abstract needs substantial work to emphasize the purpose/objectives of the study, describe the methods used, and to relate quantifiable results. Further discussion of the results themselves is required as well as linking the results obtained (what are the observations withdrawn from the data) to previous research. Please see additional comments below.**

Thanks for your comments.

**Abstract**

**Comment 1:** L11-12: Is this really true? That in arid areas most of the water comes from mountains? How about low lands? And groundwater? I think this sentence is not needed.

**Response:** Based on your suggestion, we have deleted it.

**Comment 2:** L12: should be the "processes" not "process".

**Response:** This spelling error has been corrected.

**Comment 3:** L13: "have" instead of "has".

**Response:** This spelling error has been corrected.

L10-13: The processes of water storage and runoff generation have not been fully understood in different vegetation zones in mountainous areas, which is the main obstacle blocking human cognition of hydrological processes and water resources assessment.

**Comment 4:** L15: instead of "In current study" use "In this study"

**Response:** We corrected this.

**Comment 5:** L15-17: This is an important sentence that summarizes the work done. I would suggest to rewrite it being more specific to which isotopes, which types of vegetation zones and why this is needed.

**Response:** Based on suggestions from you and other reviewers, we rewrite this part.

L13-17: In order to further understand the process of soil water movement and runoff generation in different vegetation zones (Alpine Meadow, Coniferous Forest, Mountain Grassland and Deciduous Forest) in mountain areas, this study monitored the temporal and spatial dynamics of hydrogen and oxygen stable isotopes in the precipitation and soil water of the Xiying River.

**Comment 6:** L17: Weak compared to what? Results should be quantified instead of using "weak" and "save up".

**Response:** We have clarified the evaporation intensity through the SWlc-excess of each vegetation zone and verified it through meteorological data.

L17-23: The evaporation intensity of the four vegetation zones was: Mountain Grassland > Deciduous Forest > Coniferous Forest > Alpine Meadow. Alpine meadows and coniferous forests only had evaporation in the topsoil, and the rainfall input was fully mixed with each layer of soil. Evaporation signals of mountain grasslands and deciduous forests could penetrate deep into the middle and lower layers of the soil, and precipitation quickly flows into the deep soil rapidly through the soil matrix.

**Comment 7:** L19-21: What is the result in the paper that lead to this hypothesis? The authors need to add evidence of this instead of speculating.

**Response:** This part of the result has been changed, we have added data and information.

L17-27: The results show that: The evaporation intensity of the four vegetation zones was: Mountain Grassland > Deciduous Forest > Coniferous Forest > Alpine Meadow. Alpine

meadows and coniferous forests only had evaporation in the topsoil, and the rainfall input was fully mixed with each layer of soil. Evaporation signals of mountain grasslands and deciduous forests could penetrate deep into the middle and lower layers of the soil, and precipitation quickly flows into the deep soil rapidly through the soil matrix. The soil water storage capacity order in each vegetation zone was: Alpine Meadow > Deciduous Forest > Coniferous Forest > Mountain Grassland. In addition, the water storage capacity of 0-10 cm soil was weak, and the water storage capacity of 10-40 cm was strong.

**Comment 8:** L22: What is evaporate strongly? How much?

**Response:** We have clarified the evaporation intensity through the SWlc-excess of each vegetation zone and verified it through meteorological data.

L17-19: The results show that: The evaporation intensity of the four vegetation zones was: Mountain Grassland > Deciduous Forest > Coniferous Forest > Alpine Meadow.

**Comment 9:** L21-22: The lower elevation vegetation zones within the Mountain Grassland and Deciduous forest? Aren't these areas at high altitude?

**Response:** This sentence was restated. Previously, it was to express that Mountain Grassland (2390 m) and Deciduous Forest (2097 m) are relatively low-altitude areas compared to Alpine Meadow (3637 m) and Coniferous Forest (2721 m).

**Comment 10:** L25: Delete word "reasonably"

**Response:**"Reasonably" has been deleted.

L27-28: This work will provide a new reference for the process of soil hydrology in the arid headwaters area.

**Introduction**

**Comment 1:** L39-40: Soil water in the unsaturated zone from precipitation can transform into water vapour or groundwater recharge.

**Response:** I have modified this sentence according to your suggestion.

L39-40: Soil water in the unsaturated zone can be converted to steam or groundwater recharge from precipitation.

**Comment 2:** Line 40: Delete "Its".

**Response:** "Its" has been deleted.

**Comment 3:** Line 41: Delete "very".

**Response:** This word "very" has been deleted.

**Comment 4:** Line 48: Storage is not a transport mechanism

**Response:** I agree with your comment, this problem has been corrected.

L49-50: The water seepage in the unsaturated soil zone and the evaporation of water at the air-soil interface are the main forms of soil water transport.

**Comment 5:** Line 54: is it soil water profiles?

**Response:** This part of the content has been deleted.

**Comment 6:** Line 56: Delete "In addition,".

**Response:** This part of the content has been deleted.

**Comment 7:** Line 58: Describe what the d-excess is.

**Response:** "d-excess" has been introduced in detail.

L65-67: Dansgaard (1964) proposed the concept of d-excess (d-excess=$\delta^2$H-8$\delta^{18}$O) to illustrate the intensity of evaporative fractionation. In the state of isotopic equilibrium, the value of the d-excess is 10.

**Comment 8:** Line 66-70: Delete ",and" and rewrite following sentence. It is not clear at the moment.

**Response:** This part of the content has been deleted.

**Comment 9:** Line 71: Do not use "Generally speaking" in a scientific publication

**Response:** "Generally speaking" has been deleted.

**Comment 10:** Line 71: Do "wet" areas refer to tropical regions?

**Response:** This should be a "humid area", which refers to an area with humid air and abundant rainfall.

**Comment 11:** Line 80: can better help adapt.

**Response:** This place has been corrected.

L86-89: Understanding the climatic and hydrological conditions of different vertical vegetation zones and clarifying the regulating role of vegetation in the water cycle process can help to better adapt to the influence of climate change on the hydrological process in the source area.

**Comment 12:** Line 82: "In this study" instead of "In current study".

**Response:** This place has been corrected.

L89-93: In this study, the stable isotope composition of precipitation and soil water, and soil water storage's spatiotemporal dynamics were monitored in four vegetation zones (Alpine Meadow, Coniferous Forest, Mountain Grassland and Deciduous Forest) with different hydrothermal conditions in the Xiying River Basin.

**Comment 13:** Line 82: "," after soil water.

**Response:** This place has been corrected.

L89-93: In this study, the stable isotope composition of precipitation and soil water, and soil water storage's spatiotemporal dynamics were monitored in four vegetation zones (Alpine Meadow, Coniferous Forest, Mountain Grassland and Deciduous Forest) with different hydrothermal conditions in the Xiying River Basin.

**Comment 14:** Line 83: Is it in four regions of different climate, vegetation and topographical conditions? As opposed to vegetation zones?

**Response:** Our research objects are four typical vertical vegetation belts in mountainous areas, and their climatic conditions, topography and dominant species are different.

**Comment 15:** Line 85: Then, it can be clarified that this study explores how evaporation, infiltration and storage processes differ within these four regions according to the climate, vegetation and topography.

**Response:** Yes, this was confirmed in the follow-up discussion.

L93-98: In order to explore the differences in soil water evaporation, infiltration and storage processes in these four different climates, vegetation and terrain regions, the following research objectives are proposed: (1) Exploring the evolution of isotope

evaporation signals and the "memory effects" of precipitation input, mixing and rewetting; (2) Understand the soil water storage capacity and influencing factors of four vegetation areas in the mountain areas.

:

**Comment 16:** Line 89: similarly to the previous comment, are the authors restricting the analysis to only vegetation zones? I would argue that the study compares regions with varying climatic, topographic and vegetative conditions.

**Response:** The climatic conditions, topography and dominant species of the vegetation zone are different. I think the description of the vegetation zone already contains the similarities and differences of these natural conditions.

**Study Area**

**Comment 1:** Line 99: ranges between 2000m and 5000m above sea level

**Response:** I revised this sentence.

L106-109: The basin' elevation is between 2000 m and 5000 m, which belongs to a temperate semi-arid climate with strong solar radiation, long sunshine time, and a large temperature difference between day and night.

**Data and Methods**

**Comment 1:** L110: Delete "and determination"

**Response:** This words "and determination" have been deleted.

**Comment 2:** L116: What does "parallel" mean here?

**Response:** This is a spelling error, it should be "duplicate samples". Collect duplicate samples to improve the accuracy of the experiment.

**Comment 3:** L170-172: Equation before line 170 needs reference.

**Response:** References have been added.

L188-189: Soil water storage is the thickness of water layer formed by all water in a certain soil layer (Milly, 1994)

**Results and Analysis**

**Comment 1:** L175: "PET" should be written Potential evapotranspiration (PET), then the authors can use PET but it needs complete spelling the first time it is used.

**Response:** Full spelling is used for the first time.

L178: **3.3.2 PET (Potential evapotranspiration)**

**Comment 2:** L177: I assume it is also the daily evapotranspiration? Need to make it explicit which type of evaporation.

**Response:** I double-checked the usage of the text. We calculated the daily potential evapotranspiration during the study period, and obtained the potential evapotranspiration for each month and the entire observation period by summing.

**Comment 3:** L184: Delete "generally speaking".

**Response:** This words "generally speaking" has been deleted.

**Comment 4:** L191: "temperature" instead of heat

**Response:** I have modified this.

L210-212: To explore the differences of the natural environment in different vegetation zones, air temperature, atmospheric humidity and precipitation were used to indicate each research site's temperature and moisture conditions.

**Comment 5:** L194: 72 precipitation events? Make it explicit, where all these rainfall?

**Response:** I have explained in detail here.

L215-217: During the observation period, there were 72 precipitation events in the Alpine Meadow zone, the total precipitation was 534.3 mm, which was relatively evenly distributed in each month.

**Comment 6:** L207: Rewrite sentence to "The temperature of the studied regions was ordered as follow:"

**Response:** According to your suggestion, this part has been rewritten.

L228-231: The temperature of the studied regions was ordered as follow: AM (Alpine Meadow) < CF (Coniferous Forest) < MG (Mountain Grassland) < DF (Deciduous Forest), and the humidity of the studied regions was ordered as follow: AM > CF > MG > DF (Fig. 2).

**Comment 7:** L208: Define first what AM, CF, MG, and DF mean

**Response:** Full spelling is used for the first time.

L228-231: The temperature of the studied regions was ordered as follow: AM (Alpine Meadow) < CF (Coniferous Forest) < MG (Mountain Grassland) < DF (Deciduous Forest), and the humidity of the studied regions was ordered as follow: AM > CF > MG > DF (Fig. 2).

**Comment 8:** L213: Do not use obviously in scientific publications, you can say what it was significantly different? Did you do any statistical analysis to conclude this? If so what please mention it in the results

**Response:** This has been rewritten, and the difference is analyzed in the subsequent results.

L233-235: Influenced by different water sources and complex weather conditions in the precipitation process, the isotopic composition of precipitation in four vegetation zones was different during the study period.

**Comment 9:** L211-220. This info would be better in a table

**Response:** Based on your suggestion, we have drawn a table.

L254-256: **Table 2** General characteristics of precipitation $\delta^2H$ and $\delta^{18}O$ in different vegetation areas from April to October 2017

| Vegetation zone | $\delta^2H$/‰ | | | | $\delta^{18}O$/‰ | | | |
|---|---|---|---|---|---|---|---|---|
| | Max | Min | mean | SD | Max | Min | mean | SD |
| AM | 13.7 | -163.9 | -73.1 | 36.3 | -1.3 | -23.1 | -10.0 | 4.3 |
| CF | 13.0 | -117.8 | -42.0 | 37.2 | -0.1 | -17.4 | -7.1 | 4.7 |
| MG | 4.2 | -103.1 | -37.4 | 30.5 | -0.9 | -15.1 | -5.9 | 3.9 |
| DF | 23.2 | -110.2 | -31.8 | 42.8 | 3.2 | -15.2 | -5.8 | 5.5 |

**Comment 10:** L235: "The low temperature environment of Alpine Meadow and abundant and uniform precipitation events made the monthly mean values of $\delta$ 2H and $\delta$ 18O change little" how much?

**Response:** Based on the opinions reviewer, this part' content has been revised.

L262-265: The low-temperature environment and abundant precipitation events in alpine meadows make the monthly average of $\delta^2H$ and $\delta^{18}O$ of soil water more depleted than other vegetation zones (-69.4‰~-51.6‰, $\delta^2H$; -7.5‰~-10.3‰, $\delta^2H$).

**Comment 11:** L239: "Evaporation fractionation of soil water isotopes in Coniferous Forests was more intense." More intense than what? These kind of statements need quantification.

**Response:** According to the opinions of the three reviewers, this sentence is meaningless here. This part mainly analyzes the temporal changes of soil water isotopes in different vegetation zones. Therefore, I deleted this part of the content.

**Comment 12:** L277-278: "With the decrease of altitude, the soil water evaporation became stronger and stronger, except soil in Deciduous Forest". This sentence does not make sense, please rewrite and quantify stronger.

**Response:** Based on your suggestion, I have changed and explained it here.

L308-312: With the decrease in altitude, the slope of the soil waterline in all vegetation zones except for the Deciduous Forest soil waterline decreases (Alpine Meadow: 6.4; Coniferous Forest: 4.7; Mountain Grassland:3.4; Deciduous Forest:4.1), indicating that the evaporation of soil moisture is getting stronger.

**Discussion**

**I am in agreement with the comments of Referee 1 and Referee 2 concerning the discussion. It feels more like a summary of previous studies. The authors need to refer to the results and put them in context of previous work and how their study is contributes to that pool of knowledge.**

**Comment 1:** L323-325: "The soil water storage capacity of Alpine Meadow with low temperature and rainy weather was obviously higher than that of other vegetation zones." The authors need to explain how this conclusion is evident from their data without using words such as "obviously" referred to figure 7 for discussion.

**Response: Response:** We logically sorted out the full text based on the reviewers' comments. The discussion on runoff generation in the watershed does not match the theme of this manuscript (Evaporation, infiltration and storage of soil water in different vegetation zones in the Qilian mountains: a stable isotope perspective). Therefore, the manuscript focused on soil moisture's evaporation, infiltration, and storage mechanism in the study area. Based on this, we reorganized this part of the content. Your comments have further improved the logic of the article.

L370-494: **5.1 Evaporation of soil moisture in different vegetation zones**

In the arid river source area, the replenishment of soil moisture mainly comes from precipitation. The slope of the regional atmospheric precipitation line can reflect the strength of local evaporation. In Alpine Meadow, due to low atmospheric temperature, low cloud base

height, and low air-saturated water vapor loss, it is weakly affected by secondary evaporation during precipitation. The slope of LMWL (8.4) is even higher than that of GMWL (Zhang et al., 2012). As the altitude decreases, the secondary evaporation under the cloud strengthens, and the slope of the LMWL of each vegetation zone decreases (Pang et al., 2011) (Fig 5). The dynamic changes of lc-excess of soil profiles in different vegetation areas reflect the process of soil water evaporation caused by drought during the study period. The monthly average value of SWlc-excess in Alpine Meadow was less than 0, and the minimum value was -11.9‰ (July). Although the vegetation belt is subject to different degrees of evaporation each month, it is less affected by drought and it is difficult for evaporation to penetrate into the middle and lower soils. The SWlc-excess of the Coniferous Forest belt is greater than that of the alpine meadow from April to June. The evaporation is the strongest in July (-11.2‰, lc-excess). Similar to alpine meadows, evaporation mainly occurs in the top soil. The vegetation coverage of Mountain Grassland is low, and the arid environment makes the isotopes of the surface soil produce strong evaporation signals (lc-excess is close to -40‰). In most samples, the SWlc-excess of the 60-80 cm soil layer is negative. The evaporation signal moves to the lower layer of the soil. (Zimmermann et al., 1966; Barnes and Allison, 1988). Similar evaporation signals have been found in the Mediterranean and arid climate regions (Sprenger et al., 2016b; McCutcheon et al., 2017). Evaporation signal only exists in the surface soil in humid areas, and there is no difference between lc-excess and 0 in the soil layer below 20cm (Sprenger et al., 2017). The monthly surface soil evaporation of Deciduous Forest is less than that of Mountain Grassland from April to June, and it is greater than mountain grassland after July, which is mainly due to the influence of vegetation and reservoirs. 
[revised manuscript text omitted]

**Comment 2:** L440-442. Fix this sentence grammatically

**Response:** This part of the content has been replaced by new content.

**Conclusion**

**Comment 1:** L457: Storage capacity decreased (instead of weakened)

**Response:** This problem has been corrected.

L515-520: Moisture and temperature conditions were the key factors that restrict the soil water storage capacity in different vegetation zones. Each vegetation area's soil water storage capacity is: Alpine Meadow > Deciduous Forest > Coniferous Forest > Mountainous Grassland. The water storage capacity of the surface soil in each vegetation zone was weak, and more water was stored in the middle and lower soil with higher clay content.

**Comment 2:** L461: Soil "water" evaporation in spring…

**Response:** This problem has been corrected.

L499-501: Soil water evaporation in spring and summer and insufficient precipitation during the drought period were the main driving forces leading to isotopic enrichment in surface soil.

**Comment 3:** L463: Is it "isotopic" instead of "isotopci"?

**Response:** This problem has been corrected.

L499-501: Soil water evaporation in spring and summer and insufficient precipitation during the drought period were the main driving forces leading to isotopic enrichment in surface soil.

**Comment 4:** L463-465. This sentence needs fixing. I could not understand what it conveys.

**Response:** I have revised this sentence.

L503-506: In the Mountain Grassland and Deciduous Forest zone, drought caused the evaporation signal to penetrate deep into the middle and lower soils. The SWlc-excess below 70 cm of the ground surface was still negative.

---

## Author Response (AR1)

**Revision Notes**

Dear Editor and Reviewers:

Thank you for your letter and for the reviewers' comments concerning our manuscript entitled "Evaporation, infiltration and storage of soil water in different vegetation zones in Qilian mountains: From a perspective of stable isotopes" (Manuscript Number: Hess-2021-376).

According to the editor and reviewers' comments, we have revised our manuscript carefully. The revised portions have been marked in red in the revised version of the manuscript. The main corrections and the response to the reviewers' comments are as follows.

**Responses to the editor and reviewer's comments:**

**Response to Editor**

**Comment 1:** Please, consider all the online comments suggested by the three reviewers, to whom the editorial team expresses its gratitude. The comments are highly pertinent and relevant to improve the quality of your manuscript. Additionally, I ask you to carefully check the figures and tables' captions, as I find many of those to be poorly written (namely: Table 1, Fig. 4, Fig. 5, and Fig. 6).

**Response 1:** We carefully revised and responded to the comments of the three reviewers. Additionally, we double-checked and revised the title of the chart.

**Comment 2:** Authors are responsible for preparing their papers in correct English, so please check and improve language editing.

**Response 2:** Regarding language issues, we have made revisions through a specialized language polishing agency.

[Figure]

**Editing Certificate**

This document certifies that the manuscript

**Evaporation, infiltration and storage of soil water in different vegetation zones in the Qilian mountains: a stable isotope perspective**

prepared by the authors

**Guofeng Zhu, Liyuan Sang, Yuwei Liu, Zhuanxia Zhang, Lei Wang**

was edited for proper English language, grammar, punctuation, spelling, and overall style by one or more of the highly qualified native English speaking editors at AJE.

This certificate was issued on **January 20, 2022** and may be verified on the AJE website using the verification code **0778-A646-3143-E16E-846P**.

[Figure]

Neither the research content nor the authors' intentions were altered in any way during the editing process. Documents receiving this certification should be English-ready for publication; however, the author has the ability to accept or reject our suggestions and changes. To verify the final AJE edited version, please visit our verification page at aje.com/certificate. If you have any questions or concerns about this edited document, please contact AJE at support@aje.com.

AJE provides a range of editing, translation, and manuscript services for researchers and publishers around the world.

**Response to Reviewer #1**

**Reviewer#1: This is an interesting study conducted in a mountain region in China. The research is based on the use of stable isotope in water to understand the main mixing processes and hydrological processes. The manuscript is logically organized and clearly illustrated. However, there are serious issues that in my opinion should prevent the publication of this manuscript in the present form. First of all, the English language is very poor, and sometimes (especially in the Introduction) it's hard to follow the reading. Secondly, the are serious flaws in the Abstract and mostly in the Introduction that fails to reach the focal point of this work (see specific comments below). Third, the discussion is well complemented with literature references but is quite often vague and appears to be not well supported by the observations. No reference to figures and tables are reported in the discussion and it seems that the processes explained by the Authors are based on a previous knowledge of the area and by results taken from the literature than supported by their own results. I suggest to more strictly base the inference on hydrological processes on the observed results.**

**In the end I suggest to resubmit this manuscript after fixing all these major issues.**

Thanks for your comments.

**Specific comments**

**Comment 1:** The title does not read well. I suggest changing into "Evaporation, infiltration and soil storage in different mountain zones" or "Evaporation, infiltration and soil storage in different vegetation zones in a mountain catchment". Perhaps also "Evaporation, infiltration and soil storage in different mountain zones: an isotope perspective". But there is no strict need to stress the isotopic perspective, in my opinion.

**Response 1:** Based on the suggestions of the two reviewers, I have revised the title of the manuscript.

L1-3: Evaporation, infiltration and storage of soil water in different vegetation zones in the Qilian Mountains: A stable isotope perspective

**Comment 2:** The abstract lacks to report the main objective, or the research questions.

**Response 2:** I added the main research objectives and questions in the abstract according to your suggestion.

L13-17: To further understand the process of soil water movement and runoff generation in different vegetation zones (alpine meadow, coniferous forest, mountain grassland, and deciduous forest) in mountainous areas, this study monitored the temporal and spatial dynamics of hydrogen and oxygen stable isotopes in the precipitation and soil water of the Xiying River Basin.

**Comment 3:** The introduction suffers from different weak points and needs a severe restructuring.

1) It is not clear what different vegetation zones are, and what role they play in water exchange in the environment, and why they are important in this research.

2) The text focuses too much on the variability of the isotopic composition in vegetated environments without going deeper into the main physical processes that still need to be understood. Isotopes are just a tool, and the goal here is to understand hydrological processes with the help of isotopes, not which are the factors affecting isotopic variability.

3) Very importantly, no research gaps is put forward. We understand that studying and understanding hydrological processes in different vegetated areas is important but what is the real problem here? As a result, the specific objectives are disconnected from the rest of the

Introduction and fluctuate in their own space. Moreover, what is the memory effect? Why is it important? What is not known about it? How does it fit the general story behind this paper?

**Response 3:** I rewrote the introduction based on your suggestions and other reviewers' comments. The memory effect refers to the indicative profile produced by the stable isotope of soil water in response to the event after precipitation or drought. We highlighted its importance in the manuscript. The memory effect refers to the indicative profile produced by the response of stable isotope of soil water to precipitation or drought. We highlighted its importance in the manuscript.

L31-92: In arid inland river basins, changes in climate and vegetation will affect the hydrological cycle (Tetzlaff et al., 2013; Sharma et al., 2021). As an important part of the water cycle, soil water in the unsaturated zone can be converted from precipitation to stream or groundwater recharge. Determining soil water's evaporation, infiltration, and storage properties is critical for understanding the regional hydrological cycle and water balance under the background of climate and vegetation changes (Brooks et al., 2010; Grant and Dietrich, 2017).

Isotopes, as "fingerprints" of water, have been used to track ecohydrological characteristics, such as evaporation (Barnes and Allison, 1988; Zhu et al., 2021b), groundwater recharge (Koeniger et al., 2016), infiltration paths (Tang and Feng, 2004; Duvert et al., 2016; Zhu et al., 2021a), evapotranspiration distribution (Xiao et al., 2018; Gibson et al., 2021) and water absorption by plants (Rothfuss and Javaux, 2017).

Water seepage in the unsaturated soil zone and the water evaporation at the air–soil interface are the main forms of soil water transport. Seasonal variations in precipitation isotopes are often used to track the process of soil water leakage (Stumpp et al., 2012). During the piston infiltration process, precipitation with different $\delta^2H$ and $\delta^{18}O$ peaks are retained in the soil profile and gradually disappears as the infiltration depth increases (Sprenger et al., 2016a), while the preferential flow will keep these peaks until reaching the deep soil layer (Peralta-Tapia et al., 2015). During a precipitation event, the response of the water isotopes in the surface soil to precipitation is more obvious, changing to nearly that of the stable isotopes of the precipitation. With the deepening of the soil layer, the seasonal variation in precipitation isotope signals rapidly attenuates, and the influence of precipitation on soil water gradually weakens (Sprenger et al., 2017). Evapotranspiration is the main form of soil water dissipation. Because the mass of hydrogen and oxygen atoms that make up water molecules are related to their thermodynamic properties, isotope fractionation of water

will occur in the process of the water cycle. The evaporation of liquid water produces water vapor enriched in $^1$H and $^{16}$O, while the remaining water is enriched in $^2$H and $^{18}$O (Ferretti et al., 2003). Dansgaard (1964) proposed the concept of d-excess (d-excess=$\delta^2$H-8$\delta^{18}$O) to illustrate the intensity of evaporative fractionation. In the state of isotopic equilibrium, the d-excess is 10. Compared with d-excess, lc-excess can better explain the evaporative fractionation process. The main reason is that lc-excess of precipitation and soil water changes smoothly and has relatively small seasonal changes (Landwehr et al., 2014). The dynamic changes in isotopes record the signal of soil water evaporation. This enrichment from dynamic fractionation exists in soil water isotopes in different climatic regions. Compared with temperate regions, the signals of evaporation in arid and Mediterranean environments penetrate deeper into the soil (Sprenger et al., 2016b). Some water is stored in the soil after evaporation and seepage. The water storage capacity of humid areas is higher than that of arid areas, the water storage capacity of forests is higher than that of grasslands, and the water storage capacity of the surface soil layer is lower than that of deeper soil layers with higher clay content (Heinrich et al., 2019; Sprenger et al., 2019; Kleine et al., 2020; Snelgrove et al., 2021).

In alpine mountains, climate warming accelerates the melting of glaciers and frozen soil, and the dynamic interaction between water bodies stored in different media becomes the main influence on the water cycle (Penna et al., 2018). Previous studies on the evaporation, infiltration, and storage of soil water have mostly focused on different climatic regions or vegetation types in the same climatic region. Understanding the climatic and hydrological conditions of different vertical vegetation zones and clarifying the regulating role of vegetation in the water cycle can help better adapt to climate change's influences on the hydrological cycle in source areas. This study monitored the stable isotope composition of precipitation and soil water and the spatiotemporal dynamics of soil water storage in four vegetation zones (alpine meadow, coniferous forest, mountain grassland, and deciduous forest) with different hydrothermal conditions in the Xiying River Basin. To explore the differences in soil water evaporation, infiltration, and storage processes in these four different climates, vegetation types, and terrain types, the following research objectives were proposed: (1) to explore the evolution of isotope evaporation signals and the "memory effects" of precipitation input, mixing and rewetting; and (2) to understand the soil water storage capacity and influencing factors of four vegetation areas in mountainous areas.

**Comment 4:** L349-354. This paragraph and the related Fig. 7 fit much better in the Results than in the discussion. I suggest restructuring this part.

**Response 4:** According to your suggestion, I have adjusted this part of the content to the Results.

L334-355: **4.4 Variations in the water storage capacity of the 0-40 cm soil layer in different vegetation areas**

This study used soil water to calculate the water storage of the 0-40 cm soil layer in the four vegetation zones during the observation period (Fig 7). The water storage capacity of the alpine meadow gradually decreased from April to July (209.7~167.2 mm), and the water storage capacity increased after July (167.2~201.8 mm). The monthly average water storage capacity was the lowest at 0-10 cm (43.0 mm) and the highest at 30-40 cm (51.7 mm). The water storage capacity of the coniferous forest gradually decreased from April to July (150.1~101.2 mm), and the water storage capacity increased after July (101.2~160.0 mm). The monthly average water storage capacity was the lowest at 0-10 cm (28.0 mm) and the highest at 30-40 cm (40.0 mm). The water storage capacity of the mountain grassland gradually decreased from April to July (80.3~64.0 mm), and the water storage capacity increased after July (64.0~104.6 mm). The monthly average water storage capacity was the lowest at 0-10 cm (17.5 mm) and the highest at 20-30 cm (22.0 mm). The water storage capacity of the deciduous forest gradually decreased from April to June (159.3~104.0 mm), the water storage capacity increased from June to August (104.0~154.0 mm), and there was a decrease from August to October (154.0~111.8 mm). The monthly average water storage capacity was the lowest at 0-10 cm (29.1 mm) and the highest at 20-30 cm (35.0 mm). In general, the soil water storage capacity of the 0-10 cm soil layer was less than that of the other soil layers. The order of the water storage capacity of the 0-40 cm soil layer in the four vegetation zones is AM > DF > CF > MG.

**Comment 5:** L365-366. Which are the results that lead the Authors to believe this? Please, explain.

**Response 5:** I reinterpreted this point of view.

L410-413: The soil moisture profile showed a trend of water increasing from top to bottom, indicating the influence of the previous precipitation. The soil was humid, so the replenishment of soil water by precipitation had the characteristics of top-down piston replenishment.

**Comment 6:** L422-424. Again, here we need some experimental evidence about this process. Section 5.3. All this is interesting but it sounds a bit general and vague, these statements look not so related to the results, there are no references to the figures, and the reader has the impression that the Authors present their own preliminary idea that does not reflect the data. I'm happy to be mistaken here but we need to have evidence of all the described processes. Moreover, the title does not reflect the content of the section.

**Response 6:** We logically sorted out the full text based on the reviewers' comments. The discussion on runoff generation in the watershed does not match the theme of this manuscript (Evaporation, infiltration and storage of soil water in different vegetation zones in the Qilian mountains: a stable isotope perspective). Therefore, the manuscript focused on soil moisture's evaporation, infiltration, and storage mechanism in the study area. Based on this, we reorganized this part of the content.

[revised manuscript text omitted]

**Minor comments and technical corrections**

**Comment 7:** L58-59. D-excess is introduced here without any explanation on its formulation and physical meaning.

**Response 7:** "d-excess" has been introduced in detail.

L60-62: Dansgaard (1964) proposed the concept of d-excess (d-excess=$\delta^2$H-8$\delta^{18}$O) to illustrate the intensity of evaporative fractionation. In the state of isotopic equilibrium, the d-excess is 10.

**Comment 8:** L96. Is this the long-term average runoff? Please, specify.

**Response 8:** This is the average runoff over the years, which has been clarified.

L98-100: The average annual runoff of the Xiying River is 388 million m3, which is mainly replenished by mountain precipitation and melting water of ice and snow.

**Comment 9:** Table 1. Are the meteorological parameters averages? Over which period of time? Please, specify.

**Response 9:** Based on your suggestion, I have changed and explained them here.

L137-139: **Table 1** Basic data of each Vegetation zone from April to October 2017 (*Long*-Longitude, *Lat*-Latitude, *Alt*-Altitude, *T*-Air Temperature (daily mean temperature), *P*-Precipitation (total precipitation during the observation period), *h*-Relative Humidity (daily mean relative humidity))

**Comment 10:** Section 3 (Methods). No explanations about the determination of the gravimetric water content are given. Please fix this issue.

**Response 10:** Based on your suggestion, I have added the determination of the gravimetric water content.

L184-187: *W* is the gravimetric water content, which is expressed by the following formula:

$$W = \frac{M_1 - M_2}{M_2} \times 100\% \tag{5}$$

in the formula, $M_1$ is the gravimetric value of wet soil (g), and $M_2$ is the gravimetric value of dry soil (g).

**Comment 11:** L140-142. I suggest including a reference to the correction of the memory effect (e.g., Penna et al., 2012 and/or Qu et al., 2019).

Penna, D., Stenni, B., Šanda, M., Wrede, S., Bogaard, T. A., Michelini, M., Fischer, B. M. C., Gobbi, A., Mantese, N., Zuecco, G., Borga, M., Bonazza, M., Sobotková, M., Ä˝ejková, B., and Wassenaar, L. I.: Technical Note: Evaluation of between-sample memory effects in the analysis of $\delta^2$H and $\delta^{18}$O of water samples measured by laser spectroscopes, Hydrol. Earth Syst. Sci., 16, 3925–3933, https://doi.org/10.5194/hess-16-3925-2012, 2012.

Qu, D, Tian, L, Zhao, H, Yao, P, Xu, B, Cui, J. Demonstration of a memory calibration method in water isotope measurement by laser spectroscopy. Rapid Commun Mass Spectrom. 2020; 34:e8689. https://doi.org/10.1002/rcm.8689

**Response 11:** I have added the necessary references here.

L147-149: To avoid the "memory effect" of isotope analysis, we discarded the first two injection values and used the average value of the last four injections as the final result (Penna et al., 2012; Qu et al., 2020).

**Comment 12:** L169. Please explain the role of 10 in the equation.

**Response 12:** Here, we calculate the water storage capacity of the 10cm soil layer, so we multiply it by 10.

**Comment 13:** L176. Add "2017".

**Response 13:** This problem has been corrected.

L191-194: During the study period (April-October 2017), in the Xiying River Basin, the potential evapotranspiration was 872.8 mm, the daily evapotranspiration ranged from 7.5 mm (July 14) to 0.9 mm (October 9), showing a fluctuating trend around July, and the PET value in April-July was higher than that in August-October.

**Comment 14:** L194: Precipitation events?

**Response 14:** This problem has been corrected.

L208-210: During the observation period, there were 72 precipitation events in the alpine meadow zone, and the total precipitation was 534.3 mm, which was relatively evenly distributed each month.

**Comment 15:** L213-220. This part can be reported in a Table or in a boxplot.

**Response 15:** Based on your suggestion, we have drawn a table.

L246-247: **Table 2** General characteristics of precipitation $\delta^2H$ and $\delta^{18}O$ in different vegetation areas from April to October 2017

| Vegetation zone | $\delta^2H$/‰ | | | | $\delta^{18}O$/‰ | | | |
|---|---|---|---|---|---|---|---|---|
| | Max | Min | mean | SD | Max | Min | mean | SD |
| AM | 13.7 | -163.9 | -73.1 | 36.3 | -1.3 | -23.1 | -10.0 | 4.3 |
| CF | 13.0 | -117.8 | -42.0 | 37.2 | -0.1 | -17.4 | -7.1 | 4.7 |
| MG | 4.2 | -103.1 | -37.4 | 30.5 | -0.9 | -15.1 | -5.9 | 3.9 |
| DF | 23.2 | -110.2 | -31.8 | 42.8 | 3.2 | -15.2 | -5.8 | 5.5 |

**Comment 16:** Fig. 3. Use different colours to distinguish between the different variables. Particularly, I suggest using different closures for the two isotopes and then keep them in all the other graphs.

**Response 16:** According to your suggestion, I modified Figure 3.

L248-251:

[Figure]

**Fig. 3** Time series of rainfall and isotope characteristics in different vegetation zones in Xiying River Basin, with dotted lines indicating the date of soil water sampling

**Comment 17:** L263? Does "deep" mean "grey"? In that case use the correct term.

**Response 17:** This problem has been corrected.

L290-292:

**Fig. 4** Heat map of the soil depth profile of $\delta^2H$, $\delta^{18}O$, lc-excess and GWC in different vegetation zones, and the layer lacking measurement is indicated by grey color

**Comment 18:** L266. The diagram is normally called "dual isotope space".

**Response 18:** This problem has been corrected.

L294-295: Isotope data of precipitation and soil water obtained from different vegetation zones are shown in dual-isotope space in Fig. 5.

**Comment 19:** L272-274. This part can/could be moved to the discussion.

**Response 19:** We have moved this part to the discussion according to your suggestion.

L363-368: Due to a low atmospheric temperature, low cloud base height, and low air-saturated water vapor loss, the alpine meadow zone was weakly affected by secondary evaporation during precipitation. There, the slope of the LMWL (8.4) was even higher than that of the GMWL (Zhang et al., 2012). As the altitude decreased, the secondary evaporation under clouds strengthened, and the slope of the LMWL of each vegetation zone decreased (Pang et al., 2011) (Fig 5).

**Comment 20:** L311. As far as I understand the plot does not show the "differences" but the raw values. Please, correct.

**Response 20:** This problem has been corrected.

L332-333:

**Fig. 6** the variation of $\delta^2H$, $\delta^{18}O$, lc-excess and GWC in different vegetation zones in each sampling

**Comment 21:** L385-386. Add a reference to a figure to corroborate the statement.

**Response 21:** This problem has been corrected.

**Comment 22:** L409. I think the correct citation is Amin et al., 2020.

**Response 22:** This problem has been corrected.

**Comment 23:** L463. Typo.

**Response 23:** This problem has been corrected.

**Response to Reviewer #2**

**Reviewer#2: This a potentially interesting paper, but one that needs major attention before it is suitable for publication. The paper is poorly written in places. While I have sympathy with authors having to write in a second language, which is something that I cannot do, some sections of the paper are very difficult to follow. More importantly, the sections of the paper are not well linked. It is not clear from the Introduction how the paper addresses the important issues. The same can be said about the Discussion where it is not clear how the data in this paper inform the issues being discussed. For example, runoff generation is mentioned in the introduction and appears in the general conclusions, but there is no discussion as to how the data in the paper help us understand it (there are several similar examples as well). The sections describing the data tend to be very generalised and the data description needs to be more informative. Moreover, the data need to be presented in the paper or as a supplement.**

**Overall, the paper needs to be rewritten so that the data are discussed in a more rigorous manner that help understand the aims. I am not convinced that it actually addresses important issues or that the aims of understanding the memory effect or runoff generation are advanced by this study.**

Thanks for your comments.

**Specific comments**

**Title**

**Comment 1:** Having a title that is grammatically incorrect is not a good way to promote your research. Something like: "Evaporation, infiltration and storage of soil water in different vegetation zones in the Qilian mountains: a stable isotope perspective" would be better.

**Response:** Based on the suggestions of the two reviewers, I have revised the title of the manuscript.

L1-3: Evaporation, infiltration and storage of soil water in different vegetation zones in the Qilian Mountains: A stable isotope perspective

**Abstract**

**The abstract needs improvement. Abstracts are important as they are what convince the readers to look at the rest of the paper. They should convey not only what has been**

**studied and why, but should also contain enough details so that the main conclusions are evident. This abstract needs improving, specifically:**

**Comment 1:** Be specific: "different water bodies in different vegetation zones" does not convey what you have done.

**Response:** I have explained this point in detail.

L13-17: To further understand the process of soil water movement and runoff generation in different vegetation zones (alpine meadow, coniferous forest, mountain grassland, and deciduous forest) in mountainous areas, this study monitored the temporal and spatial dynamics of hydrogen and oxygen stable isotopes in the precipitation and soil water of the Xiying River Basin.

**Comment 2:** Avoid qualitative terms such as "weak"

**Response:** The article carried out a more quantitative expression, such as "The evaporation intensity of the four vegetation zones was: Mountain Grassland > Deciduous Forest > Coniferous Forest > Alpine Meadow.".

**Comment 3:** Some of the sentences are unclear. I am not sure what "The water storage capacity of surface soil was weak in vegetation zones" really means as surely all the catchment is vegetated?)

**Response:** In our results, the soil water storage capacity of 0-10 cm is less than that of other soil layers, and we have clarified the soil depth here. Here, we want to show the water storage of different soil layers. In our sampling site, the soil is covered by dominant species.

L24-27: Each vegetation zone's soil water storage capacity followed the order of alpine meadow > deciduous forest > coniferous forest > mountain grassland. In addition, the water storage capacity of shallow soils in different types of vegetation areas was weaker than that of deep soils.

**Comment 4:** There are several grammatical and spelling errors (Nvertheless) that detract from the work

**Response:** We carefully checked and revised the grammar and spelling of the manuscript.

**Introduction**

**The introduction is also not very clear. Some of this reflects the writing style and occasional poor grammar. As well, there needs to be a much clearer explanation of the background. The explanations are vague and would not convey much meaning to anyone not working in the field. There needs to be clearer explanations and more precise terminology.**

**Comment 1:** L31-33. Not very clear what you mean here.

**Response:** For readers to better understand, I re-narrate this sentence.

L31-34: In arid inland river basins, changes in climate and vegetation will affect the hydrological cycle (Tetzlaff et al., 2013; Sharma et al., 2021). As an important part of the water cycle, soil water in the unsaturated zone can be converted from precipitation to stream or groundwater recharge.

**Comment 2:** L48. "Storage" is not a transport mechanism.

**Response:** I agree with your comment, this problem has been corrected.

L44-45: Water seepage in the unsaturated soil zone and the water evaporation at the air－soil interface are the main forms of soil water transport.

**Comment 3:** L50. Do you mean on the ground surface or in the near-surface part of the soils?

**Response:** We rewrite this sentence.

L58-60: The evaporation of liquid water produces water vapor enriched in $^1H$ and $^{16}O$, while the remaining water is enriched in $^2H$ and $^{18}O$ (Ferretti et al., 2003).

**Comment 4:** L48-75. This would not be readily understandable to many readers who had not worked with these types of data. It is too generally worded and needs details. This paragraph is important as it sets the framework for using the stable isotopes to understand processes.

(1) Define that you are discussing $^{18}O$ and $^2H$ data (there are lots of stable isotopes!).

(2) Terms such as "makes soil water isotopes enriched" are vague. Specifically, evaporation enriches the residual water in $^{18}O$ or $^2H$ (or increases the $\delta^{18}O$ and $\delta^2H$ values)

(3) Likewise, "soil moisture fractionation is positively correlated with evapotranspiration but negatively correlated with precipitation". Are you talking about the magnitude or sign?

(4) How significant?

(5) Define the d-excess (briefly)

(6) L63-70. Lacks detail and is unclear.

**Response:** According to the suggestions of three reviewers, this part was rewritten to solve the above problems: (1) We identified stable isotopes of hydrogen and oxygen; (2) According to your suggestion, the expression has been changed; (3) and (4) According to the reviewer's suggestion, we introduced the evaporation process more and deleted the influencing factors of evaporation; (5) We defined "d-excess"; (6) I gave a detailed description of this part to make it more expressive of the status quo of the research.

L44-92: Water seepage in the unsaturated soil zone and the water evaporation at the air–soil interface are the main forms of soil water transport. Seasonal variations in precipitation isotopes are often used to track the process of soil water leakage (Stumpp et al., 2012). During the piston infiltration process, precipitation with different $\delta^2H$ and $\delta^{18}O$ peaks are retained in the soil profile and gradually disappears as the infiltration depth increases (Sprenger et al., 2016a), while the preferential flow will keep these peaks until reaching the deep soil layer (Peralta-Tapia et al., 2015). During a precipitation event, the response of the water isotopes in the surface soil to precipitation is more obvious, changing to nearly that of the stable isotopes of the precipitation. With the deepening of the soil layer, the seasonal variation in precipitation isotope signals rapidly attenuates, and the influence of precipitation on soil water gradually weakens (Sprenger et al., 2017). Evapotranspiration is the main form of soil water dissipation. Because the mass of hydrogen and oxygen atoms that make up water molecules are related to their thermodynamic properties, isotope fractionation of water will occur in the process of the water cycle. The evaporation of liquid water produces water vapor enriched in $^1H$ and $^{16}O$, while the remaining water is enriched in $^2H$ and $^{18}O$ (Ferretti et al., 2003). Dansgaard (1964) proposed the concept of d-excess (d-excess=$\delta^2H$-8$\delta^{18}O$) to illustrate the intensity of evaporative fractionation. In the state of isotopic equilibrium, the d-excess is 10. Compared with d-excess, lc-excess can better explain the evaporative fractionation process. The main reason is that lc-excess of precipitation and soil water changes smoothly and has relatively small seasonal changes (Landwehr et al., 2014). The dynamic changes in isotopes record the signal of soil water evaporation. This enrichment from dynamic fractionation exists in soil water isotopes in different climatic regions. Compared with temperate regions, the signals of evaporation in arid and Mediterranean environments penetrate deeper into the soil (Sprenger et al., 2016b). Some water is stored in the soil after evaporation and seepage. The water storage capacity of humid areas is higher than that of arid areas, the water storage capacity of forests is higher than that of grasslands, and the water storage capacity of the surface soil layer is lower than that of deeper soil layers

with higher clay content (Heinrich et al., 2019; Sprenger et al., 2019; Kleine et al., 2020; Snelgrove et al., 2021).

In alpine mountains, climate warming accelerates the melting of glaciers and frozen soil, and the dynamic interaction between water bodies stored in different media becomes the main influence on the water cycle (Penna et al., 2018). Previous studies on the evaporation, infiltration, and storage of soil water have mostly focused on different climatic regions or vegetation types in the same climatic region. Understanding the climatic and hydrological conditions of different vertical vegetation zones and clarifying the regulating role of vegetation in the water cycle can help better adapt to climate change's influences on the hydrological cycle in source areas. This study monitored the stable isotope composition of precipitation and soil water and the spatiotemporal dynamics of soil water storage in four vegetation zones (alpine meadow, coniferous forest, mountain grassland, and deciduous forest) with different hydrothermal conditions in the Xiying River Basin. To explore the differences in soil water evaporation, infiltration, and storage processes in these four different climates, vegetation types, and terrain types, the following research objectives were proposed: (1) to explore the evolution of isotope evaporation signals and the "memory effects" of precipitation input, mixing and rewetting; and (2) to understand the soil water storage capacity and influencing factors of four vegetation areas in mountainous areas.

**Comment 5:** L76-78. Not clear what you mean by this. Are the water resources more unstable or are they transitioning?

**Response:** I have reinterpreted this sentence for the sake of understanding.

L75-77: In alpine mountains, climate warming accelerates the melting of glaciers and frozen soil, and the dynamic interaction between water bodies stored in different media becomes the main influence on the water cycle (Penna et al., 2018).

**Comment 6:** L84. "Heat conditions" do you mean temperatures?

**Response:** We want to express the vegetation zone under different moisture and temperature conditions. Based on this, I re-narrate this sentence.

L83-86: This study monitored the stable isotope composition of precipitation and soil water and the spatiotemporal dynamics of soil water storage in four vegetation zones (alpine meadow, coniferous forest, mountain grassland, and deciduous forest) with different hydrothermal conditions in the Xiying River Basin.

**Comment 7:** L82-90. These are fine as general aims, but can you explain (briefly) why this is important (i.e. what are you doing that is new, what are the broader implications?). There is a disconnect here between the broad general themes in the rest of the introduction and your specific study. Also, runoff generation and the memory effect are not explicitly discussed in any depth in the paper (need to make sure that your aims are actually what you discuss).

**Response:** Previous studies on soil moisture evaporation, infiltration and storage have mostly focused on different climatic regions or vegetation types under the same climatic region, and there are few uses of isotope technology to trace the hydrological processes in the mountain vegetation vertical zone.

**Comment 8:** L88. If it is important, define the memory effect and explain why we need to understand it.

**Response:** The "memory effect" means that the temporal and spatial changes of the stable isotope profile of soil water can reflect and characterize the input, mixing, and rewetting process of precipitation. Understanding the "memory effect" helps us trace the dynamic changes of climate and soil hydrology.

**Study Area**

**This section needs referencing. Also a few more details as to the spatial variation of rainfall and temperature (I presume that the highest rainfall and lowest mean temperatures are in the mountains?)**

**Response:** I added a reference. Table 1 shows the spatial variation of precipitation and temperature.

L137-139: **Table 1** Basic data of each Vegetation zone from April to October 2017 (*Long*-Longitude, *Lat*-Latitude, *Alt*-Altitude, *T*-Air Temperature (daily mean temperature), *P*-Precipitation (total precipitation during the observation period), *h*-Relative Humidity (daily mean relative humidity))

| Vegetation zone | Geographical parameter | | | Meteorological parametes | | | Number of samples | |
|---|---|---|---|---|---|---|---|---|
| | *Long*(°E) | *Lat*(°N) | *Alt*(m) | *T*(℃) | *P*(mm) | *h*(%) | Precipitation | Soil |
| Alpine Meadow | 101°51'16" | 37°33'28" | 3637 | -0.19 | 595.1 | 69.2 | 72 | 47 |
| Coniferous Forest | 101°53'23" | 37°41'50" | 2721 | 3.34 | 431.9 | 66.6 | 42 | 41 |
| Mountain Grassland | 102°00'25" | 37°50'23" | 2390 | 6.6 | 363.5 | 60.4 | 37 | 54 |
| Deciduous Forest | 102°10'56" | 37°53'27" | 2097 | 7.9 | 262.5 | 59.8 | 40 | 53 |

**Comment 1:** L94. What is a "first-class tributary"?

**Response:** I changed the description of this sentence.

L95-98: As the largest tributary of the Shiyang River, it is formed by the Shuiguan River, Ningchang River, Xiangshui River, and Tatu River converging from southwest to northeast and ultimately flows into the Xiying Reservoir.

**Comment 2:** L98-101. Probably worth reporting the Koppen Zones.

**Response:** We used the Köppen climate zone.

L101-110: The basin elevation is between 2000 m and 5000 m, corresponding to a temperate semiarid climate with strong solar radiation, a long sunshine time, and a large temperature difference between day and night. The average annual temperature of the basin is 6°C, the annual average evaporation is 1133 mm, the annual average precipitation is 400 mm, and the precipitation from June to September accounts for 69% of the annual precipitation. Precipitation increases with elevation, while temperature decreases with elevation in this area, (Table 1) (Ma et al., 2018). The zonal differentiation of vegetation in the basin is dominated by deciduous forest, mountain grassland, cold temperate coniferous forest, and alpine meadow. The soils mainly include lime, chestnut, alpine shrub meadow, and desert soil (Fig. 1).

**Comment 3:** L103-106. Refer to Fig. 1.

**Response:** Figure 1 has been referenced here.

L107-110: The zonal differentiation of vegetation in the basin is dominated by deciduous forest, mountain grassland, cold temperate coniferous forest, and alpine meadow. The soils mainly include lime, chestnut, alpine shrub meadow, and desert soil (Fig. 1).

**Comment 4:** Fig. 1. What is the inset on the left-hand map?

**Response:** The complete nine-dotted line is shown here.

**Data and Methods**

**The methods used here are standard and suitable for the project. As with much of the rest of the paper, there are a few details lacking and the explanations are not very clear.**

**Comment 1:** L111-113. It would be helpful here or in Section 2 to outline what 2017 was like in terms of rainfall and temperature as these vary year-by-year. In particular, distinguish between long-term averages and values in the sampling year.

**Response:** I have outlined precipitation and temperature.

L117-121: In 2017, the precipitation in the alpine meadows, coniferous forests, mountain grasslands and deciduous forests was 595.1 mm, 431.9 mm, 363.5 mm and 262.5 mm, respectively. The average daily temperatures in the alpine meadows, coniferous forests, mountain grasslands and deciduous forests were -0.19°C, 3.34°C, 6.6°C and 7.9°C, respectively (Table 1).

**Comment 2:** L116-122. This is rather a clunky description (not sure that you need to specify explicitly that you wrote dates on bottles). What do you mean by "four parallel samples" being also collected?

**Response:** I re-narrate the soil sampling process.

L124-128: Three duplicate samples were collected for each soil layer. A collected soil sample was placed into a 50 mL glass bottle, and the bottle mouth was sealed with Parafilm marked with the sampling date; the sample was then frozen for storage until experimental analysis. Each sample was collected separately in an aluminum box.

**Comment 3:** L143. It's "permil" not "thousands". As written, "thousandths of the Vienna Standard Mean Ocean Water (VSMOW)" is meaningless.

**Response:** This error has been corrected.

L149-150: The analysis results were relative to VSMOW (Vienna Standard Mean Ocean Water):

**Comment 4:** Section 3.2. The analysis is only part of the uncertainty. Did you perform multiple extractions on the same sample to test the uncertainty associated with that. This will undoubtedly be higher and needs to be considered.

**Response:** During the sampling process, we collected duplicate samples to improve the accuracy of the experiment.

**Comment 5:** Section 3.3.1 The line-conditioned excess is less used than the d-excess (but is potentially more informative). You should explain what it is (and define the term). The explanation "The physical meaning of lc-excess is expressed as the deviation degree between isotopic values in samples and LMWL, which indicates the non-equilibrium dynamic fractionation process caused by evaporation (Landwehr et al., 2014; Sprenger et al., 2017)" is not very clear.

**Response:** Based on your suggestion, we described lc-excess in detail.

L156-170: The linear relationship between δ²H and δ¹⁸O in precipitation and soil water is defined as the LMWL (local meteoric water line) and SWL (soil waterline), respectively, are of great significance for studying the evaporative fractionation of stable isotopes during the water cycle. We further calculated the line-conditioned excess for each soil water and precipitation sample. The lc-excess in different water bodies can characterize the evaporation index of different water bodies relative to the local precipitation (Landwehr and Coplen, 2006).

$$lc - excess = \delta^2 H - a \times \delta^2 H - b \tag{2}$$

where $a$ and $b$ are the slope and intercept of the LMWL, respectively, and δ²H and δ¹⁸O are the isotopic values of hydrogen and oxygen in the sample. The physical meaning of lc-excess is expressed as the degree of deviation of the isotope value in the sample from the LMWL, indicating the nonequilibrium dynamic fractionation process caused by evaporation. Generally, the change in lc-excess in local precipitation is mainly affected by different water vapor sources, and the annual average is 0. Since the stable isotopes in soil water are enriched by evaporation, the average lc-excess is usually negative (Landwehr et al., 2014; Sprenger et al., 2017).

**Comment 6:** Section 3.3.2. More details are needed as to where these data are derived from. Are they local data measured at the field site or interpolated estimates? Application of the Penman-Monteath equation is very data sensitive. What do you think the errors are here?

**Response:** These data are from nearby weather stations. According to the literature, we revised the formula.

L172-179: The potential evapotranspiration was calculated based on the Penman-Monteath equation (Allen et al., 1998):

$$PET = \frac{0.408\Delta(R_n - G) + \gamma \dfrac{900}{T + 273} u^2(e_s - e_a)}{\Delta + \gamma(1 + 0.34u^2)} \tag{3}$$

where PET is the daily potential evapotranspiration (mm day$^{-1}$), $R_n$ is the net radiation (MJ m$^2$ day$^{-1}$), $G$ is the soil heat flux density (MJ m$^2$ day$^{-1}$), $\gamma$ is the psychrometric constant (kPa°C$^{-1}$), $u_2$ is the wind speed at 2 m height (m s$^{-1}$), $T$ is the mean daily air temperature at 2 m height

(°C), Δ is the slope of the vapor pressure curve (kPa°C⁻¹), $e_a$ is the actual vapor pressure (kPa) and $e_s$ is the saturated vapor pressure (kPa). These data come from nearby weather stations.

**Comment 7:** Section 3.3.3. These are based on your measurements, yes? Again, do you have estimates of uncertainties. Also, some of the techniques (eg moisture content) need more detail.

**Response:** This data comes from our actual measurement, which is constant, and we added the calculation of soil moisture.

L184-187: $W$ is the gravimetric water content, which is expressed by the following formula:

$$W = \frac{M_1 - M_2}{M_2} \times 100\% \qquad (5)$$

in the formula, $M_1$ is the gravimetric value of wet soil (g), and $M_2$ is the gravimetric value of dry soil (g).

**Results**

**This section suffers from the shortcomings of the rest of the paper. The explanations are not very clear and are often overly general. Also, I cannot see where the raw data are (no Table or Appendix); presenting the actual data is required.**

**Comment 1:** L175-178. How precise are these values (i.e. is the 1dp precision warranted)? What was the rainfall during those times?

**Response:** We use FAO Penman-Monteith to calculate the potential daily evapotranspiration (possible evapotranspiration) in the study area (the software calculation results are kept to three decimal places, and we keep one decimal place in the study). This    illustrates the date when the maximum and minimum values appear, and there may be no rainfall on that day.

**Comment 2:** L180-182. Not very clearly worded.

**Response:** I restated this sentence.

L194-196: The input of summer precipitation and ice/snow meltwater increased runoff, resulting in a trend similar to PET.

**Comment 3:** L213-220. The ranges in stable isotope values are probably more useful. Suggest that you report the range and the mean (you can omit the SD as that is less useful). Also report the number of observations, so we get an idea of how much data there is. Ideally the mean should be weighted by precipitation amount (it is not clear that that is the case, but you should be clear whether it is).

**Response:** We describe the average value and variation range of the isotope, preserve the SD, and add the number of observations according to the recommendations. The average amount of isotopes is based on the weighted average of precipitation or soil water content.

L228-237: The mean values of $\delta^2H$ and $\delta^{18}O$ in the alpine meadow zone (number of samples: 72) were -73.1‰±36.3‰ (-163.9~13.7‰) and -10.0‰±4.3‰ (-23.1~-1.3‰), respectively. The average $\delta^2H$ and $\delta^{18}O$ values of the coniferous forest zone (number of samples: 42) were -42.0‰±37.2‰ (-117.8~13.0‰) and -7.1‰±4.7‰ (-17.4~-0.1‰), respectively. The average $\delta^2H$ and $\delta^{18}O$ values of the mountain grassland zone (number of samples: 37) were -37.4‰±30.5‰ (-103.1~4.2‰) and -5.9‰±3.9‰ (-15.1~-0.9‰), respectively. The average $\delta^2H$ and $\delta^{18}O$ values of the deciduous forest zone (number of samples: 40) were -31.8‰±42.8‰ (-110.2~23.2‰) and -5.8‰±5.5‰ (-15.2~3.2‰), respectively (Table 2).

**Comment 4:** L220-224. Poorly worded.

**Response:** I restated this sentence.

L237-242: The maximum isotopic values of the four vegetation zones appeared on August 4 (AM: 13.7‰, $\delta^2H$; -1.3‰, $\delta^{18}O$), August 10 (CF: 13.0‰, $\delta^2H$; -0.1‰, $\delta^{18}O$), August 7 (MG: 4.2‰, $\delta^2H$; -0.9‰, $\delta^{18}O$) and August 13 (DF: 23.2‰, $\delta^2H$; 3.2‰, $\delta^{18}O$). The highest temperature in each vegetation zone appeared on July 27. The high temperature caused the precipitation to undergo strong below-cloud evaporation during the fall, leading to the enrichment of isotopes.

**Comment 5:** L224-227. This isn't really that obvious from Fig. 3. Can you report the magnitudes in the text?

**Response:** Here, we have shown the seasonal variation trend of precipitation isotope and added the maximum value of each vegetation zone and the time of its appearance in the previous analysis.

L237-242: The maximum isotopic values of the four vegetation zones appeared on August 4 (AM: 13.7‰, $\delta^2H$; -1.3‰, $\delta^{18}O$), August 10 (CF: 13.0‰, $\delta^2H$; -0.1‰, $\delta^{18}O$), August 7 (MG: 4.2‰, $\delta^2H$; -0.9‰, $\delta^{18}O$) and August 13 (DF: 23.2‰, $\delta^2H$; 3.2‰, $\delta^{18}O$). The

highest temperature in each vegetation zone appeared on July 27. The high temperature caused the precipitation to undergo strong below-cloud evaporation during the fall, leading to the enrichment of isotopes.

**Comment 6:** L232-260. This section has too little detail in it to follow. You need to explain the data more specifically (avoid vague terms such as "depletion" or "enrichment" and report some values). More importantly where are the data? Fig. 4 is labelled as a "heat map" but seems to be the values (I think) and they are on Fig. 5. However, these also need to be in a Table somewhere.

**Response:** According to your suggestion, I plot the data into a table and use the data for analysis.

L253-288: The low-temperature environment and abundant precipitation events in the alpine meadows make the monthly average $\delta^2$H and $\delta^{18}$O of soil water more depleted than other vegetation zones (-69.4~-51.6‰, $\delta^2$H; -7.5~-10.3‰, $\delta^2$H). Despite this, the SWlc-excess of most samples at this station was still negative, and there were different degrees of evaporation in the process of precipitation penetrating the soil and mixing with original pore water, among which evaporation fractionation was stronger in July (-11.9‰ lc-excess) and October (-14.5‰ lc-excess). The soil water isotopes of the coniferous forests gradually changed seasonally. From April to July, precipitation was scarce, the temperature rose, and the isotopes of soil water were gradually enriched on the surface (-52.7~-29.5‰, $\delta^2$H; -7.0~-2.1‰, $\delta^2$H), reaching the peak value of the observation period in July (-29.5‰, $\delta^2$H; -2.1‰, $\delta^{18}$O), and continuous rainfall input from late July to mid-August resulted in soil water isotope depletion (-57.0‰, $\delta^2$H; -8.1‰, $\delta^{18}$O). SWlc-excess was an obvious fractionation signal opposite to the trend of isotope change, reaching the lowest value (-26.3‰) in the sampling period in July, and the change in air temperature and precipitation controlled the evaporation intensity. From April to July, the isotopic value of surface soil water in the mountain grasslands was higher ($\delta^{18}$O was greater than zero), and SWlc-excess was lower than -30‰. During this period, the evaporation and fractionation of shallow soil water were intense. Similar to in the coniferous forests, in the mountain grasslands, the input of heavy precipitation from late July to mid-August led to the depletion of soil water isotopes. There was only sporadic rainfall in the deciduous forests from April to July, and the soil water isotopes were gradually enriched on the surface (-46.1~-18.2‰, $\delta^2$H; -4.7~0.2‰, $\delta^2$H), reached a peak in June when there was no rainfall event (-18.2‰, $\delta^2$H; 0.2‰, $\delta^{18}$O), and then became depleted (-53.2‰, $\delta^2$H; -5.2‰, $\delta^{18}$O). In addition, due to the influence of the

Xiying Reservoir and vegetation coverage, the isotopic enrichment degree of soil water in this vegetation zone was lower than that in the mountain grasslands. As the most intuitive form of water change, the gravimetric water content was always at a low value in July (AM: 21.0; CF: 14.8; MG: 11.9; DF: 14.9), when the evaporation was the strongest, and it was most obvious in shallow soil (Table 3) (Fig. 4).

**Table 3** General characteristics of soil water $\delta^2H$, $\delta^{18}O$, lc-excess and GWC in different vegetation areas from April to October 2017

| Month | Vegetation zone | $\delta^2H$/‰ | | | $\delta^{18}O$/‰ | | | lc-excess/‰ | | | GWC/% | | |
|---|---|---|---|---|---|---|---|---|---|---|---|---|---|
| | | Max | Min | Mean | Max | Min | Mean | Max | Min | Mean | Max | Min | Mean |
| 4 | AM | -55.2 | -70.7 | -65.6 | -8.5 | -10.8 | -10.1 | -2.7 | -7.1 | -4.7 | 25.9 | 23.0.0. | 24.7 |
| | CF | -52.7 | -72.2 | -63.9 | -7.0 | -9.9 | -8.9 | -4.0 | -12.0 | -8.4 | 27.6 | 14.9 | 20.0 |
| | MG | -7.32 | -50.6 | -41.0 | 2.8 | -5.8 | -3.9 | -8.8 | -36.8 | -19.4 | 21.7 | 6.5 | 14.7 |
| | DF | -46.1 | -69.4 | -62.1 | -4.7 | -9.9 | -8.5 | -2.5 | -23.2 | -9.7 | 27.7 | 19.4 | 21.8 |
| 5 | AM | -46.1 | -76.5 | -66.4 | -7.4 | -12.2 | -10.1 | -2.6 | -7.7 | -4.9 | 32.6 | 23.2 | 28.9 |
| | CF | -45.8 | -61.9 | -53.5 | -5.3 | -8.4 | -7.0 | -9.3 | -17.7 | -13.0 | 22.6 | 9.0 | 16.1 |
| | MG | -6.7 | -47.3 | -39.2 | 2.9 | -6.5 | -4.3 | -4.5 | -36.2 | -14.4 | 15.7 | 7.6 | 11.2 |
| | DF | -30.8 | -63.5 | -53.8 | -1.9 | -9.4 | -6.9 | -3.2 | -30.1 | -13.6 | 26.0 | 11.7 | 17.7 |
| 6 | AM | -62.5 | -83.9 | -69.4 | -8.9 | -12.6 | -10.3 | -1.5 | -8.4 | -5.8 | 33.3 | 21.9 | 26.0 |
| | CF | -45.8 | -78.4 | -58.7 | -5.1 | -12.0 | -7.8 | 5.5 | -26.6 | -8.5 | 32.1 | 10.0 | 21.3 |
| | MG | -19.7 | -74.9 | -46.9 | 0.8 | -11.8 | -5.8 | 13.0 | -33.7 | -11.0 | 19.3 | 7.5 | 14.2 |
| | DF | -18.2 | -64.9 | -51.7 | 0.2 | -9.0 | -5.9 | -4.6 | -38.2 | -19.4 | 13.5 | 8.4 | 11.1 |
| 7 | AM | -47.3 | -60.1 | -51.6 | -6.9 | -8.4 | -7.5 | -8.8 | -14.8 | -11.9 | 25.4 | 19.0 | 21.0 |
| | CF | -29.5 | -51.4 | -41.6 | -2.1 | -7.9 | -5.6 | -2.6 | -26.3 | -11.2 | 24.3 | 7.2 | 14.8 |
| | MG | -10.6 | -48.4 | -39.2 | 2.3 | -6.4 | -4.1 | -5.8 | -35.8 | -16.1 | 18.7 | 6.3 | 11.9 |
| | DF | -35.1 | -69.0 | -54.1 | -1.7 | -8.7 | -5.5 | -14.8 | -35.3 | -24.5 | 18.2 | 11.8 | 14.4 |
| 8 | AM | -58.5 | -80.3 | -66.6 | -8.4 | -11.6 | -9.6 | -6.1 | -15.4 | -9.7 | 28.1 | 19.5 | 25.1 |
| | CF | -57.0 | -75.5 | -66.4 | -8.1 | -9.8 | -9.2 | -2.5 | -13.1 | -8.3 | 21.4 | 8.7 | 16.3 |
| | MG | -34.2 | -53.8 | -44.0 | -3.2 | -5.5 | -4.4 | -14.7 | -22.6 | -18.7 | 11.3 | 9.5 | 10.4 |
| | DF | -53.2 | -84.3 | -67.6 | -5.2 | -13.5 | -9.2 | 6.8 | -26.1 | -9.6 | 23.6 | 14.7 | 20.6 |
| 9 | AM | -48.0 | -79.2 | -61.0 | -7.8 | -11.1 | -9.2 | -4.3 | -10.4 | -7.2 | 29.9 | 20.3 | 25.3 |
| | CF | -52.5 | -67.7 | -60.7 | -7.8 | -10.1 | -8.8 | -0.1 | -11.3 | -6.0 | 31.3 | 9.3 | 20.5 |
| | MG | -32.3 | -45.3 | -38.8 | -3.5 | -4.4 | -4.0 | -9.1 | -23.8 | -16.5 | 15.3 | 9.1 | 13.0 |
| | DF | -30.5 | -77.0 | -59.8 | -3.1 | -11.4 | -8.2 | -1.8 | -19.3 | -9.3 | 25.8 | 14.7 | 19.1 |
| 10 | AM | -42.4 | -73.5 | -58.9 | -6.1 | -9.8 | -7.9 | -12.2 | -18.2 | -14.5 | 36.2 | 25.4 | 29.5 |
| | CF | -59.1 | -66.3 | -61.7 | -8.8 | -10.5 | -9.5 | 5.1 | -5.3 | -1.5 | 30.0 | 16.8 | 23.1 |

| | | | | | | | | | | | | |
|---|---|---|---|---|---|---|---|---|---|---|---|---|
| MG | -50.3 | -66.7 | -58.3 | -5.6 | -8.3 | -7.1 | -5.5 | -18.4 | -11.9 | 18.3 | 11.4 | 15.8 |
| DF | -38.0 | -61.8 | -48.3 | -2.7 | -8.2 | -4.9 | -11.9 | -34.8 | -23.9 | 25.5 | 8.9 | 17.2 |

**Comment 7:** L266-269. Speculative, can you provide a reference to show that these processes cause secondary evaporation.

**Response:** I have included references here. In addition, based on other reviewers ' suggestions, this part was moved to the discussion.

L363-366: Due to a low atmospheric temperature, low cloud base height, and low air-saturated water vapor loss, the alpine meadow zone was weakly affected by secondary evaporation during precipitation. There, the slope of the LMWL (8.4) was even higher than that of the GMWL (Zhang et al., 2012).

**Comment 8:** L270-275. A reference would also help here

**Response:** I have added references here. In addition, based on the suggestions of other reviewers, this part was moved to the discussion.

L366-368: As the altitude decreased, the secondary evaporation under clouds strengthened, and the slope of the LMWL of each vegetation zone decreased (Pang et al., 2011) (Fig 5).

**Comment 9:** L275-276. Seems redundant as I'm not sure where else the moisture could come from.

**Response:** This part was deleted.

**Comment 10:** L288-295. Again, lacks detail. It is difficult to follow these arguments when the data is discussed in very vague terms.

**Response:** I used data to describe this part of the content.

L310-317: During the study period, compared with that in other vegetation belts, the surface isotopic value of the soil water in the mountain grasslands was relatively enriched (-24.3‰, $\delta^2$H; -0.8‰, $\delta^{18}$O), the lc-excess was smaller and deeper into the middle and lower soil layers (-25.8‰), and the gravimetric water content was relatively low (8.4%). Due to the difference in vegetation types and the influence of reservoirs, this change did not have an obvious elevation effect. Although the elevation was low, the soil water of the deciduous forests had more depleted isotopic characteristics and higher soil moisture than those of the mountain grasslands in most samples.

**Comment 11:** L298. What is "dynamic fractionation"?

**Response:** Here should be "evaporation".

L319-321: The low-temperature natural environment made alpine meadow soil less affected by evaporation (lc-excess > -20‰), and the gravimetric water content was high (gravimetric water content > 20%) during the whole study period.

**Comment 12:** L296-309. As with much of the rest of this section, I struggled to follow the details. The explanations are not clear, there are a fair number of general statements that lack detail, and a number of findings that are not obvious. For example, "Evaporation signal can easily penetrate deep soil, which made the GWC value of all sampling activities at this site lower than 20% (Fig.6)" which seems to be at odds with "With the increase of soil depth, the fractionation signal gradually weakened".

**Response:** I used data to describe this part of the content.

L317-330: Soil profiles obtained from different vegetation zones can reflect the evaporation signals of water. The low-temperature natural environment made alpine meadow soil less affected by evaporation (lc-excess > -20‰), and the gravimetric water content was high (gravimetric water content > 20%) during the whole study period. The surface soil water of the coniferous forests was easily affected by climate and had a higher isotopic composition (-29.5‰, $\delta^2$H; -2.1‰, $\delta^{18}$O) and lower lc-excess (-26.3‰). Due to evaporation, soil water isotopes in the mountain grassland and deciduous forest areas were enriched in the surface soil layer. In particular, in the mountain grasslands, the average values of $\delta^2$H and $\delta^{18}$O in the 0-10 cm soil layer were as high as -24.4‰ and -1.2‰, respectively, and SWlc-excess was lower than -25‰, even close to -40‰ in some samples. In this case, the evaporation signals can easily penetrate the deep soil, making the gravimetric water content values at all the sampling sites lower than 20% (Fig. 4; Fig. 6).

**Discussion**

**This section has some interesting ideas in it but it is not well linked to the data in the study. You need to show how the data that you collected informs our understanding. Some of the later part of this section is written more like an introductory literature review.**

**Section 5.1**

L325-354. Some of this section describes the data (the observations on soil moisture) and that material belongs in the results.

**Response:** We reanalyzed Figure 7 in the results.

L334-355:**4.4 Variations in the water storage capacity of the 0-40 cm soil layer in different vegetation areas**

This study used soil water to calculate the water storage of the 0-40 cm soil layer in the four vegetation zones during the observation period (Fig 7). The water storage capacity of the alpine meadow gradually decreased from April to July (209.7~167.2 mm), and the water storage capacity increased after July (167.2~201.8 mm). The monthly average water storage capacity was the lowest at 0-10 cm (43.0 mm) and the highest at 30-40 cm (51.7 mm). The water storage capacity of the coniferous forest gradually decreased from April to July (150.1~101.2 mm), and the water storage capacity increased after July (101.2~160.0 mm). The monthly average water storage capacity was the lowest at 0-10 cm (28.0 mm) and the highest at 30-40 cm (40.0 mm). The water storage capacity of the mountain grassland gradually decreased from April to July (80.3~64.0 mm), and the water storage capacity increased after July (64.0~104.6 mm). The monthly average water storage capacity was the lowest at 0-10 cm (17.5 mm) and the highest at 20-30 cm (22.0 mm). The water storage capacity of the deciduous forest gradually decreased from April to June (159.3~104.0 mm), the water storage capacity increased from June to August (104.0~154.0 mm), and there was a decrease from August to October (154.0~111.8 mm). The monthly average water storage capacity was the lowest at 0-10 cm (29.1 mm) and the highest at 20-30 cm (35.0 mm). In general, the soil water storage capacity of the 0-10 cm soil layer was less than that of the other soil layers. The order of the water storage capacity of the 0-40 cm soil layer in the four vegetation zones is AM > DF > CF > MG.

**Section 5.2**

This section does not link well with the results. It is difficult to follow how your data help you make these conclusions. More justification and explanations are required. Moreover, there is little discussion of processes here – how does the data help understand how processes operate? You have concentrated on discussing the isotopic variability, without using it to understand what is going on.

This is the section where you should discuss aspects such as the memory effect and runoff generation, but you do not do so.

**Response:** According to your suggestion, I improved the discussion of this part.

**Section 5.3.**

This section reads more like an introduction. It is not clear how what you have done in this study relates to these broad general findings. As with the Introduction, you need to make a clearer link between your study and these general statements. This are all important issues, but there needs to be linkages.

Climate change in mentioned several times, but it is not clear how your study informs our understanding of its impacts. Those types of links need to be made clearer if they exist. Likewise, there are comments about groundwater recharge and runoff but no indication of how your results help understand those processes. Runoff generation was not actually discussed in the body of the paper (it appears in the introduction and the end of the discussion, but not in the discussion of the specific results).

Same comments apply to: subsurface runoff (presume that you mean interflow?); the management practices; human activities. These are topics that all appear in this section with no real link to the data in the rest of the paper.

There are also a number of superfluous details here. For example, why is mining waste (L426-428) relevant to this study.

**Response:** We logically sorted out the full text based on the reviewers' comments. The discussion on runoff generation in the watershed does not match the theme of this manuscript (Evaporation, infiltration and storage of soil water in different vegetation zones in the Qilian mountains: a stable isotope perspective). Therefore, the manuscript focused on soil moisture's evaporation, infiltration, and storage mechanism in the study area. Based on this, we reorganized this part of the content. Your comments have further improved the logic of the article.

[revised manuscript text omitted]

**Conclusions**

As with the discussion, the links to the study are not well made. In some ways this material is less general than some of the latter parts of the Discussion (Section 5.3) and it would be worth reordering so that you have the more general ideas at the end.

**Response:** Based on the opinions of the three reviewers, we re-summarized the conclusions of the manuscript.

L488-515: This work provides further insights into the movement and mixing of soil water in different vegetation zones in arid source regions. During the study period, the dynamic changes in lc-excess in the soil profiles of different vegetation zones reflected the evaporation signals caused by drought. Soil water evaporation in spring and summer and insufficient precipitation during the drought period were the main driving forces of isotopic enrichment in the surface soil. The evaporation intensity results of the four vegetation zones followed the order of mountain grassland > deciduous forest > coniferous forest > alpine meadow. In the mountain grassland and deciduous forest zones, drought caused the evaporation signal to penetrate deep into the middle and lower soil layers. The SWlc-excess below 70 cm of the ground surface remained negative. Soil water isotopes and gravimetric water content record the process of soil rewetting caused by precipitation input and mixing. The alpine meadow and coniferous forest zones were enriched in precipitation. After a short period of weak evaporation, the soil was rewetted by the next precipitation event. There was only sporadic precipitation in the mountainous grassland and deciduous forest belt from mid-May to late July. After July, the temperature dropped, and continuous precipitation wet the soil again after two months of drought. The mountain grassland and deciduous forest zones had only sporadic precipitation from mid-May to late July. With the decrease in air temperature and continuous precipitation after July, the soil was rewetted after two months of drought. Moisture and temperature conditions were the key factors that restricted the soil water storage

capacity in the different vegetation zones. The soil water storage capacity results followed the order of alpine meadow > deciduous forest > coniferous forest > mountainous grassland. The water storage capacity of the surface soil in each vegetation zone was weak, and more water was stored in the middle and lower soil layers with higher clay contents. This research is helpful to understand the hydrological cycle in different vegetation areas and can provide theoretical support for obtaining a regional ecological hydrological balance.

**Response to Reviewer #3**

**Reviewer#3: This paper presents an interesting hydrological and runoff study from the Qilian region where water and soil water samples were obtained across different climatic, topographical and vegetative conditions in order to understand the infiltration, evaporation and storage processes. The paper is well structured, but major issues need fixing as also suggested by the other referees. Overall, the English language needs to be proofread and words such as "obvious" should be avoided. The Abstract needs substantial work to emphasize the purpose/objectives of the study, describe the methods used, and to relate quantifiable results. Further discussion of the results themselves is required as well as linking the results obtained (what are the observations withdrawn from the data) to previous research. Please see additional comments below.**

Thanks for your comments.

**Abstract**

**Comment 1:** L11-12: Is this really true? That in arid areas most of the water comes from mountains? How about low lands? And groundwater? I think this sentence is not needed.

**Response:** Based on your suggestion, we have deleted it.

**Comment 2:** L12: should be the "processes" not "process".

**Response:** This spelling error has been corrected.

**Comment 3:** L13: "have" instead of "has".

**Response:** This spelling error has been corrected.

L10-13: The processes of water storage and runoff generation have not been fully understood in different vegetation zones in mountainous areas, which is the main obstacle to further understanding hydrological processes and improving water resource assessments.

**Comment 4:** L15: instead of "In current study" use "In this study"

**Response:** We corrected this.

**Comment 5:** L15-17: This is an important sentence that summarizes the work done. I would suggest to rewrite it being more specific to which isotopes, which types of vegetation zones and why this is needed.

**Response:** Based on suggestions from you and other reviewers, we rewrite this part.

L13-17: To further understand the process of soil water movement and runoff generation in different vegetation zones (alpine meadow, coniferous forest, mountain grassland, and deciduous forest) in mountainous areas, this study monitored the temporal and spatial dynamics of hydrogen and oxygen stable isotopes in the precipitation and soil water of the Xiying River Basin.

**Comment 6:** L17: Weak compared to what? Results should be quantified instead of using "weak" and "save up".

**Response:** We have clarified the evaporation intensity through the SWlc-excess of each vegetation zone and verified it through meteorological data.

L17-24: The results show that the evaporation intensities of the four vegetation zones followed the order of mountain grassland > deciduous forest > coniferous forest > alpine meadow. The soil water in the alpine meadows and coniferous forests evaporated from only the topsoil, and the rainfall input was fully mixed with each layer of soil. The evaporation signals of the mountain grasslands and deciduous forests could penetrate deep into the middle and lower layers of the soil as precipitation quickly flowed into the deep soil through the soil matrix.

**Comment 7:** L19-21: What is the result in the paper that lead to this hypothesis? The authors need to add evidence of this instead of speculating.

**Response:** This part of the result has been changed, we have added data and information.

L17-27: The results show that the evaporation intensities of the four vegetation zones followed the order of mountain grassland > deciduous forest > coniferous forest > alpine meadow. The soil water in the alpine meadows and coniferous forests evaporated from only the topsoil, and the rainfall input was fully mixed with each layer of soil. The evaporation

signals of the mountain grasslands and deciduous forests could penetrate deep into the middle and lower layers of the soil as precipitation quickly flowed into the deep soil through the soil matrix. Each vegetation zone's soil water storage capacity followed the order of alpine meadow > deciduous forest > coniferous forest > mountain grassland. In addition, the water storage capacity of shallow soils in different types of vegetation areas was weaker than that of deep soils.

**Comment 8:** L22: What is evaporate strongly? How much?

**Response:** We have clarified the evaporation intensity through the SWlc-excess of each vegetation zone and verified it through meteorological data.

L17-19: The results show that the evaporation intensities of the four vegetation zones followed the order of mountain grassland > deciduous forest > coniferous forest > alpine meadow.

**Comment 9:** L21-22: The lower elevation vegetation zones within the Mountain Grassland and Deciduous forest? Aren't these areas at high altitude?

**Response:** This sentence was restated. Previously, it was to express that Mountain Grassland (2390 m) and Deciduous Forest (2097 m) are relatively low-altitude areas compared to Alpine Meadow (3637 m) and Coniferous Forest (2721 m).

**Comment 10:** L25: Delete word "reasonably"

**Response:** "Reasonably" has been deleted.

L27-28: This work will provide a new reference for understanding soil hydrology in arid headwater areas.

**Introduction**

**Comment 1:** L39-40: Soil water in the unsaturated zone from precipitation can transform into water vapour or groundwater recharge.

**Response:** I have modified this sentence according to your suggestion.

L32-34: As an important part of the water cycle, soil water in the unsaturated zone can be converted from precipitation to stream or groundwater recharge.

**Comment 2:** Line 40: Delete "Its".

**Response:** "Its" has been deleted.

**Comment 3:** Line 41: Delete "very".

**Response:** This word "very" has been deleted.

**Comment 4:** Line 48: Storage is not a transport mechanism

**Response:** I agree with your comment, this problem has been corrected.

L44-45: Water seepage in the unsaturated soil zone and the water evaporation at the air‑soil interface are the main forms of soil water transport.

**Comment 5:** Line 54: is it soil water profiles?

**Response:** This part of the content has been deleted.

**Comment 6:** Line 56: Delete "In addition,".

**Response:** This part of the content has been deleted.

**Comment 7:** Line 58: Describe what the d-excess is.

**Response:** "d-excess" has been introduced in detail.

L60-62: Dansgaard (1964) proposed the concept of d-excess (d-excess=$\delta^2$H-8$\delta^{18}$O) to illustrate the intensity of evaporative fractionation. In the state of isotopic equilibrium, the d-excess is 10.

**Comment 8:** Line 66-70: Delete ",and" and rewrite following sentence. It is not clear at the moment.

**Response:** This part of the content has been deleted.

**Comment 9:** Line 71: Do not use "Generally speaking" in a scientific publication

**Response:** "Generally speaking" has been deleted.

**Comment 10:** Line 71: Do "wet" areas refer to tropical regions?

**Response:** This should be a "humid area", which refers to an area with humid air and abundant rainfall.

**Comment 11:** Line 80: can better help adapt.

**Response:** This place has been corrected.

L80-83: Understanding the climatic and hydrological conditions of different vertical vegetation zones and clarifying the regulating role of vegetation in the water cycle can help better adapt to climate change's influences on the hydrological cycle in source areas.

**Comment 12:** Line 82: "In this study" instead of "In current study".

**Response:** This place has been corrected.

L83-86: This study monitored the stable isotope composition of precipitation and soil water and the spatiotemporal dynamics of soil water storage in four vegetation zones (alpine meadow, coniferous forest, mountain grassland, and deciduous forest) with different hydrothermal conditions in the Xiying River Basin.

**Comment 13:** Line 82: "," after soil water.

**Response:** This place has been corrected.

L83-86: This study monitored the stable isotope composition of precipitation and soil water and the spatiotemporal dynamics of soil water storage in four vegetation zones (alpine meadow, coniferous forest, mountain grassland, and deciduous forest) with different hydrothermal conditions in the Xiying River Basin.

**Comment 14:** Line 83: Is it in four regions of different climate, vegetation and topographical conditions? As opposed to vegetation zones?

**Response:** Our research objects are four typical vertical vegetation belts in mountainous areas, and their climatic conditions, topography and dominant species are different.

**Comment 15:** Line 85: Then, it can be clarified that this study explores how evaporation, infiltration and storage processes differ within these four regions according to the climate, vegetation and topography.

**Response:** Yes, this was confirmed in the follow-up discussion.

L86-92: To explore the differences in soil water evaporation, infiltration, and storage processes in these four different climates, vegetation types, and terrain types, the following research objectives were proposed: (1) to explore the evolution of isotope evaporation signals and the "memory effects" of precipitation input, mixing and rewetting; and (2) to understand

the soil water storage capacity and influencing factors of four vegetation areas in mountainous areas.

**Comment 16:** Line 89: similarly to the previous comment, are the authors restricting the analysis to only vegetation zones? I would argue that the study compares regions with varying climatic, topographic and vegetative conditions.

**Response:** The climatic conditions, topography and dominant species of the vegetation zone are different. I think the description of the vegetation zone already contains the similarities and differences of these natural conditions.

**Study Area**

**Comment 1:** Line 99: ranges between 2000m and 5000m above sea level

**Response:** I revised this sentence.

L101-103: The basin elevation is between 2000 m and 5000 m, corresponding to a temperate semiarid climate with strong solar radiation, a long sunshine time, and a large temperature difference between day and night.

**Data and Methods**

**Comment 1:** L110: Delete "and determination"

**Response:** This words "and determination" have been deleted.

**Comment 2:** L116: What does "parallel" mean here?

**Response:** This is a spelling error, it should be "duplicate samples". Collect duplicate samples to improve the accuracy of the experiment.

**Comment 3:** L170-172: Equation before line 170 needs reference.

**Response:** References have been added.

L181-182: Soil water storage is the thickness of the water layer formed by all the water in a certain soil layer (Milly, 1994) and is expressed by the following formula:

**Results and Analysis**

**Comment 1:** L175: "PET" should be written Potential evapotranspiration (PET), then the authors can use PET but it needs complete spelling the first time it is used.

**Response:** Full spelling is used for the first time.

L171: **3.3.2 Potential evapotranspiration**

**Comment 2:** L177: I assume it is also the daily evapotranspiration? Need to make it explicit which type of evaporation.

**Response:** I double-checked the usage of the text. We calculated the daily potential evapotranspiration during the study period, and obtained the potential evapotranspiration for each month and the entire observation period by summing.

**Comment 3:** L184: Delete "generally speaking".

**Response:** This words "generally speaking" has been deleted.

**Comment 4:** L191: "temperature" instead of heat

**Response:** I have modified this.

L203-205: To explore the differences in the natural environment in different vegetation zones, air temperature, atmospheric humidity, and precipitation were used to indicate each research site's temperature and moisture conditions.

**Comment 5:** L194: 72 precipitation events? Make it explicit, where all these rainfall?

**Response:** I have explained in detail here.

L208-210: During the observation period, there were 72 precipitation events in the alpine meadow zone, and the total precipitation was 534.3 mm, which was relatively evenly distributed each month.

**Comment 6:** L207: Rewrite sentence to "The temperature of the studied regions was ordered as follow:"

**Response:** According to your suggestion, this part has been rewritten.

L221-224: The temperatures of the studied regions were ordered as follows: AM (alpine meadow) < CF (coniferous forest) < MG (mountain grassland) < DF (deciduous forest). The humidities of the studied regions were ordered as follows: AM > CF > MG > DF (Fig. 2).

**Comment 7:** L208: Define first what AM, CF, MG, and DF mean

**Response:** Full spelling is used for the first time.

L221-224: The temperatures of the studied regions were ordered as follows: AM (alpine meadow) < CF (coniferous forest) < MG (mountain grassland) < DF (deciduous forest). The humidities of the studied regions were ordered as follows: AM > CF > MG > DF (Fig. 2).

**Comment 8:** L213: Do not use obviously in scientific publications, you can say what it was significantly different? Did you do any statistical analysis to conclude this? If so what please mention it in the results

**Response:** This has been rewritten, and the difference is analyzed in the subsequent results.

L226-228: Influenced by different water sources and complex weather conditions in the precipitation process, the isotopic compositions of precipitation in the four vegetation zones were different during the study period.

**Comment 9:** L211-220. This info would be better in a table

**Response:** Based on your suggestion, we have drawn a table.

L246-247: **Table 2** General characteristics of precipitation $\delta^2H$ and $\delta^{18}O$ in different vegetation areas from April to October 2017

| Vegetation zone | $\delta^2H$/‰ | | | | $\delta^{18}O$/‰ | | | |
|---|---|---|---|---|---|---|---|---|
| | Max | Min | mean | SD | Max | Min | mean | SD |
| AM | 13.7 | -163.9 | -73.1 | 36.3 | -1.3 | -23.1 | -10.0 | 4.3 |
| CF | 13.0 | -117.8 | -42.0 | 37.2 | -0.1 | -17.4 | -7.1 | 4.7 |
| MG | 4.2 | -103.1 | -37.4 | 30.5 | -0.9 | -15.1 | -5.9 | 3.9 |
| DF | 23.2 | -110.2 | -31.8 | 42.8 | 3.2 | -15.2 | -5.8 | 5.5 |

**Comment 10:** L235: "The low temperature environment of Alpine Meadow and abundant and uniform precipitation events made the monthly mean values of $\delta 2H$ and $\delta 18O$ change little" how much?

**Response:** Based on the opinions reviewer, this part' content has been revised.

L253-255: The low-temperature environment and abundant precipitation events in the alpine meadows make the monthly average $\delta^2H$ and $\delta^{18}O$ of soil water more depleted than other vegetation zones (-69.4~-51.6‰, $\delta^2H$; -7.5~-10.3‰, $\delta^2H$).

**Comment 11:** L239: "Evaporation fractionation of soil water isotopes in Coniferous Forests was more intense." More intense than what? These kind of statements need quantification.

**Response:** According to the opinions of the three reviewers, this sentence is meaningless here. This part mainly analyzes the temporal changes of soil water isotopes in different vegetation zones. Therefore, I deleted this part of the content.

**Comment 12:** L277-278: "With the decrease of altitude, the soil water evaporation became stronger and stronger, except soil in Deciduous Forest". This sentence does not make sense, please rewrite and quantify stronger.

**Response:** Based on your suggestion, I have changed and explained it here.

L298-302: With the decrease in altitude, the slope of the SWL in all vegetation zones except for the deciduous forest SWL decreased (AM: 6.4; CF: 4.7; MG: 3.4; DF: 4.1), indicating that the evaporation of soil moisture increased. On the one hand, the vegetation coverage of the deciduous forest site was higher.

**Discussion**

**I am in agreement with the comments of Referee 1 and Referee 2 concerning the discussion. It feels more like a summary of previous studies. The authors need to refer to the results and put them in context of previous work and how their study is contributes to that pool of knowledge.**

**Comment 1:** L323-325: "The soil water storage capacity of Alpine Meadow with low temperature and rainy weather was obviously higher than that of other vegetation zones." The authors need to explain how this conclusion is evident from their data without using words such as "obviously" referred to figure 7 for discussion.

**Response: Response:** We logically sorted out the full text based on the reviewers' comments. The discussion on runoff generation in the watershed does not match the theme of this manuscript (Evaporation, infiltration and storage of soil water in different vegetation zones in the Qilian mountains: a stable isotope perspective). Therefore, the manuscript focused on soil moisture's evaporation, infiltration, and storage mechanism in the study area. Based on this, we reorganized this part of the content. Your comments have further improved the logic of the article.

[revised manuscript text omitted]

**Comment 2:** L440-442. Fix this sentence grammatically

**Response:** This part of the content has been replaced by new content.

**Conclusion**

**Comment 1:** L457: Storage capacity decreased (instead of weakened)

**Response:** This problem has been corrected.

L507-512: Moisture and temperature conditions were the key factors that restricted the soil water storage capacity in the different vegetation zones. The soil water storage capacity results followed the order of alpine meadow > deciduous forest > coniferous forest > mountainous grassland. The water storage capacity of the surface soil in each vegetation zone was weak, and more water was stored in the middle and lower soil layers with higher clay contents.

**Comment 2:** L461: Soil "water" evaporation in spring…

**Response:** This problem has been corrected.

L491-493: Soil water evaporation in spring and summer and insufficient precipitation during the drought period were the main driving forces of isotopic enrichment in the surface soil.

**Comment 3:** L463: Is it "isotopic" instead of "isotopci"?

**Response:** This problem has been corrected.

L491-493: Soil water evaporation in spring and summer and insufficient precipitation during the drought period were the main driving forces of isotopic enrichment in the surface soil.

**Comment 4:** L463-465. This sentence needs fixing. I could not understand what it conveys.

**Response:** I have revised this sentence.

L495-498: In the mountain grassland and deciduous forest zones, drought caused the evaporation signal to penetrate deep into the middle and lower soil layers. The SWlc-excess below 70 cm of the ground surface remained negative.

---

## Referee Report (RR1)

**The authors have made considerable efforts in addressing the comments of the reviewers. However, there are still some issues with this paper. The text is still not very clear in places, although the grammar is much better. In part it is due to some complex sentences and specific phrases. This is mostly in the introduction. Overall the results and discussion are much better as they have a reasonable level of detail.**

**The paper still does not convey a strong reason as to why this work is of interest to the broader international community. For an international journal such as HESS, it is important to show how the work informs studies by groups working on similar problems elsewhere. Some better explanation of this is needed in the section on motivation and in the Conclusions.**

**I have annotated some of the responses to my original comments below (in green) where I think there are still issues. These are mainly explanations that are not clear or where more context is needed/**

Abstract

The abstract needs improvement. Abstracts are important as they are what convince the readers to look at the rest of the paper. They should convey not only what has been studied and why, but should also contain enough details so that the main conclusions are evident. This abstract needs improving, specifically:

**The abstract was improved but there still are several qualitative statements. For example the sentence below gives a relative order but no values and still uses terms such as "weak" that are not clear.**

24-27: Each vegetation zone's soil water storage capacity followed the order of alpine meadow > deciduous forest > coniferous forest > mountain grassland. In addition, the water storage capacity of shallow soils in different types of vegetation areas was weaker than that of deep soils.

Introduction

Comment 3: L50. Do you mean on the ground surface or in the near-surface part of the soils?

Response: We rewrite this sentence. L58-60: The evaporation of liquid water produces water vapor enriched in $^{1}H$ and $^{16}O$, while the remaining water is enriched in $^{2}H$ and $^{18}O$ (Ferretti et al., 2003).

**This does not answer the question as to where this is taking place.**

Comment 4: L48-75. This would not be readily understandable to many readers who had not worked with these types of data. It is too generally worded and needs details. This paragraph is important as it sets the framework for using the stable isotopes to understand processes. (1) Define that you are discussing $^{18}O$ and $^{2}H$ data (there are lots of stable isotopes!). (2) Terms such as "makes soil water isotopes enriched" are vague. Specifically, evaporation enriches the residual water in $^{18}O$ or $^{2}H$ (or increases the $\delta^{18}O$ and $\delta^{2}H$ values) (3) Likewise, "soil moisture fractionation is positively correlated with evapotranspiration but negatively correlated with precipitation". Are you talking about the magnitude or sign? (4) How significant? (5) Define the d-excess (briefly) (6) L63-70. Lacks detail and is unclear.

Response: According to the suggestions of three reviewers, this part was rewritten to solve the above problems: (1) We identified stable isotopes of hydrogen and oxygen; (2) According to your suggestion, the expression has been changed; (3) and (4) According to the reviewer's suggestion, we introduced the evaporation process more and deleted the influencing factors of evaporation; (5) We defined "d-excess"; (6) I gave a detailed description of this part to make it more expressive of the status quo of the research.

**This is still not very clear in places (see comments on the modified text below). This is an important section and there is more explanation than was in the original paper. However, some of the issues (definition of lc-index and explanation of the memory effect) are still not there.**

L44-92: Water seepage in the unsaturated soil zone and the water evaporation at the air– soil interface are the main forms of soil water transport. Seasonal variations in precipitation isotopes are often used to track the process of soil water leakage (Stumpp et al., 2012).

During the piston infiltration process, precipitation with different $\delta^2H$ and $\delta^{18}O$ peaks are retained in the soil profile and gradually disappears as the infiltration depth increases (Sprenger et al., 2016a), while the preferential flow will keep these peaks until reaching the deep soil layer (Peralta-Tapia et al., 2015). During a precipitation event, the response of the water isotopes in the surface soil to precipitation is more obvious, changing to nearly that of the stable isotopes of the precipitation. With the deepening of the soil layer, the seasonal variation in precipitation isotope signals rapidly attenuates, and the influence of precipitation on soil water gradually weakens (Sprenger et al., 2017).

**This says the same thing twice and could be clearer (all that it says is that the variation of stable isotopes in near-surface soil water are likely to reflect the rainfall variation but that these variations are attenuated with depth unless preferential flow occurs).**

Evapotranspiration is the main form of soil water dissipation.

Because the mass of hydrogen and oxygen atoms that make up water molecules are related to their thermodynamic properties, isotope fractionation of water will occur in the process of the water cycle. The evaporation of liquid water produces water vapor enriched in $^1H$ and $^{16}O$, while the remaining water is enriched in $^2H$ and $^{18}O$ (Ferretti et al., 2003).

**Confusing as written and you probably only need the last sentence.**

Dansgaard (1964) proposed the concept of d-excess (d-excess=$\delta^2H-8\delta^{18}O$) to illustrate the intensity of evaporative fractionation. In the state of isotopic equilibrium, the d-excess is 10.

**Not strictly true, d is related to humidity and the value of 10 relates to average global humidity**

Compared with d-excess, lc-excess can better explain the evaporative fractionation process. The main reason is that lc-excess of precipitation and soil water changes smoothly and has relatively small seasonal changes (Landwehr et al., 2014).

**This is not a good explanation of the lc-excess (there is an explanation below, but something is needed here – at least define the term)**

The dynamic changes in isotopes record the signal of soil water evaporation. This enrichment from dynamic fractionation exists in soil water isotopes in different climatic regions. Compared with temperate regions, the signals of evaporation in arid and Mediterranean environments penetrate deeper into the soil (Sprenger et al., 2016b). Some water is stored in the soil after evaporation and

seepage. The water storage capacity of humid areas is higher than that of arid areas, the water storage capacity of forests is higher than that of grasslands, and the water storage capacity of the surface soil layer is lower than that of deeper soil layers with higher clay content (Heinrich et al., 2019; Sprenger et al., 2019; Kleine et al., 2020; Snelgrove et al., 2021).

In alpine mountains, climate warming accelerates the melting of glaciers and frozen soil, and the dynamic interaction between water bodies stored in different media becomes the main influence on the water cycle (Penna et al., 2018).

**Not clear – what are "dynamic interactions between water bodies stored in different media" and how do they influence the water cycle.**

 Previous studies on the evaporation, infiltration, and storage of soil water have mostly focused on different climatic regions or vegetation types in the same climatic region. Understanding the climatic and hydrological conditions of different vertical vegetation zones and clarifying the regulating role of vegetation in the water cycle can help better adapt to climate change's influences on the hydrological cycle in source areas. This study monitored the stable isotope composition of precipitation and soil water and the spatiotemporal dynamics of soil water storage in four vegetation zones (alpine meadow, coniferous forest, mountain grassland, and deciduous forest) with different hydrothermal conditions in the Xiying River Basin. To explore the differences in soil water evaporation, infiltration, and storage processes in these four different climates, vegetation types, and terrain types, the following research objectives were proposed: (1) to explore the evolution of isotope evaporation signals and the "memory effects" of precipitation input, mixing and rewetting; and (2) to understand the soil water storage capacity and influencing factors of four vegetation areas in mountainous areas.

**What the "memory effects" are and why they are important is still not explained.**

**Comment 6:** L84. "Heat conditions" do you mean temperatures?

**Response:** We want to express the vegetation zone under different moisture and temperature conditions. Based on this, I re-narrate this sentence.

L83-86: This study monitored the stable isotope composition of precipitation and soil water and the spatiotemporal dynamics of soil water storage in four vegetation zones (alpine meadow, coniferous forest, mountain grassland, and deciduous forest) with different hydrothermal conditions in the Xiying River Basin.

**"Hydrothermal conditions" no clearer than "heat conditions" – do you mean temperatures?**

**Comment 7:** L82-90. These are fine as general aims, but can you explain (briefly) why this is important (i.e. what are you doing that is new, what are the broader implications?). There is a disconnect here between the broad general themes in the rest of the introduction and your specific study. Also, runoff generation and the memory effect are not explicitly discussed in any depth in the paper (need to make sure that your aims are actually what you discuss).

**Response:** Previous studies on soil moisture evaporation, infiltration and storage have mostly focused on different climatic regions or vegetation types under the same climatic region, and there are few uses of isotope technology to trace the hydrological processes in the mountain vegetation vertical zone.

**This section (L78-88) of the revised manuscript still does not give a sense of importance or broader implications.**

 **Comment 8:** L88. If it is important, define the memory effect and explain why we need to understand it.

**Response:** The "memory effect" means that the temporal and spatial changes of the stable isotope profile of soil water can reflect and characterize the input, mixing, and rewetting process of precipitation. Understanding the "memory effect" helps us trace the dynamic changes of climate and soil hydrology.

**Still not explained in the paper (L89-90 just states "to explore the evolution of isotope evaporation signals and the "memory effects" of precipitation input, mixing and rewetting").**

Study area

**Comment 4:** Fig. 1. What is the inset on the left-hand map?
**Response:** The complete nine-dotted line is shown here.

**Not clear what the figure shows or what the response means. The caption is uninformative.**

Methods

Comment 4: Section 3.2. The analysis is only part of the uncertainty. Did you perform multiple extractions on the same sample to test the uncertainty associated with that. This will undoubtedly be higher and needs to be considered.

Response: During the sampling process, we collected duplicate samples to improve the accuracy of the experiment.

**That is good but you need to explain this in the text.**

Results

Comment 1: L175-178. How precise are these values (i.e. is the 1dp precision warranted)? What was the rainfall during those times?

Response: We use FAO Penman-Monteith to calculate the potential daily evapotranspiration (possible evapotranspiration) in the study area (the software calculation results are kept to three decimal places, and we keep one decimal place in the study). This illustrates the date when the maximum and minimum values appear, and there may be no rainfall on that day.

**Regardless of the calculations, I am still sceptical that the results are that accurate.**

**Overall, however, the presentation of the Results and Discussion is much better and the added details help get the message across.**

**Conclusions** As with the discussion, the links to the study are not well made. In some ways this material is less general than some of the latter parts of the Discussion (Section 5.3) and it would be worth reordering so that you have the more general ideas at the end.

**Response:** Based on the opinions of the three reviewers, we re-summarized the conclusions of the manuscript.

**This section still does not explain the true value of the study and why it is important to a broader readership. What is it that you have done here that informs work elsewhere? This is important for a paper in an international journal.**

---

## Author Response (AR2)

**Revision Notes**

Dear Editor and Reviewers:

Thank you for your letter and the reviewers' comments concerning our manuscript entitled "Evaporation, infiltration and storage of soil water in different vegetation zones in the Qilian Mountains: A stable isotope perspective" (Manuscript Number: Hess-2021-376).

According to the reviewers' comments, we have made careful revisionsto the manuscript. In the revised version of the manuscript, the revised portions have been marked in red. The main corrections and the response to the reviewers' comments are as follows.

**Responses to the reviewer's comments:**

**The authors have made considerable efforts in addressing the comments of the reviewers. However, there are still some issues with this paper. The text is still not very clear in places, although the grammar is much better. In part it is due to some complex sentences and specific phrases. This is mostly in the introduction. Overall the results and discussion are much better as they have a reasonable level of detail.**

**The paper still does not convey a strong reason as to why this work is of interest to the broader international community. For an international journal such as HESS, it is important to show how the work informs studies by groups working on similar problems elsewhere. Some better explanation of this is needed in the section on motivation and in the Conclusions.**

**I have annotated some of the responses to my original comments below (in green) where I think there are still issues. These are mainly explanations that are not clear or where more context is needed.**

Thanks for your comments.

**Abstract**

**Comment:** The abstract needs improvement. Abstracts are important as they are what convince the readers to look at the rest of the paper. They should convey not only what has been studied and why, but should also contain enough details so that the main conclusions are evident. This abstract needs improving, specifically:

The abstract was improved but there still are several qualitative statements. For example the sentence below gives a relative order but no values and still uses terms such as "weak" that are not clear.

24-27: Each vegetation zone's soil water storage capacity followed the order of alpine meadow > deciduous forest > coniferous forest > mountain grassland. In addition, the water storage capacity of shallow soils in different types of vegetation areas was weaker than that of deep soils.

**Response:** According to your suggestion, we have further improved the Abstract section, including quantification and examination of the main conclusions.

L18-20: The results show that the order of soil water evaporation intensities in the four vegetation zones was mountain grassland ($SWL_{slop}$: 3.4) > deciduous forest ($SWL_{slop}$: 4.1) > coniferous forest ($SWL_{slop}$: 4.7) > alpine meadow ($SWL_{slop}$: 6.4).

L25-30: Each vegetation zone's water storage capacity of the 0-40 cm soil layer followed the order of alpine meadow (46.9 mm) > deciduous forest (33.0 mm) > coniferous forest (32.1 mm) > mountain grassland (20.3 mm). In addition, the 0-10cm soil layer has the smallest soil water storage capacity (alpine meadow:43.0 mm; coniferous forest: 28.0 mm; mountain grassland: 17.5 mm; deciduous forest: 29.1 mm).

**Introduction**

**Comment 1:** Comment 3: L50. Do you mean on the ground surface or in the near-surface part of the soils?

Response: We rewrite this sentence.

L58-60: The evaporation of liquid water produces water vapor enriched in $^1H$ and $^{16}O$, while the remaining water is enriched in $^2H$ and $^{18}O$ (Ferretti et al., 2003).

This does not answer the question as to where this is taking place.

**Response:** We re-answered the question about where this is taking place, showing that evaporation mainly occurred in the near-surface part of the soils.

L54-57: Evaporation mainly occurred in the near-surface part of the soils (0-10 cm), and the light isotope molecules ($^1H$ and $^{16}O$) evaporated preferentially, resulting in the enrichment of heavy isotopes ($^2H$ and $^{18}O$) on the soil surface (Ferretti et al., 2003).

**Comment 2:** Comment 4: L48-75. This would not be readily understandable to many readers who had not worked with these types of data. It is too generally worded and needs details. This paragraph is important as it sets the framework for using the stable isotopes to understand processes. (1) Define that you are discussing $^{18}O$ and $^2H$ data (there are lots of stable isotopes!). (2) Terms such as "makes soil water isotopes enriched" are vague. Specifically, evaporation enriches the residual water in $^{18}O$ or $^2H$ (or increases the $\delta^{18}O$ and

δ²H values) (3) Likewise, "soil moisture fractionation is positively correlated with evapotranspiration but negatively correlated with precipitation". Are you talking about the magnitude or sign? (4) How significant? (5) Define the d-excess (briefly) (6) L63-70. Lacks detail and is unclear.

Response: According to the suggestions of three reviewers, this part was rewritten to solve the above problems: (1) We identified stable isotopes of hydrogen and oxygen; (2) According to your suggestion, the expression has been changed; (3) and (4) According to the reviewer's suggestion, we introduced the evaporation process more and deleted the influencing factors of evaporation; (5) We defined "d-excess"; (6) I gave a detailed description of this part to make it more expressive of the status quo of the research.

This is still not very clear in places (see comments on the modified text below). This is an important section and there is more explanation than was in the original paper. However, some of the issues (definition of lc-index and explanation of the memory effect) are still not there.

**Response:** At the appropriate place in the text, we added the definition of lc-index (L61-64) and the explanation of the "memory effect" (L47-53).

L62-65: Landwehr and Coplen (2006) defined line conditioned excess as the difference between the δ²H value of the water sample and the δ¹⁸O linear transform value of the same sample, where the linear transformation reflects the relevant referenced meteoric water relationship.

L48-54: The dynamic water process reflected by the displacement of the isotope signal on the soil profile is called the "memory effect". Understanding the "memory effect" will help us to trace the dynamic changes in climate and soil hydrology (Kleine et al., 2020). The change of stable isotopes in near-surface soil water may reflect the precipitation variation, but these variations decrease with depth unless there is preferential flow (Peralta-Tapia et al., 2015; Sprenger et al., 2016; Sprenger et al., 2017).

**Comment 3:** L44-92: Water seepage in the unsaturated soil zone and the water evaporation at the air-soil interface are the main forms of soil water transport. Seasonal variations in precipitation isotopes are often used to track the process of soil water leakage (Stumpp et al., 2012).

During the piston infiltration process, precipitation with different δ2H and δ18O peaks are retained in the soil profile and gradually disappears as the infiltration depth increases (Sprenger et al., 2016a), while the preferential flow will keep these peaks until reaching the

deep soil layer (Peralta-Tapia et al., 2015). During a precipitation event, the response of the water isotopes in the surface soil to precipitation is more obvious, changing to nearly that of the stable isotopes of the precipitation. With the deepening of the soil layer, the seasonal variation in precipitation isotope signals rapidly attenuates, and the influence of precipitation on soil water gradually weakens (Sprenger et al., 2017).

This says the same thing twice and could be clearer (all that it says is that the variation of stable isotopes in near-surface soil water are likely to reflect the rainfall variation but that these variations are attenuated with depth unless preferential flow occurs)

**Response:** Based on your suggestion, we have revised this section.

L48-54: The dynamic water process reflected by the displacement of the isotope signal on the soil profile is called the "memory effect". Understanding the "memory effect" will help us to trace the dynamic changes in climate and soil hydrology (Kleine et al., 2020). The change of stable isotopes in near-surface soil water may reflect the precipitation variation, but these variations decrease with depth unless there is preferential flow (Peralta-Tapia et al., 2015; Sprenger et al., 2016; Sprenger et al., 2017).

**Comment 4:** Evapotranspiration is the main form of soil water dissipation. Because the mass of hydrogen and oxygen atoms that make up water molecules are related to their thermodynamic properties, isotope fractionation of water will occur in the process of the water cycle. The evaporation of liquid water produces water vapor enriched in $^1$H and $^{16}$O, while the remaining water is enriched in $^2$H and $^{18}$O (Ferretti et al., 2003).

Confusing as written and you probably only need the last sentence.

**Response:** We have edited this section to make it more concise.

L54-57: Evaporation mainly occurred in the near-surface part of the soils (0-10 cm), and the light isotope molecules ($^1$H and $^{16}$O) evaporated preferentially, resulting in the enrichment of heavy isotopes ($^2$H and $^{18}$O) on the soil surface (Ferretti et al., 2003).

**Comment 5:** Dansgaard (1964) proposed the concept of d-excess (d-excess=$\delta^2$H-8$\delta^{18}$O) to illustrate the intensity of evaporative fractionation. In the state of isotopic equilibrium, the d-excess is 10.

Not strictly true, d is related to humidity and the value of 10 relates to average global humidity

**Response:** By further reviewing the literature, we have revised our interpretation of d-excess.

L57-62: Dansgaard (1964) proposed the concept of d-excess (d-excess=$\delta^2$H-8$\delta^{18}$O) to illustrate the intensity of evaporation fractionation. Assuming that evaporation occurs in the atmosphere with a humidity of 75%, it shows that the d-excess value of atmospheric moisture accounts for the d-excess value of 10‰ in the atmospheric moisture, which conforms to the worldwide average isotopic labelling of meteoric waters.

**Comment 6:** Compared with d-excess, lc-excess can better explain the evaporative fractionation process. The main reason is that lc-excess of precipitation and soil water changes smoothly and has relatively small seasonal changes (Landwehr et al., 2014).

This is not a good explanation of the lc-excess (there is an explanation below, but something is needed here-at least define the term)

**Response:** We added the definition of lc-excess.

L62-65: Landwehr and Coplen (2006) defined line conditioned excess as the difference between the $\delta^2$H value of the water sample and the $\delta^{18}$O linear transform value of the same sample, where the linear transformation reflected the relevant referenced meteoric water relationship.

**Comment 7:** The dynamic changes in isotopes record the signal of soil water evaporation. This enrichment from dynamic fractionation exists in soil water isotopes in different climatic regions. Compared with temperate regions, the signals of evaporation in arid and Mediterranean environments penetrate deeper into the soil (Sprenger et al., 2016b). Some water is stored in the soil after evaporation and seepage. The water storage capacity of humid areas is higher than that of arid areas, the water storage capacity of forests is higher than that of grasslands, and the water storage capacity of the surface soil layer is lower than that of deeper soil layers with higher clay content (Heinrich et al., 2019; Sprenger et al., 2019; Kleine et al., 2020; Snelgrove et al., 2021).

In alpine mountains, climate warming accelerates the melting of glaciers and frozen soil, and the dynamic interaction between water bodies stored in different media becomes the main influence on the water cycle (Penna et al., 2018).

Not clear-what are "dynamic interactions between water bodies stored in different media" and how do they influence the water cycle.

**Response:** We briefly described the dynamic interactions between water bodies and its influence on the water cycle.

L80-87: Interactions between precipitation and the soil-plant-atmosphere system determine the distribution of water in various storage reservoirs and the subsequent release of water vapor to the atmosphere. These interactions include mainly interception, throughfall, canopy drip, snow accumulation and ablation, infiltration, surface and subsurface runoff, soil moisture, and the partitioning of evapotranspiration between canopy evaporation, transpiration, and soil evaporation. As the main links of the hydrological cycle, these processes have a profound impact on regional water balance and flux distribution.

**Comment 8:** Previous studies on the evaporation, infiltration, and storage of soil water have mostly focused on different climatic regions or vegetation types in the same climatic region. Understanding the climatic and hydrological conditions of different vertical vegetation zones and clarifying the regulating role ofvegetation in the water cycle can help better adapt to climate change's influences on the hydrological cycle in source areas. This study monitored the stable isotope composition of precipitation and soil water and the spatiotemporal dynamics of soil water storage in four vegetation zones (alpine meadow, coniferous forest, mountain grassland, and deciduous forest) with different hydrothermal conditions in the Xiying River Basin. To explore the differences in soil water evaporation, infiltration, and storage processes in these four different climates, vegetation types, and terrain types, the following research objectives were proposed: (1) to explore the evolution of isotope evaporation signals and the "memory effects" of precipitation input, mixing and rewetting; and (2) to understand the soil water storage capacity and influencing factors of four vegetation areas in mountain areas.

What the "memory effects" are and why they are important is still not explained.

**Response:** At the appropriate place in the text, we added the explanation of the "memory effect".

L48-54: The dynamic water process reflected by the displacement of the isotope signal on the soil profile is called the "memory effect". Understanding the "memory effect" will help us to trace the dynamic changes in climate and soil hydrology (Kleine et al., 2020). The change of stable isotopes in near-surface soil water may reflecs the precipitation variation, but these variations decrease with depth unless there is preferential flow (Peralta-Tapia et al., 2015; Sprenger et al., 2016; Sprenger et al., 2017).

**Comment 9:** Comment 6: L84. "Heat conditions" do you mean temperatures?

Response: We want to express the vegetation zone under different moisture and temperature

conditions. Based on this, I re-narrate this sentence.

L83-86: This study monitored the stable isotope composition of precipitation and soil water and the spatiotemporal dynamics of soil water storage in four vegetation zones (alpine meadow, coniferous forest, mountain grassland, and deciduous forest) with different hydrothermal conditions in the Xiying River Basin.

"Hydrothermal conditions" no clearer than "heat conditions" – do you mean temperatures?

**Response:** What we want to express here are four vegetation zones with different temperatures and humidity, all of which have been modified.

L93-96: In this study, we monitored the stable isotope composition of precipitation and soil water and the spatio-temporal dynamics of soil water storage in four vegetation zones (alpine meadow, coniferous forest, mountain grassland, and deciduous forest) at different temperatures and humidity in the Xiying River basin.

**Comment 10:** Comment 7: L82-90. These are fine as general aims, but can you explain (briefly) why this is important (i.e. what are you doing that is new, what are the broader implications?). There is a disconnect here between the broad general themes in the rest of the introduction and your specific study. Also, runoff generation and the memory effect are not explicitly discussed in any depth in the paper (need to make sure that your aims are actually what you discuss).

Response: Previous studies on soil moisture evaporation, infiltration and storage have mostly focused on different climatic regions or vegetation types under the same climatic region, and there are few uses of isotope technology to trace the hydrological processes in the mountain vegetation vertical zone.

This section (L78-88) of the revised manuscript still does not give a sense of importance or broader implications.

**Response:** We have add the importance of this study and its broader implications.

L102-108: It is hoped that this study can further improve the understanding of the water cycle process and provide a scientific theoretical reference for water resource utilization and ecological restoration in fragile environments. More importantly, it can provide paradigms for research at different spatial scales (latitude zone, longitude zone, watershed, etc.) based on the knowledge of soil moisture evaporation, infiltration, and water storage in typical vertical vegetation zones.

**Comment 11:** Comment 8: L88. If it is important, define the memory effect and explain why we need tounderstand it.

Response: The "memory effect" means that the temporal and spatial changes of the stable isotope profile of soil water can reflect and characterize the input, mixing, and rewetting process of precipitation. Understanding the "memory effect" helps us trace the dynamic changes of climate and soil hydrology.

Still not explained in the paper (L89-90 just states "to explore the evolution of isotope evaporation signals and the "memory effects" of precipitation input, mixing and rewetting").

**Response:** At the appropriate place in the text, we added the explanation of the "memory effect" (L47-53).

L48-54: The dynamic water process reflected by the displacement of the isotope signal on the soil profile is called the "memory effect". Understanding the "memory effect" will help us to trace the dynamic changes in climate and soil hydrology (Kleine et al., 2020). The change of stable isotopes in near-surface soil water may reflect the precipitation variation, but these variations decrease with depth unless there is preferential flow (Peralta-Tapia et al., 2015; Sprenger et al., 2016; Sprenger et al., 2017).

**Study area**

**Comment:** Comment 4: Fig. 1. What is the inset on the left-hand map?

Response: The complete nine-dotted line is shown here.

Not clear what the figure shows or what the response means. The caption is uninformative

**Response:** To make the diagram more clearly express our research area, we have removed the inset on the left-hand map and added relevant information in the caption.

L126-128:

[Figure]

**Fig. 1** Study area and location of sampling points(a. The location of the Xiying River Basin in China; b. The terrain and sampling points of the Xiying River Basin)

**Methods**

**Comment:** Comment 4: Section 3.2. The analysis is only part of the uncertainty. Did you perform multiple extractions on the same sample to test the uncertainty associated with that. This will undoubtedly be higher and needs to be considered.

Response: During the sampling process, we collected duplicate samples to improve the accuracy of the experiment.

That is good but you need to explain this in the text

**Response:** We have added instructions for duplicate samples in the manuscript.

L140-144: Three duplicate samples were collected for each soil layer. We placed the collected soil sample into a 50 mL glass bottle, sealed the bottle mouth with Parafilm and marked the sampling date. We froze the sample for storage until experimental analysis. Each sample was collected separately in an aluminum box.

**Results**

**Comment:** Comment 1: L175-178. How precise are these values (i.e. is the 1dp precision warranted)? What was the rainfall during those times?

Response: We use FAO Penman-Monteith to calculate the potential daily evapotranspiration (possible evapotranspiration) in the study area (the software calculation results are kept to three decimal places, and we keep one decimal place in the study). This illustrates the date when the maximum and minimum values appear, and there may be no rainfall on that day.

Regardless of the calculations, I am still sceptical that the results are that accurate.

Overall, however, the presentation of the Results and Discussion is much better and the added details help get the message across.

**Response:** We checked the calculation process and accuracy again, and the results show that it is reliable.

**Conclusions**

**Comment:** As with the discussion, the links to the study are not well made. In some ways this material is less general than some of the latter parts of the Discussion (Section 5.3) and it would be worth reordering so that you have the more general ideas at the end.

Response: Based on the opinions of the three reviewers, we re-summarized the conclusions of the manuscript.

This section still does not explain the true value of the study and why it is important to a broader readership. What is it that you have done here that informs work elsewhere? This is important for a paper in an international journal.

**Response:** Based on the findings of the study, we describe its broader value and reference for future research.

L535-543: The research results can be applied to arid and semi-arid alpine regions and have reference significance for latitude and longitude differentiation. This study mainly emphasized the Spatio-temporal heterogeneity of soil water evaporation, infiltration, and water storage in different vegetation zones. These results are of great value for understanding regional hydrological processes and ecological restoration services in environmentally fragile areas. Furthermore, we hope this study can be used as a basic statement because we continue to use stable water isotopes as a data source to understand hydrological processes from the perspective of process mechanisms.

---

## Author Response (AR4)

Revision Notes

Dear Editor:

Thank you for your comments concerning our manuscript entitled "Evaporation, infiltration and storage of soil water in different vegetation zones in the Qilian Mountains: A stable isotope perspective" (Manuscript Number: Hess-2021-376). According to your comments, we have made careful revised the manuscript. The main corrections are as follows.

**Responses to the editor's comments:**

**Thank you very much for submitting your replies to the excellent comments made by the reviewers, and for the revised version of the manuscript. As I stated before, this theme is very interesting and will enrich this special issue. In my opinion, this manuscript can be published after the authors carry out the following technical corrections:**

**Comment 1:** Line 25: Use "⋯vegetation zone…" instead of "…vegetation zone' s…".

**Response:** We have used "⋯vegetation zone…" instead of "…vegetation zone' s…".

**Comment 2:** Line 86: Remove the word "flux".

**Response:** We removed "flux".

**Comment 3:** Line 96: Use "temperature" instead of "temperatures".

**Response:** We have used "temperature" instead of "temperatures".

**Comment 4:** Line 102: (Suggestion) Use "We hope this study⋯" instead of "It is hoped that this study⋯".

**Response:** We have used "We hope this study⋯" instead of "It is hoped that this study⋯".

**Comment 5:** Line 107: Please check " ⋯ typical vertical vegetation zones", as vertical vegetation is commonly used to address building green facades.

**Response:** In order to facilitate the reader's understanding, we have modified "...typical vertical vegetation zones" as "... mountain vegetation zones".

**Comment 6:** Line 217: (Suggestion) Fig. 2 caption could be more descriptive.

**Response:** We have revised the title of Fig. 2, the new title is "Climatic and hydrological conditions of Xiying River basin"

**Comment 7:** Line 322: Correct font type/size of Fig. 5 caption.
**Response:** This issue has been corrected.

**Comment 8:** Lines 512-513: Use "Soil water evaporation in spring and summer, and insufficient precipitation during the drought period, were the main⋯".
**Response:** We have used "Soil water evaporation in spring and summer, and insufficient precipitation during the drought period, were the main⋯" instead of "Soil water evaporation in spring and summer and insufficient precipitation during the drought period were the main …".

**Comment 9:** Line 515: (Suggestion) Remove the word "results".
**Response:** We removed "results".

**Comment 10:** Lines 521-522: Please check this sentence: "The alpine meadow and coniferous forest zones were enriched in precipitation".
**Response:** In order to facilitate the reader's understanding, we have modified "The alpine meadow and coniferous forest zones were enriched in precipitation" as "The alpine meadow and coniferous forest zones have many precipitation events. ".

**Comment 11:** Line 536: (Suggestion) Use "⋯and can be valuable for latitude⋯" instead of "⋯and have reference significance for latitude⋯".
**Response:** We have used "⋯and can be valuable for latitude⋯" instead of "⋯and have reference significance for latitude⋯".

**Comment 12:** Line 537: Use "spatiotemporal" instead of "Spatio-temporal".
**Response:** We have used "spatiotemporal" instead of "Spatio-temporal".

**Comment 13:** Lines 538-539: (Suggestion) Use "These results are important for⋯" instead of "These results are of great value for⋯".

**Response:** We have used "These results are important for⋯" instead of "These results are of great value for⋯".

**Comment 14:** Lines 540-542: I find this sentence awkward. I would suggest eliminating it, or to simplify it for easiness of reading.
**Response:** After careful consideration, we have decided to delete this sentence.

**Comment 15:** Line 543: References must be moved to the end of the manuscript (it must be the last section of the manuscript).
**Response:** References have been moved to the end of the manuscript.

**More, we have further checked and revised on the use of English throughout the entire manuscript.**